# Global Optimality for Non-linear Constrained Restoration Problems via Invexity

**Samuel Pinilla, & Jeyan Thiyagalingam**

*Scientific Computing Department, Science and Technology Facilities Council
`samuel.pinilla@stfc.ac.uk`

## Abstract

Signal restoration is an important constrained optimization problem with significant applications in various domains. Although non-convex constrained optimization problems have been shown to perform better than convex counterparts in terms of reconstruction quality, convex constrained optimization problems have been preferred for their global optima guarantees. Despite the success of non-convex methods in many applications, it is not an overstatement to say that there is little or no hope for non-convex problems to ensure global optima. In this paper, for the first time, we propose a family of *invex functions* for handling constrained inverse problems using the non-convex setting along with guarantees for their global optima - *invex function* is a mapping where any critical point is a global minimizer. We also develop relevant theories to extend the global optima guarantee to a family of quasi-invex functions - the largest set of optimizable mappings. Our theoretical results show that the proposed family of invex and quasi-invex functions can aid in extending existing convex optimization algorithms, such as the alternating direction of multipliers and accelerated proximal gradient methods. Moreover, our numerical tests show that the proposed family of invex/quasi-invex functions overcome the performance of convex mappings in terms of reconstruction quality across several relevant metrics and imaging tasks.

## 1 Introduction

Signal restoration is an optimization problem that seeks to estimate an unknown original signal $\boldsymbol{x} \in \mathbb{R}^n$ from a set of noisy observations $\boldsymbol{y} \in \mathbb{R}^m$. This optimization problem falls under the purview of inverse problems, which are common across all disciplines, from physics (Mountrakis et al., 2011; Vouras et al., 2022), to medical imaging (Ginat & Gupta, 2014; Pinilla et al., 2022b) to signal processing (Pinilla et al., 2023), and to machine learning (Harris et al., 2020; Wei et al., 2022). In general, an inverse problem can be formulated as follows:

$$\underset{\boldsymbol{x} \in \mathbb{R}^n}{\text{minimize}} \quad g(\boldsymbol{x}) \ \text{ subject to } f(\boldsymbol{x}) \leq \epsilon, \tag{1}$$

where $\epsilon > 0$, $f(\boldsymbol{x})$ is a mapping constructed as a reconstruction error (fidelity term), and $g(\boldsymbol{x})$ is a regularizer to model the priors of the signal such as sparsity, low-rankness, and smoothness (Beck, 2017, Chapter 10). Here, $f(\boldsymbol{x})$ is often considered to be differentiable (i.e., smooth), while $g(\boldsymbol{x})$ is not. For some imaging tasks, the fidelity term has the form of $f(\boldsymbol{A}\boldsymbol{x} - \boldsymbol{y})$, where $\boldsymbol{A} \in \mathbb{R}^{m \times n}$. When $m < n$ for $\boldsymbol{A}$, the inverse problem reduces to *compressive sensing*. This technique is extensively exploited in many areas such as microscopy, optical imaging, and spectroscopy (Arce et al., 2013; Jerez et al., 2020; Guerrero et al., 2020).

Solving (1), a guarantee for global optimality is of paramount importance to ensure that the best-possible solution is found subject to the relevant constraints (Parikh & Boyd, 2014; Pinilla et al., 2022a). Convex constructs of $f(\boldsymbol{x}), g(\boldsymbol{x})$, as demonstrated in the literature (Beck & Teboulle, 2009; Beck, 2017), are helpful for seeking guaranteed global optima, when either the measurements can be considered as ideal and noiseless (i.e., $\epsilon = 0$) (Parikh & Boyd, 2014) or when the regularizer $g(\boldsymbol{x})$ exactly models the priors of the data. However, real-world problems do not fit this assumption, where noise is rather prevalent in all measurements (i.e., $\epsilon > 0$), and convex regularizers can easily inaccurately model the sparsity, low-rankness, smoothness, and abnormal patterns in the data (Candes et al., 2008; Laghrib et al., 2023). Although approaches with assumptions tied to noise models (such as Gaussianity) perform better than plain convex approaches, they too fail as the real-world noise models often deviate from these assumptions (Sun & Pong, 2022; Tankovich et al., 2021).

---

*Rutherford Appleton Laboratory, Harwell, UK.

Table 1: State-of-the-art proposition for $f(\boldsymbol{x})$, and $g(\boldsymbol{x})$ in optimization problem (1).

| Ref | Type of function | | Global optima |
|---|---|---|---|
| | $f(\boldsymbol{x})$ | $g(\boldsymbol{x})$ | |
| (Sun & Pong, 2022) | Geman-McClure | SCAD | No |
| (Carrillo et al., 2010) | Huber loss | Log-sum penalty | No |
| (Carrillo et al., 2016) | Cauchy loss function | $\ell_1$-norm | No |
| (Pinilla et al., 2022a) | $\ell_2$-norm | invex | **Yes** |
| **Proposed** | invex/quasi-invex | invex/quasi-invex | **Yes** |

An alternative approach is to consider non-convex mappings for $f(\boldsymbol{x})$ and $g(\boldsymbol{x})$. From (Kaul & Kaur, 1985), it can be seen that there exists a large collection of mappings that can be well-optimized compared to pure convex settings. We summarize important non-convex mappings from the literature in Table 1. These are a set of well-accepted mappings that perform better than convex ones in terms of restoration quality (Pinilla et al., 2022a; Sun & Pong, 2022). Various regularizers have been considered in the literature to overcome the limitations of pure convex approaches, such as the $\ell_1$-norm, total-variation (Chambolle, 2004; Selesnick et al., 2020) and nuclear norm (Candès & Recht, 2006), which are the continuous and convex surrogates of the $\ell_0$-pseudo norm and rank, respectively (Fu, 2014). More specifically, several interpolations between the $\ell_0$-pseudonorm and the $\ell_1$-norm have been explored as regularizers including the $\ell_p$-quasinorms (where $0 < p < 1$) (Marjanovic & Solo, 2012), Capped-$\ell_1$ penalty (Zhang et al., 2007), Log-Sum Penalty (Candes et al., 2008), Convex-concave penalty (Kovalev & Gasnikov, 2022), Minimax Concave Penalty (Zhang, 2010), Geman Penalty (Geman & Yang, 1995), and the Smoothly Clipped Absolute Deviation (SCAD) (Fan & Li, 2001). Furthermore, $\ell_2$-norm incorporated as part of the constraint in (1) is often replaced by other loss functions to reflect different noise models or robustness requirements in the noisy data $\boldsymbol{y}$, such as those outlined in (Carrillo et al., 2016; Zoubir et al., 2012). Concrete alternative reconstruction error functions include the Cauchy loss (Carrillo et al., 2016), the Huber function (Hedman et al., 2021), and the Tukey biweight loss (Mosteller & Tukey, 1977). However, while offering outstanding restoration quality, none of these can offer a guarantee for global optima, which would have been ideal for providing the soundness of these approaches.

The lack of approaches for guaranteeing the global minima in non-convex settings leaves any optimization-based algorithmic solution incomplete and non-unique. It hence cannot be categorically accepted as the best possible solution despite their improved performance over convex functions. Motivated by this problem of seeking guarantees for global optimality when solving non-convex problems, we investigate broader classes of functions that exploit the concept of *invexity* and *quasi-invexity* (Hanson, 1981; Reiland, 1990), which have been initially defined in early 1980s (Hanson, 1981). The specialty of invex functions is that *any critical point in an invex function is a global minimizer itself* (Pinilla et al., 2022a). Quasi-invex mappings generalize invex functions and hence are the largest set of optimizable functions as outlined in (Kaul & Kaur, 1985). This places the quasi-invex functions in a special and unique position for handling inverse problems. Although mathematical implications of quasi-invex (and hence invex) functions are well developed (Syed et al., 2013; Barik & Honorio, 2021; Pinilla et al., 2022a), their practical utility is less understood (Zălinescu, 2014). To the best of our knowledge, there is no existing body of work in the invex literature on applying and developing quasi-invex (and invex)-based methods for seeking global optimality in constrained optimization problems, such as signal restoration.

In this paper, we focus on demonstrating the practical utility of assuming $f(\boldsymbol{x}), g(\boldsymbol{x})$ to be invex/quasi-invex to handle inverse problems as in (1), with guarantees for global optima. Hence, in contrast with results in (Pinilla et al., 2022a), we make the following key contributions in:

- We develop a set of mathematical tools to construct a family of $f(\boldsymbol{x})$ and $g(\boldsymbol{x})$ in (1) with proved invexity/quasi-invexity for model-based inverse imaging problems. Additionally, we prove that our proposed family of invex/quasi-invex functions is closed under summation - a result that has not been available before in the invex literature of optimization (Zălinescu, 2014).

- We prove global optima under the invex/quasi-invex settings. These results extend the capability of *alternating direction method of multipliers* and *accelerated proximal gradient* methods to handle non-convex cases otherwise limited to convex settings. We also report their convergence rates, which show both methods using invex/quasi-invex mappings are as efficient as solving the well-established convex optimization problems;

- We conduct an effective evaluation of the proposed approach in handling a number of signal restoration problems against state-of-the-art algorithms and baselines.

## 2 Preliminaries

The $i$-th entry of a vector $\boldsymbol{w}$, is $\boldsymbol{w}[i]$. For vectors, $\|\boldsymbol{w}\|_p$ is the $\ell_p$-norm. An open ball is defined as $B(\boldsymbol{x}; r) = \{\boldsymbol{y} \in \mathbb{R}^n : \|\boldsymbol{y} - \boldsymbol{x}\|_2 < r\}$. The operation $\mathrm{conv}(\mathcal{A})$ represents the convex hull of the set $\mathcal{A}$. We use $\sigma_i(\boldsymbol{W})$ to denote the $i$-th singular value of a matrix $\boldsymbol{W}$ in descending order.

We present several concepts needed for the development of this paper starting with the definition of a locally Lipschitz continuous function.

**Definition 1.** A function $f : \mathbb{R}^n \to \mathbb{R}$ is locally Lipschitz continuous at a point $\boldsymbol{x} \in \mathbb{R}^n$ if there exist scalars $K > 0$ and $\epsilon > 0$ such that for all $\boldsymbol{y}, \boldsymbol{z} \in B(\boldsymbol{x}, \epsilon)$ we have $|f(\boldsymbol{y}) - f(\boldsymbol{z})| \leq K\|\boldsymbol{y} - \boldsymbol{z}\|_2$.

Since the ordinary directional derivative being the most important tool in optimization does not necessarily exist for locally Lipschitz continuous functions, it is required to introduce the concept of subdifferential (Bagirov et al., 2014) which is calculated in practice as follows.

**Theorem 1.** (Bagirov et al., 2014, Theorem 3.9) Let $f : \mathbb{R}^n \to \mathbb{R}$ be a locally Lipschitz continuous function at $\boldsymbol{x} \in \mathbb{R}^n$, and define $\Omega_f = \{\boldsymbol{x} \in \mathbb{R}^n | f$ is not differentiable at the point $\boldsymbol{x}\}$. Then the subdifferential of $f$ is given by

$$\partial f(\boldsymbol{x}) = \mathrm{conv}\ (\{\boldsymbol{\zeta} \in \mathbb{R}^n| \text{ exists } (\boldsymbol{x}_i) \in \mathbb{R}^n \setminus \Omega_f \text{ such that } \boldsymbol{x}_i \to \boldsymbol{x} \text{ and } \nabla f(\boldsymbol{x}_i) \to \boldsymbol{\zeta}\}). \quad (2)$$

The notion of subdifferential is given for locally Lipschitz continuous functions because it is always nonempty (Bagirov et al., 2014, Theorem 3.3). Based on this, we present the concept of invexity.

**Definition 2.** Let $f : \mathbb{R}^n \to \mathbb{R}$ be locally Lipschitz; then $f$ is invex if there exists a function $\eta : \mathbb{R}^n \times \mathbb{R}^n \to \mathbb{R}^n$ such that $\forall \boldsymbol{x}, \boldsymbol{y} \in \mathbb{R}^n, \forall \boldsymbol{\zeta} \in \partial f(\boldsymbol{y})$ we have $f(\boldsymbol{x}) - f(\boldsymbol{y}) \geq \boldsymbol{\zeta}^T \eta(\boldsymbol{x}, \boldsymbol{y})$.

It is well known that a convex function simply satisfies this definition for $\eta(\boldsymbol{x}, \boldsymbol{y}) = \boldsymbol{x} - \boldsymbol{y}$. The following theorem (Mishra & Giorgi, 2008, Theorem 4.33) makes connection between invexity and its well-known optimum property which motivates the design of invex optimization problems.

**Theorem 2.** (Mishra & Giorgi, 2008, Theorem 4.33) Let $f : \mathbb{R}^n \to \mathbb{R}$ be locally Lipschitz. Then the following statements are equivalent: 1) $f$ is invex. 2) Every point $\boldsymbol{y} \in \mathbb{R}^n$ that satisfies $\boldsymbol{0} \in \partial f(\boldsymbol{y})$ is a global minimizer of $f$.

We finalize this section by introducing a generalization of invexity; indeed (Reiland, 1990) introduced also the following classes of generalized convex functions.

**Definition 3.** Let $f : \mathbb{R}^n \to \mathbb{R}$ be locally Lipschitz; then $f$ is quasi-invex if there exists $\eta : \mathbb{R}^n \times \mathbb{R}^n \to \mathbb{R}^n$ such that $\forall \mathbf{x}, \mathbf{y} \in \mathbb{R}^n, \forall \boldsymbol{\zeta} \in \partial f(\mathbf{y})$ we have $f(\mathbf{x}) - f(\mathbf{y}) \leq 0 \implies \boldsymbol{\zeta}^T \eta(\mathbf{x}, \mathbf{y}) \leq 0$.

## 3 Invex and Quasi-Invex Function Sets

We address the necessity of handling the issues around real-world data by constructing invex/quasi-invex functions for data fidelity loss $f(\boldsymbol{x})$, and regularizer $g(\boldsymbol{x})$ functions. To this end, we first define the family of functions this paper investigates and their properties.

**Definition 4.** (Admissible Function) Let $h : \mathbb{R}^n \to \mathbb{R}$ such that $h(\boldsymbol{x}) = \sum_{i=1}^n s(|\boldsymbol{x}[i]|)$, where $s : [0, \infty) \to [0, \infty)$ and $s'(w) > 0$ for $w \in (0, \infty)$. If $s$ with $s(0) = 0$ such that $s(w)/w^2$ is non-increasing on $(0, \infty)$, then $h(\boldsymbol{x})$ is said to be an *admissible function*.

Definition 4 places a requirement on $f(\boldsymbol{x})$ and $g(\boldsymbol{x})$ in (1) to operate element-wise on positive values. This condition is mild since it holds for the vast majority of inverse problem formulations, including loss functions in deep learning, whether they are convex or non-convex (Carrillo et al., 2016; Wen et al., 2018; Sun & Pong, 2022). Using element-wise mapping offers notable advantages, especially in developing high-performance algorithms. One key advantage lies in the ability to parallelize, effectively reducing runtimes, making them suitable for addressing large-scale problems (Parikh & Boyd, 2014; Wang & Chen, 2022; Zoubir et al., 2012). Other conditions in Definition 4 are crucial as they establish a vital connection to invexity, which we discuss next in Theorem 3 (see Appendix A for the proof).

**Theorem 3.** Let $f, g : \mathbb{R}^n \to \mathbb{R}$ be two admissible functions as in Definition 4, such that $f(\boldsymbol{x}) = \sum_{i=1}^n s_f(|\boldsymbol{x}[i]|)$, and $g(\boldsymbol{x}) = \sum_{i=1}^n s_g(|\boldsymbol{x}[i]|)$. Then the following holds: i) $f(\boldsymbol{x})$, and $g(\boldsymbol{x})$ are invex; ii) $h_s(\boldsymbol{x}) = \alpha f(\boldsymbol{x}) + \beta g(\boldsymbol{x})$ is an admissible function (therefore invex) for every $\alpha, \beta \geq 0$; iii) $h_c(\boldsymbol{x}) = \sum_{i=1}^n (s_f \circ s_g)(|\boldsymbol{x}[i]|)$ is an admissible function. In addition to these, if we assume that $s'_f(w), s'_g(w) \geq 0$ for $w \in (0, \infty)$, then $f(\boldsymbol{x}), g(\boldsymbol{x})$, and $h_s(\boldsymbol{x}), h_c(\boldsymbol{x})$ are all quasi-invex.

The essential breakthrough of Theorem 3 compared with results in the invex literature such as (Pinilla et al., 2022a) lies in the fact that Definition 4 introduces *the first family of invex/quasi-invex functions closed under summation*. This implies that the sum of invex/quasi-invex functions is also invex/quasi-invex. The significance of this result manifests on two fronts. Firstly, Theorem 3 paves the way for establishing theoretical guarantees for global optima of (1) beyond the realm of convexity. Secondly, it bestows practical benefits to practitioners, as both Definition 4 and the third statement of Theorem 3 offer a systematic methodology for constructing invex/quasi-invex functions. These results also give away another side consequence: for any admissible function $f(\boldsymbol{x}) = \sum_{i=1}^{n} s_f(|\boldsymbol{x}[i]|)$ such that $s_f'(w) \geq 0$, then $f$ is not invex. The proof stems from the fact that $s_f'(w)$ can be zero in non-minimizer values, leading to its quasi-invexity. In the following sections, we leverage the results in Theorem 3 to present illustrative examples of invex/quasi-invex functions that apply in diverse scientific domains.

## 3.1 INVEX REGULARIZERS

In addition to the list of original invex regularizer functions presented in (Pinilla et al., 2022a, Lemma 1), consider the following Theorem 4 (See Appendix B for proof).

**Theorem 4.** For $p \in (0, 1]$, $\epsilon > 0$, the following mapping

$$(LL_p\text{-}regularizer) \quad g(\boldsymbol{x}) = \sum_{i=1}^{n} \log(1 + (|\boldsymbol{x}[i]| + \epsilon)^p), \tag{3}$$

is an admissible function. Additionally, let $\boldsymbol{D} \in \mathbb{R}^{(n-1) \times n}$ be the first-order difference matrix (further details in Appendix B). If $g$ is an admissible function, then $g_{TV}(\boldsymbol{x}) = g(\boldsymbol{D}\boldsymbol{x})$ is invex.

The key to prove Theorem 4 is centred around the fact that any admissible function is invex, and it is a point-wise non-increasing when divided by a quadratic mapping. It is worth mentioning that the invex function presented in equation (3) is one of the pivotal functions in the domain of compressive sensing (Keshavarzian et al., 2019). Although the analysis of equation (3) is valid with or without $\epsilon$, we prefer to retain $\epsilon$ to satisfy Definition 1. Although the literature identified this function as non-convex (Wang & Chen, 2022; Liu et al., 2018), there is no notion of invexity or proof of invexity.

The second part of Theorem 4 presents invex generalizations of the classical convex total-variation (TV) regularizer (Chambolle, 2004; Selesnick et al., 2020) with the $\ell_1$-norm replaced by invex functions - missing result in the invex literature such as (Pinilla et al., 2022a). TV-based regularization is effective in various applications such as computer tomography (Sun et al., 2019), hyperspectral imaging (Iordache et al., 2012; Wang et al., 2023), and computer vision (Estrela et al., 2016), where the goal is purely signal restoration. Classical TV-based algorithms suffer from an issue where the $\ell_1$-norm tends to underestimate the amplitudes of signal discontinuities owing to the limitation of the $\ell_1$-norm regularization itself (Selesnick et al., 2020). Several studies in the literature (Liu et al., 2018; Selesnick et al., 2020) show that $\ell_p$-quasinorm instead of $\ell_1$-norm can significantly address this issue, albeit the literature not demonstrating the benefits ascribed to the invexity. Again, similar to equation (3), the literature identified this function as non-convex (Wang & Chen, 2022; Liu et al., 2018) without any notion of these functions being invex or proof of invexity.

Driven by the success of equation (3) in compressive sensing (Keshavarzian et al., 2019), we propose its TV-like regularizer version. Further insights of Theorem 4, including the extension of TV-like regularizers to two- and three-dimensional signals, are presented in Section B.1.

## 3.2 INVEX FIDELITY LOSSES

Now, to complement the list of invex regularizers stated in Section 3.1, we present five invex data fidelity functions $f(\boldsymbol{x})$ in the following theorem.

**Theorem 5.** All the following functions for $c, \delta > 0$, and $\alpha \in \mathbb{R}$ are admissible

$$(Cauchy) \quad f(\boldsymbol{x}) = \sum_{i=1}^{m} \log\left(1 + \frac{\boldsymbol{x}^2[i]}{\delta^2}\right) \tag{4}$$

$$(Geman\text{-}McClure) \quad f(\boldsymbol{x}) = \sum_{i=1}^{m} \frac{2\boldsymbol{x}^2[i]}{\boldsymbol{x}^2[i] + 4\delta^2} \tag{5}$$

$$(Welsh) \quad f(\boldsymbol{x}) = \sum_{i=1}^{n} 1 - \exp\left(\frac{-\boldsymbol{x}^2[i]}{2\delta^2}\right) \tag{6}$$

$$(Adaptive\ robust) \quad f(\boldsymbol{x}) = \sum_{i=1}^{n} \frac{|\alpha - 2|}{\alpha} \left( \left( \frac{(\boldsymbol{x}[i]/c)^2}{|\alpha - 2|} + 1 \right)^{\alpha/2} - 1 \right) \tag{7}$$

$$f(\boldsymbol{x}) = \sum_{i=1}^{m} \log\left(1 + \boldsymbol{x}^2[i]\right) - \frac{\boldsymbol{x}^2[i]}{2\boldsymbol{x}^2[i] + 2} \tag{8}$$

Fidelity losses given in equations (4)-(7) have existed in the literature without ever being identified as invex functions or the benefits being ascribed to their invexity. Equation (6) was identified as an invex function, and its invexity was proved in (Syed et al., 2013; 2014). Equations (4)-(7) are found in the literature replacing the traditional $\ell_2$-norm. This was to account for different noise models and improve the solution's robustness towards noisy measurements $\boldsymbol{y}$ in (1). Equations (4)-(7) have been used to improve the performance of the original Variational Autoencoder (Kingma & Welling, 2014) by incorporating equations (4)-(6) as robust loss functions in (Barron, 2019). Finally, equation (8) is a novel invex function we propose to provide more flexibility around losses (harder to achieve in a convex setting). The proof of Theorem 5 is deferred to Appendix C.

### 3.3 Quasi-invex Regularizers

We list three quasi-invex function constructs for $g(\boldsymbol{x})$ in Theorem 6 below.

**Theorem 6.** For constants $\lambda, p \in (0, 1]$, and $\epsilon > 0$, the following mappings are quasi-invex

$$g_\theta(\boldsymbol{x}) = \lambda \sum_{i=1}^{n} r_\theta(\boldsymbol{x}[i]); \; \theta \geq 1, \; r_\theta(\boldsymbol{x}[i]) = \begin{cases} \lambda|\boldsymbol{x}[i]| - \boldsymbol{x}^2[i]/(2\theta) & \text{if } |\boldsymbol{x}[i]| < \theta\lambda \\ \theta\lambda^2/2 & \text{if } |\boldsymbol{x}[i]| \geq \theta\lambda \end{cases} \tag{9}$$

$$g_\theta(\boldsymbol{x}) = \sum_{i=1}^{n} \min\left(\log(1 + (|\boldsymbol{x}[i]| + \epsilon)^p), \theta\right); \; \theta > 0. \tag{10}$$

Further, let $\boldsymbol{D} \in \mathbb{R}^{(n-1)\times n}$ be the first-order difference matrix. If $g$ is admissible such that $g(\boldsymbol{x}) = \sum_{i=1}^{n} s_g(|\boldsymbol{x}[i]|)$ with $s'_g(w) \geq 0$ for $w \in (0, \infty)$, then $g_{TV}(\boldsymbol{x}) = g(\boldsymbol{D}\boldsymbol{x})$ is quasi-invex.

The critical aspect of proving Theorem 6 is centred around the quasi-invex construct in Theorem 3 and that it is a point-wise non-increasing when divided by a quadratic mapping. Equation (9) referred to as the Minimax Concave Penalty (MCP) (Zhang, 2010) in the non-convex literature, with a range of applications in various domains (Kovalev & Gasnikov, 2022; Zhang, 2010). Although this mapping function is standard in imaging applications (Ding et al., 2018; Song et al., 2022), the knowledge of its quasi-invexity has never been identified nor proven. For the first time, we provide formal proof of MCP being quasi-invex in Appendix D. Additional functions, such as the Clipped Absolute Deviation (SCAD) (Fan & Li, 2001), are proven to be quasi-invex in Section D. Furthermore, following an analysis analogous to Theorem 4, we can extend to two- and three-dimensional TV-like quasi-invex regularizers. In addition to these three quasi-invex constructs, we propose another way to produce quasi-invex loss functions. We present this in Lemma 1 below.

### 3.4 Quasi-invex Fidelity Losses

We finalize the exposition on quasi-invex functions by presenting the following Lemma 1 for constructing a quasi-invex function for $f(\boldsymbol{x})$. This result states that the maximum of two invex functions (not necessarily with respect to the same $\eta$) is quasi-invex. This lemma has applications in robust optimization or robust imaging problems (Vargas et al., 2018b), where controlling the accuracy of the final solution is crucial. The route to robustness is achieved by minimizing the maximum between two invex functions, which will bound the highest error to a desirable value (Boyd et al., 2004, Section 1.2.2). The proof of this lemma is deferred to Appendix E.

**Lemma 1.** Let $f, h : \mathbb{R}^n \to \mathbb{R}$ be two invex functions as in Definition 2 (not necessarily with respect to the same $\eta$). Then, $q(\boldsymbol{x}) = \max(f(\boldsymbol{x}), h(\boldsymbol{x}))$ is quasi-invex.

## 4 Applications

In this section, we demonstrate the use of invex/quasi-invex functions on several applications and guarantees for global optima, which has only been possible for convex functions. To this end, we commence by (1) ensures global optima when $f(\boldsymbol{x}), g(\boldsymbol{x})$ are admissible functions - missing result

in the invex literature such as (Pinilla et al., 2022a). The importance of this result, which we prove in Appendix F, lies in the fact that it is the first time that global optima is achieved for a family of invex constrained optimization problems (for the noiseless case $\epsilon = 0$). The key to the proof is that both $f, g$ functions are admissible.

To complement this result, we provide an extensive list of applications that would directly benefit from the global optima guarantees of (1), and the family of invex/quasi-invex constructs in Theorem 3 summarized in Table 2, and for the reasons of brevity, we selectively discuss three of the important applications. These include two imaging tasks, namely, compressive image restoration and total variation filtering, and two algorithmic applications, namely, extended Alternating Direction Method of Multipliers (ADMM) in Section 4.3, and extended accelerated proximal gradient method (APGM) which we present due to space limitations in Appendix J. Additionally, the extended guarantees of ADMM allows to state the global optimality of solving program (1) when $f(\boldsymbol{x}), g(\boldsymbol{x})$ are the family of quasi-invex functions introduced in Theorem 3.

Table 2: List of Invex and Quasi-invex functions studied in this work.

| Reference | Invex | Application | | Reference | Quasi-invex | Application |
|---|---|---|---|---|---|---|
| (Wang et al., 2022) | equation (3) | Compressive sensing | | (Song et al., 2022) | equation (9) | Model selection, ADMM |
| This work | Invex $g_{TV}(\boldsymbol{x})$ | Tomography Total variation filtering | | | | |
| (Tankovich et al., 2021) | equation (4) | Robust Learning | | This work | equation (10) | Dictionary Learning |
| (Park et al., 2021) | equation (5) | Neural Radiance Fields | | | | |
| (Psaros et al., 2022) | equations (6),(7) | Adaptive Filtering | | This work | Quasi-invex $g_{TV}(\boldsymbol{x})$ | Inpainting |
| This work | equation (8) | Supervised Learning | | | | |

### 4.1 COMPRESSIVE IMAGE RESTORATION

The reconstruction of a signal/image from a compressed set of measurements is essential in a range of scientific domains that rely on the analysis and interpretation of image content (De los Reyes et al., 2017). In this context, program (1) becomes an inverse problem that aims at recovering an image $\boldsymbol{f} \in \mathbb{R}^n$ assuming that $\boldsymbol{f}$ has a $k$-sparse representation $\boldsymbol{\theta} \in \mathbb{R}^n$ ($k \ll n$ non-zero elements) in a basis $\boldsymbol{\Psi} \in \mathbb{R}^{n \times n}$, that is $\boldsymbol{f} = \boldsymbol{\Psi x}$, in order to ensure uniqueness under some conditions. Examples of this sparse basis $\boldsymbol{\Psi}$ in imaging are the Wavelet (also Haar Wavelet) transform, cosine, and Fourier representations (Foucart & Rauhut, 2013). Hence, one can work with the abstract noisy model $\boldsymbol{y} = \boldsymbol{H \Psi x} = \boldsymbol{Ax} + \boldsymbol{\eta}$, where $\boldsymbol{\eta}$ is the noise, $\boldsymbol{A}$ encapsulates the product between acquisition matrix $\boldsymbol{H}$, and $\boldsymbol{\Psi}$, with $\ell_2$-normalized columns (Arce et al., 2013; Candès & Wakin, 2008). Under this setup, compressive sensing enables the recovery of $\boldsymbol{x}$ using much fewer samples than predicted by the Nyquist criterion (Candès & Wakin, 2008). When the regularizer $g(\boldsymbol{x})$ takes the convex form of $\ell_1$-norm, $f(\boldsymbol{Ax} - \boldsymbol{y})$ is the $\ell_2$-norm, and when the sampling matrix $\boldsymbol{A}$ satisfies the *restricted isometry property* (RIP) for any $k$-sparse vector $\boldsymbol{x} \in \mathbb{R}^n$, i.e., $(1 - \delta_{2k})\|\boldsymbol{x}\|_2^2 \le \|\boldsymbol{Ax}\|_2^2 \le (1 + \delta_{2k})\|\boldsymbol{x}\|_2^2$ for $\delta_{2k} < \frac{1}{3}$ (Foucart & Rauhut, 2013, Theorem 6.9), it has been proved that $\boldsymbol{x}$ can be exactly recovered by solving (1) with $\epsilon = 0$.

It is classically known that (1) has unique solution under convexity assumptions on $g(\boldsymbol{x})$, and $f(\boldsymbol{Ax} - \boldsymbol{y})$ for $\epsilon = 0$. We are interested in applying invex and quasi-invex regularizers and fidelity functions. It has been proven that, when $g(\boldsymbol{x})$ takes the particular invex form in (Pinilla et al., 2022a, Theorem 4), and $f(\boldsymbol{Ax} - \boldsymbol{y})$ the $\ell_2$-norm, $\boldsymbol{x}$ can be exactly recovered by program (1) for $\epsilon = 0$ (Pinilla et al., 2022a). Below in Theorem 7, we further generalize this result to all the invex/quasi-invex constructs as in Theorem 3 - missing result in the invex literature such as (Pinilla et al., 2022a).

**Theorem 7.** Assume $\boldsymbol{Ax} + \boldsymbol{\eta} = \boldsymbol{y}$, where $\boldsymbol{x} \in \mathbb{R}^n$ is $k$-sparse, the matrix $\boldsymbol{A} \in \mathbb{R}^{m \times n}$ ($m < n$) with $\ell_2$-normalized columns that satisfies the RIP condition with $\delta_{2k} < \frac{4}{\sqrt{41}}$, the noise vector $\boldsymbol{\eta}$ satisfies $f(\boldsymbol{Ax} - \boldsymbol{y}) < \epsilon$ for $\epsilon > 0$ and $\boldsymbol{y} \in \mathbb{R}^m$ is a noisy measurement vector. If $g(\boldsymbol{x})$, and $f(\boldsymbol{x})$ are admissible functions where $g(\boldsymbol{x})$ satisfies the triangle inequality (further details in Appendix G), then the solution $\boldsymbol{x}^\star$ of (1) holds $\|\boldsymbol{x} - \boldsymbol{x}^\star\|_1 \le C\beta_{k,g}(\boldsymbol{x}) + D\sqrt{\epsilon}\upsilon_{f,\eta}(\boldsymbol{x})$, where the constants $C, D > 0$ depend only on $\delta_{2k}$, and $\beta_{k,g}(\boldsymbol{x})$, $\upsilon_{f,\eta}(\boldsymbol{x})$. We show that a similar result holds for quasi-invex constructs as in Theorem 3. In addition, if $\epsilon = 0$ then $\boldsymbol{x}$ can be exactly recovered by solving (1) using both invex/quasi-invex constructs.

The key to prove Theorem 7 is centred around the properties of invex/quasi-invex constructs in Theorem 3. We present the proof of Theorem 7 in Appendix G. It also includes insights on using Lemma 1 to have more quasi-invex functions, for which Theorem 7 is also valid. We highlight that Theorem 7 is in the same form of convex recovery result (Foucart & Rauhut, 2013, Theorem 6.12), suggesting the effectiveness of our proposed family of invex/quasi-invex functions.

## 4.2 Total Variation Filtering

Total Variation (TV) regularization is a deterministic technique that safeguards discontinuities in images (Ehrhardt & Betcke, 2016; Selesnick et al., 2020). It is based on the principle that images with sharp edges have high variation while smooth transitions will show low variations (Lin et al., 2019; Liu et al., 2018). These variations are modeled using the first-order difference matrix in Theorem 4 (more details in Appendix B) and have been instrumental on a wide range of image restoration problems, including denoising (Hu & Selesnick, 2020), deblurring (Selesnick et al., 2020), inpainting (Schönlieb et al., 2009), infrared super-resolution (Liu et al., 2018), and magnetic resonance imaging (Ehrhardt & Betcke, 2016), among others (Chambolle, 2004).

Mathematically, classical TV regularization of an image $\boldsymbol{u} \in \mathbb{R}^n$ is the solution of

$$\underset{\boldsymbol{x} \in \mathbb{R}^n}{\text{minimize}} \quad g_{TV}(\boldsymbol{x}) + \frac{1}{2\lambda} \|\boldsymbol{x} - \boldsymbol{u}\|_2^2, \tag{11}$$

for $\lambda \in (0, 1]$ (typical choice in practice), where $g_{TV}(\boldsymbol{x})$ is the sum of the absolute value differences between adjacent pixel values, i.e., $\ell_1$-norm of differences between adjacent pixel values.By solving for $\boldsymbol{u}$, TV regularization encourages the output image to have smooth regions while maintaining sharp edges. The main advantage of (11) is that it has no extraneous local minima, and the minimizer is unique (Selesnick et al., 2020). However, owing to the limitations of $\ell_1$-norm (Selesnick et al., 2020; Liu et al., 2018), optimization problem in (11) can have undesirable effects, for example, in denoising, by underestimating signal discontinuities' amplitudes.

We improve program (11) by incorporating the invex/quasi-invex regularizers $g_{TV}(\boldsymbol{x})$ presented in Theorems 4 and 6 as follows (extensions for two-dimensional in Section B.1).

**Theorem 8.** Consider the optimization problem in (11) for invex and quasi-invex TV regularizers as in Theorems 4, and 6, respectively. Then: 1) The function $h(\boldsymbol{x}) = g_{TV}(\boldsymbol{x}) + \frac{1}{2\lambda} \|\boldsymbol{x} - \boldsymbol{u}\|_2^2$ is convex for a proper range of $\lambda$. 2) The resolvent operator $(\boldsymbol{I} + \lambda \partial g_{TV})^{-1}$ is a singleton.

*It is classically known that the sum of two invex functions (or quasi-invex and invex) is not necessarily invex (Mishra & Giorgi, 2008). Then, the above result is helpful to both the invexity and imaging communities, complementing Theorem 3.* We present the proof of Theorem 8 in Appendix H, which uses the fact that any admissible function is point-wise non-increasing when divided by a quadratic mapping. This analysis helps extend the original ADMM algorithm in Section 4.3. A set of mappings called *prox-regular* or *weakly convex*, can be found in the literature (Wang et al., 2019b; Rockafellar & Wets, 2009), for which Theorem 8 holds - a function is prox-regular if program (11) is convex for some $\lambda$. However, it is crucial to discern a critical disparity between prox-regular functions and admissible mappings. Prox-regularity implies quasi-invexity (proven for the first time in Appendix H). This highlights the intrinsic advantage of admissible functions in attaining global optima, showcasing their prowess in contrast to prox-regular functions.

## 4.3 Extended Alternating Direction Method of Multipliers (ADMM)

The ADMM algorithm has a special place in a range of signal restoration and imaging problems under convex settings with linear equality constraints (Boyd et al., 2011). It is particularly favored for handling large-scale problems (Glowinski, 2014) with many constraints given its amenability for parallelization (Boyd et al., 2011). Some notable applications of ADMM include blind ptychography (Li, 2023), phase retrieval (Liang et al., 2017), computer tomography (Wei et al., 2022), unrolling (Xie et al., 2019), and magnetic resonance imaging (Sun et al., 2016).

The ADMM method solves problems such as in (1), where the constraint has the form $f(\boldsymbol{Ax} - \boldsymbol{y})$ (e.g. Section 4.1), and program (11) which is rewritten using $\boldsymbol{S} \in \mathbb{R}^{m \times n}$, $\boldsymbol{P} \in \mathbb{R}^{m \times p}$ as

$$\underset{\boldsymbol{x} \in \mathbb{R}^n, \boldsymbol{z} \in \mathbb{R}^p}{\text{minimize}} \quad h_1(\boldsymbol{x}) + h_2(\boldsymbol{z}) \text{ subject to } \boldsymbol{Sx} + \boldsymbol{Pz} = \boldsymbol{y}. \tag{12}$$

We provide details on how to cast (1) into program (12 )in Appendix I. To solve (12), we form the scaled augmented Lagrangian as $\mathcal{L}_\rho(\boldsymbol{x}, \boldsymbol{z}, \boldsymbol{v}) = h_1(\boldsymbol{x}) + h_2(\boldsymbol{z}) + \frac{\rho}{2} \|\boldsymbol{Sx} + \boldsymbol{Pz} - \boldsymbol{y} + \boldsymbol{v}\|_2^2$, where $\boldsymbol{v} \in \mathbb{R}^m$ is the dual variable, and $\rho > 0$. The optimization of $\mathcal{L}_\rho(\boldsymbol{x}, \boldsymbol{z}, \boldsymbol{v})$ is summarized as

$$\boldsymbol{x}^{(t+1)} := \underset{\boldsymbol{x} \in \mathbb{R}^n}{\arg\min} \left( h_1(\boldsymbol{x}) + \frac{\rho}{2} \|\boldsymbol{Sx} + \boldsymbol{Pz}^{(t)} - \boldsymbol{y} + \boldsymbol{v}^{(t)}\|_2^2 \right)$$

$$\boldsymbol{z}^{(t+1)} := \underset{\boldsymbol{z} \in \mathbb{R}^p}{\arg\min} \left( h_2(\boldsymbol{z}) + \frac{\rho}{2} \|\boldsymbol{Sx}^{(t+1)} + \boldsymbol{Pz} - \boldsymbol{y} + \boldsymbol{v}^{(t)}\|_2^2 \right) \tag{13}$$

$$\boldsymbol{v}^{(t+1)} := \boldsymbol{v}^{(t)} + \boldsymbol{Sx}^{(t+1)} + \boldsymbol{Pz}^{(t+1)} - \boldsymbol{y}.$$

The convergence of the above algorithm is well-known and established for convex $h_1(\boldsymbol{x})$, and $h_2(\boldsymbol{z})$ (Boyd et al., 2011). However, the convergence properties of ADMM are unknown in the quasi-invex (hence invex) space. Here, we extend the ADMM algorithm so that its benefits are available to the signal restoration problems under settings based on Theorem 3, as follows.

**Theorem 9.** Let $h_1(\boldsymbol{x}), h_2(\boldsymbol{z})$ be admissible functions or quasi-invex constructs in Theorem 3, with $\rho\sigma_n(\boldsymbol{S}) \geq 1$, and $\rho\sigma_p(\boldsymbol{P}) \geq 1$. Assume there exists $(\boldsymbol{x}^*, \boldsymbol{z}^*, \boldsymbol{v}^*)$ for which $\mathcal{L}_\rho(\boldsymbol{x}^*, \boldsymbol{z}^*, \boldsymbol{v}) \leq \mathcal{L}_\rho(\boldsymbol{x}^*, \boldsymbol{z}^*, \boldsymbol{v}^*) \leq \mathcal{L}_\rho(\boldsymbol{x}, \boldsymbol{z}, \boldsymbol{v}^*)$, for all $\boldsymbol{x}, \boldsymbol{z}$, and $\boldsymbol{v}$. Then, equation (13) satisfies: i) Residual $\|\boldsymbol{r}^{(t)}\|_2 = \|\boldsymbol{S}\boldsymbol{x}^{(t)} + \boldsymbol{P}\boldsymbol{z}^{(t)} - \boldsymbol{y}\|_2 \rightarrow 0$ and $\boldsymbol{v}^{(t)} \rightarrow \boldsymbol{v}^*$ as $t \rightarrow \infty$ at a rate of $\mathcal{O}(1/t)$, where $\boldsymbol{v}^*$ is the dual optimal point; ii) $h_1(\boldsymbol{x}^{(t)}) + h_2(\boldsymbol{z}^{(t)})$ approaches the optimal value.

Proof of Theorem 9 relies on Theorem 8 which ensures uniqueness in the estimations of $\boldsymbol{x}^{(t+1)}, \boldsymbol{z}^{(t+1)}$ in equation (13). This uniqueness is not obtained by prox-regular functions which highlights the significance of our proposed family of invex/quasi-invex functions. The proof is presented in Appendix I along with applications that satisfy the mild constraints on $\boldsymbol{S}, \boldsymbol{P}$. Appendix I also provides the proximal solution for the studied invex/quasi-invex functions along with their computational time. These solutions are used to perform the numerical experiments in Section 5.

## 5 EXPERIMENTAL EVALUATION

We performed three experiments to evaluate the utility of the proposed invex/quasi-invex functions against the state-of-the-art, and we provide only the key results in Table 3. Additional set of results can be found in Appendix K. Across the experiments below we use the following metrics: absolute error (AbsError), peak signal-to-noise ratio (PSNR), root mean square error (RMSE), and structural similarity index measure (SSIM). Lower (higher) values for AbsError and RMSE (PSNR and SSIM) imply better signal restoration quality. In addition to AbsError and RMSE, we have also used square root of the AbsError (SqrtError), and log of the RMSE as additional metrics.

**Experiment 1**: Here, we focus on reconstructing three-dimensional volumes for Computed Tomography (CT). CT is a powerful imaging technique with applications to various domains, including medical and material sciences (Ginat & Gupta, 2014; Jørgensen et al., 2021). Mathematically, CT is an inverse program modeled as in (1) where the constraint is given by $f(\boldsymbol{A}\boldsymbol{x} - \boldsymbol{y}) \leq \epsilon$ with $\boldsymbol{A} \in \mathbb{R}^{m \times n}$ describing the image formation model of the noisy measurements $\boldsymbol{y} \in \mathbb{R}^m$ (Xiang et al., 2021). Many model-based recovery algorithms have been proposed to solve (1) using $g(\boldsymbol{x})$ as the $\ell_1$-norm such as the Iterative Shrinkage/Thresholding Algorithm, ADMM, and Primal-Dual Hybrid Gradient algorithm (Kamilov et al., 2023). However, the performance of the model-based strategies has been overcome by hybrid methods (i.e., unrolled networks) such as the Fast Iterative Shrinkage Thresholding Network (FISTA-Net) (Xiang et al., 2021). Here, we consider FISTA-Net as the state-of-the-art in CT due to its performance and because FISTA-Net solves (1) where $g(\boldsymbol{x})$ is the convex $\ell_1$-norm and the training loss function $f(\boldsymbol{x})$ is the $\ell_2$-norm. Then, the invex/quasi-invex regularizers and losses to be tested accordingly in this experiment are equations (3), (9), (10), and equations (4)-(8), respectively. To train FISTA-Net (with invex/quasi-invex functions above), we follow the experimental setting as in (Xiang et al., 2021). The training image dataset contains 2,378 full-dose CT images from eight of ten material sciences objects[1]. The reference images were reconstructed by `iradon` operator in MATLAB using all 720 views (Han & Ye, 2018; Jin et al., 2017). The validation set consists of 409 images and 330 slices for the test set.

**Experiment 2**: Here we focus on image spectral reconstruction (ST), which has profound applications in medical imaging, remote sensing, and material sciences (Wang et al., 2023). ST focuses on estimating the spectral reflectance of scenes from a set of RGB images captured at different wavelengths (Cai et al., 2022). Mathematically, spectral imaging is an inverse program modeled as in (1) where the constraint is given by $f(\boldsymbol{A}\boldsymbol{x} - \boldsymbol{y}) \leq \epsilon$ with $\boldsymbol{A} \in \mathbb{R}^{m \times n}$ describing the image formation model of the noisy measurements $\boldsymbol{y} \in \mathbb{R}^m$ (Arce et al., 2013). Conventional model-based methods adopt hand-crafted priors such as sparsity, total variation, and non-local similarity to regularize the reconstruction procedure Arguello et al. (2023). However, these methods result in poor generalization ability due to convex regularizers like $\ell_1$-norm, and unsatisfactory reconstruction quality (Cai et al., 2022). Deep learning methods such as the Multi-stage Spectral-wise Transformer (MST++) (Cai et al., 2022) has shown to exhibit the best performance because it does not employ regularizer. Originally the fidelity term $f(\boldsymbol{x})$ of MST++ is the $\ell_2$-norm. Then, the invex losses to be tested here are equations (4)-(8). To train MST++, we followed the setting in (Cai et al., 2022)

---

[1]Acquired with the ISIS Neutron and Muon Source at the Rutherford Appleton Laboratory

Table 3: Performance Results: Best: green, and the worst: red.

| Experiment 1 (Combination of functions used to train FISTA-Net) | | | | | | | |
|---|---|---|---|---|---|---|---|
| **Regularizer** | Metrics | equation (4) | equation (5) | equation (6) | equation (7) | equation (8) | $\ell_2$-norm |
| equation (3) | AbsError | 0.1676 | 0.1331 | 0.1543 | 0.1246 | 0.1429 | 0.1834 |
| | PSNR | 39.33 | 40.40 | 39.68 | 40.78 | 40.04 | 38.99 |
| | RMSE | 0.0101 | 0.0075 | 0.0086 | 0.0080 | 0.0093 | 0.0110 |
| | SSIM | 0.9590 | 0.9602 | 0.9608 | 0.9620 | 0.9614 | 0.9596 |
| equation (9) | AbsError | 0.3307 | 0.2909 | 0.3112 | 0.2798 | 0.3079 | 0.3464 |
| | PSNR | 37.75 | 38.32 | 38.00 | 38.65 | 38.47 | 37.50 |
| | RMSE | 0.0221 | 0.0176 | 0.0156 | 0.0132 | 0.0116 | 0.0286 |
| | SSIM | 0.9523 | 0.9551 | 0.9542 | 0.9570 | 0.9561 | 0.9533 |
| equation (10) | AbsError | 0.1859 | 0.1716 | 0.1493 | 0.1401 | 0.1601 | 0.2025 |
| | PSNR | 39.00 | 39.96 | 39.33 | 40.71 | 39.55 | 38.68 |
| | RMSE | 0.0113 | 0.0081 | 0.0101 | 0.0087 | 0.0096 | 0.0125 |
| | SSIM | 0.9577 | 0.9590 | 0.9597 | 0.9610 | 0.9603 | 0.9583 |
| (Xiang et al., 2021) $\ell_1$-norm | AbsError | 0.4106 | 0.4004 | 0.3814 | 0.3726 | 0.3906 | 0.4213 |
| | PSNR | 37.45 | 37.94 | 37.69 | 38.44 | 38.18 | 37.22 |
| | RMSE | 0.0290 | 0.0130 | 0.0222 | 0.0180 | 0.0151 | 0.0420 |
| | SSIM | 0.9510 | 0.9540 | 0.9530 | 0.9560 | 0.9550 | 0.9520 |
| Experiment 2 (Loss functions used to train MST++) | | | | | | | |
| Metrics | | equation (4) | equation (5) | equation (6) | equation (7) | equation (8) | $\ell_2$-norm |
| AbsError | | 0.1975 | 0.2056 | 0.2241 | 0.1830 | 0.1900 | 0.2145 |
| SqrtError | | 1.3752 | 1.3982 | 1.4465 | 1.4219 | 1.3530 | 1.3315 |
| RMSE | | 2.6847 | 2.9390 | 3.2465 | 2.1315 | 2.4710 | 2.2887 |
| LogRMSE | | 0.2683 | 0.2851 | 0.2943 | 0.2534 | 0.2606 | 0.2765 |
| Experiment 3 (Total variation filtering using ADMM) | | | | | | | |
| Metrics | | TV-$\ell_p$ | | Invex TV-$LL_p$ | | Quasi-invex TV-$LL_p$ | $\ell_1$-norm |
| SSIM | | 0.6137 | | 0.6320 | | 0.6227 | 0.6050 |
| MS-SSIM | | 0.9192 | | 0.9235 | | 0.9149 | 0.9106 |
| ADMM-residual | | $2.3 \times 10^{-3}$ | | $1.8 \times 10^{-3}$ | | $2.0 \times 10^{-3}$ | $2.8 \times 10^{-3}$ |

and Appendix K. The image dataset is from (Arad et al., 2022), which contains 1,000 RGB-spectral pairs. This dataset is split into the train, valid, and test subsets in the ratio of 18:1:1.

**Experiment 3**: This experiment focuses on solving (11). As we mentioned in Section 4.2 the classical total variation filtering, with global optima guarantees, is defined using the $\ell_1$-norm, then the invex/quasi-invex regularizers to be tested are the TV-regularizer version of $\ell_p$-quasinorm (TV-$\ell_p$), and equation (3) (Invex TV-$LL_p$) equation (10) (Quasi-invex TV-$LL_p$). We defer the exact details around the ADMM implementation to the supplementary material (Section K). We fixed the number of iterations of the ADMM method to 1,000 for all tested regularizers, and we included the smallest residual value $\|\boldsymbol{r}^{(t)}\|_2$ across iterations to characterize the accuracy of the restored image. The dataset employed here has 40 images[1]. These images contain the neutron attenuation properties of the object, which helps analyze the material structure. Here, in addition to SSIM, we also use Multi-scale SSIM (MS-SSIM) and residual error of the ADMM (ADMM-residual) as metrics.

## 6 DISCUSSION, LIMITATIONS AND CONCLUSIONS

In this paper, by defining a family of admissible functions with relevant properties, we identified invex/quasi-invex functions to support real-world signal processing problems - signal restoration. We then provided proof for these functions' invex/quasi-invex behaviors and global optimality beyond the realm of convexity with their convergence rate(s). Specifically, the breakthrough of Theorem 3 compared with current invex literature lies in that the sum of two proposed invex/quasi-invex functions is also invex/quasi-invex. And the results in Theorem 9 and Appendix J show that both ADMM and APGM methods with invex/quasi-invex losses, *are as efficient as solving the well-established convex optimization problems*.

We provided detailed evaluations in Table 3 and Appendix K of these functions in imaging tasks such as computer tomography, spectral imaging, and total-variation regularization. These results show significant benefits of using the proposed family of invex/quasi-invex functions from theoretical and empirical aspects. Specifically, they overcome the performance of convex mappings in terms of reconstruction quality across several relevant metrics and the studied imaging tasks. The superiority of the tested invex/quasi-invex functions is because they provide a more accurate model of the sparsity and abnormal patterns in the data than convex mappings (Sun & Pong, 2022).

While our theoretical analysis to handle signal restoration problems is the first of its kind and performs well on a number of tasks, its applications beyond imaging tasks are yet to be explored. More specifically, the application of these algorithms around deep learning research yet to be explored, which can pave a way to improve several downstream applications, such as low-rank matrix recovery and alike. As a pure algorithm, a number of things can be improved, such as decoupling the $\eta$ from $\upsilon_{f,\eta}(x)$, improving the efficiency of the algorithm beyond what ADMM and APGM can offer.

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

# A  Proof of Theorem 3

In order to establish a versatile framework for invex functions, we provide proof of invexity for a specific class of summation functions. The key result is encapsulated in the following lemma, which reveals a fundamental connection: the invexity of the summation function, obtained by applying a unidimensional real function to distinct entries of a vector, hinges upon the invexity of this underlying function.

**Lemma 2 (Invexity of Function Sum).** Let $g : \mathbb{R}^n \to \mathbb{R}$ be a function defined as

$$g(\boldsymbol{x}) = \sum_{i=1}^{n} r(\boldsymbol{x}[i]), \tag{14}$$

where $r : \mathbb{R} \to \mathbb{R}$. If $r(w)$ is an invex function then $g(\boldsymbol{x})$ is also invex.

*Proof.* Let $g : \mathbb{R}^n \to \mathbb{R}$ be a function defined as $g(\boldsymbol{x}) = \sum_{i=1}^{n} r(\boldsymbol{x}[i])$. Assume $r : \mathbb{R} \to \mathbb{R}$ is invex. Then, from the invexity of $r(w)$ we have that there exists $\eta_r : \mathbb{R} \times \mathbb{R} \to \mathbb{R}$ such that

$$r(w_1) - r(w_2) \geq \zeta_{w_2} \cdot \eta_r(w_1, w_2), \tag{15}$$

for all $w_1, w_2 \in \mathbb{R}$, and any $\zeta_{w_2} \in \partial r(w_2)$. Take $\boldsymbol{x}, \boldsymbol{y} \in \mathbb{R}^n$. Then, we have

$$r(\boldsymbol{x}[i]) - r(\boldsymbol{y}[i]) \geq \zeta_{\boldsymbol{y}[i]} \cdot \eta_r(\boldsymbol{x}[i], \boldsymbol{y}[i]), \tag{16}$$

for any $i = 1, \ldots, n$, and $\forall \zeta_{\boldsymbol{y}[i]} \in \partial r(\boldsymbol{y}[i])$. From the above inequality we conclude that for any $\boldsymbol{\zeta} \in \partial g(\boldsymbol{y})$

$$\sum_{i=1}^{n} r(\boldsymbol{x}[i]) - r(\boldsymbol{y}[i]) \geq \sum_{i=1}^{n} \zeta_{\boldsymbol{y}[i]} \cdot \eta_r(\boldsymbol{x}[i], \boldsymbol{y}[i])$$

$$g(\boldsymbol{x}) - g(\boldsymbol{y}) \geq \sum_{i=1}^{n} \zeta_{\boldsymbol{y}[i]} \cdot \eta_r(\boldsymbol{x}[i], \boldsymbol{y}[i])$$

$$g(\boldsymbol{x}) - g(\boldsymbol{y}) \geq \boldsymbol{\zeta}^T \eta(\boldsymbol{x}, \boldsymbol{y}), \tag{17}$$

such that $\eta(\boldsymbol{x}, \boldsymbol{y}) = [\eta_r(\boldsymbol{x}[1], \boldsymbol{y}[1]), \ldots, \eta_r(\boldsymbol{x}[n], \boldsymbol{y}[n])]^T$, and $\boldsymbol{\zeta} = [\zeta_{\boldsymbol{y}[1]}, \ldots, \zeta_{\boldsymbol{y}[n]}]^T$. Thus, from equation (17) the results holds. $\square$

Now we prove Theorem 3 using the result in Lemma 2.

*Proof.* We proceed by cases.

- Let $f$ be an admissible function. Then according to Definition 4 we know that $f(\boldsymbol{x}) = \sum_{i=1}^{n} s(|\boldsymbol{x}[i]|)$, for some $s : [0, \infty) \to [0, \infty)$ with $s'(w) > 0$ where $t \in (0, \infty)$. Since the structure of $f$ fits assumption in Lemma 2 then if $s(w)$ is invex then $f(\boldsymbol{x})$ is invex. Take $w_1, w_2 \in (0, \infty)$, and define $\eta : (0, \infty)^2 \to \mathbb{R}$ as

$$\eta(w_1, w_2) = \begin{cases} 0 & \text{if } s(w_1) > s(w_2) \\ \frac{s(w_1) - s(w_2)}{(\zeta_*)^2} \zeta_* & \text{otherwise ,} \end{cases} \tag{18}$$

where $\zeta_*$ is an element in $\partial s(w_2)$ of minimal absolute value which satisfies $\frac{\zeta_* \cdot \zeta}{(\zeta_*)^2} \geq 1$ for all $\zeta \in \partial s(w_2)$. The existence of $\zeta_*$ is guaranteed because $s'(w) > 0$ and therefore $0 \notin \partial s(w_2)$ (Mishra & Giorgi, 2008, Page 64). From the above equation it is clear that for all $w_1, w_2$ we have

$$s(w_1) - s(w_2) \geq \eta(w_1, w_2) \cdot \zeta_{w_2}, \quad \forall \zeta_{w_2} \in \partial s(w_2) \tag{19}$$

which means $s$ is invex. Therefore $f(\boldsymbol{x})$ is invex.

- Let $f, g$ be an admissible functions. Considering the result in previous statement, it is enough to show that $h = \beta f + \alpha g$ is an admissible function for any $\beta, \alpha \geq 0$. By definition we have that $h(0) = 0$, and $h'(w) > 0$. In addition, since both $f, g$ are positive functions it implies that $h(w)/w^2$ is non-increasing. Thus the result holds.

- Let $f, g : \mathbb{R}^n \to \mathbb{R}$ be two admissible functions as in Definition 4, such that $f(\boldsymbol{x}) = \sum_{i=1}^{n} s_f(|\boldsymbol{x}[i]|)$, and $g(\boldsymbol{x}) = \sum_{i=1}^{n} s_g(|\boldsymbol{x}[i]|)$. Define $h_c(\boldsymbol{x}) = \sum_{i=1}^{n} (s_f \circ s_g)(|\boldsymbol{x}[i]|)$. Then, observe that $(s_f \circ s_g)(0) = s_f(s_g(0)) = s_f(0) = 0$. Additionally, since $s'_f(w), s'_g(w) > 0$, then $(s_f \circ s_g)'(w) > 0$ (by the chain rule) for all $t \in (0, \infty)$. Finally, we know that $s'_f(w), s'_g(w) > 0$ implies $s_f(w), s_g(w) > 0$ to be strictly increasing. Therefore, for any $w_1 < w_2$ we know $(s_f \circ s_g)(w_1) < (s_f \circ s_g)(w_2)$, which implies $\frac{(s_f \circ s_g)(w_1)}{(w_1)^2} > \frac{(s_f \circ s_g)(w_2)}{(w_2)^2}$. Thus the result holds.

Thus we proved the first part of this theorem. $\qquad \square$

Now we proceed to prove the second part of this theorem.

*Proof.* Let $f : \mathbb{R}^n \to \mathbb{R}$ be two admissible functions as in Definition 4, such that $f(\boldsymbol{x}) = \sum_{i=1}^{n} s_f(|\boldsymbol{x}[i]|)$, and $s_f(w) \geq 0$. Define the set $\mathcal{D} = \{w \in (0, \infty)|s'(w) > 0\}$. In consequence, from first statement of this theorem proved above we know that $s_f(w)$ is invex in $\mathcal{D}$ for some function $\eta_D$. Then, define for any $w_1, w_2 \in (0, \infty)$ the following function

$$\eta(w_1, w_2) = \begin{cases} \eta_D(w_1, w_2) & \text{if } w_1, w_2 \in \mathcal{D} \\ 0 & \text{otherwise} . \end{cases} \tag{20}$$

Considering the above $\eta(w_1, w_2)$ function (which is non-zero), take $w_1, w_2 \in (0, \infty)$, and assume

$$s_f(w_1) - s_f(w_2) \leq 0. \tag{21}$$

If $w_1, w_2 \in \mathcal{D}$, then we know $s_f(w)$ is invex, which implies that

$$s_f(w_1) - s_f(w_2) \geq \eta(w_1, w_2) \cdot \zeta_{w_2} = \eta_D(w_1, w_2) \cdot \zeta_{w_2}$$
$$0 \geq \eta_D(w_1, w_2) \cdot \zeta_{w_2}, \tag{22}$$

for all $\zeta_{w_2} \in \partial s_f(w_2)$, and therefore quasi-invex. Otherwise, if either $w_1 \notin \mathcal{D}$ or $w_2 \notin \mathcal{D}$, it is clear to conclude that $\eta(w_1, w_2) \cdot \zeta_{w_2} = 0$, for all $\zeta_{w_2} \in \partial s_f(w_2)$. Thus, the result holds.

Finally, to establish the validity of the remaining two statements when $f$ is quasi-invex, we follow a similar procedure as described above. In the interest of conciseness, we omit the detailed exposition. Thus the results holds. $\qquad \square$

## B  PROOF OF THEOREM 4

We prove Theorem 4 proceeding by cases, exploiting the result in Lemma 2, because equation (3) is the sum of unidimensional real functions applied to different entries of a vector.

EQUATION (3)

*Proof.* From the definition of function $g(\boldsymbol{x})$ in equation (3) it is clear that the only aspect we have to prove is that $\log(1 + w^p)/w^2$ is non-increasing for any $w > 0$, and fixed $p \in (0, 1)$. Take $r(w) = \log(1 + w^p)$. Observe that the first derivative of $h(w) = r(w)/w^2$ is given by $h'(w) = \frac{1}{w^3} \left( \frac{pw^p}{1 + w^p} - 2\log(1 + w^p) \right)$. Since $\frac{pw^p}{1 + w^p} - 2\log(1 + w^p) < 0$, then we have that $h'(w) < 0$, which leads to conclude that $r(w)/w^2$ is non-increasing on $(0, \infty)$. Then it is clear that $r(w)/w^2$ is non-increasing on $(0, \infty)$. $\qquad \square$

Now we prove the second part of this theorem.

*Proof.* define the first-order difference matrix $\boldsymbol{D} \in \mathbb{R}^{(n-1) \times n}$ as

$$\boldsymbol{D} = \begin{bmatrix} -1 & 1 & & \\ & -1 & 1 & \\ & & \ddots & \ddots \\ & & & -1 & 1 \end{bmatrix}. \tag{23}$$

Observe that it is clear that $\boldsymbol{D}$ is full row rank. Then, define $g_{TV}(\boldsymbol{x}) = g(\boldsymbol{D}\boldsymbol{x})$ where $g$ is an admissible function. It is worth noticing that since $g$ is point-wise non-increasing when divided by a quadratic mapping, then $g$ is continuously differentiable. In addition, due to the fact that $\boldsymbol{D}$ is full row rank, we appeal to (Pinilla et al., 2022a, Lemma 2) that ensures $g_{TV}(\boldsymbol{x})$ is invex. Thus the result holds. □

### B.1 ADDITIONAL DISCUSSION ON THEOREM 4

In this section we discuss the 2D and 3D total-variation-like extensions of invex functions.

#### 2D TOTAL-VARIATION-LIKE REGULARIZER

Let be $g$ an admissible function. Then, the 2D total-variation-like regularizer based on is defined as

$$g_{TV}(\boldsymbol{x}) = g(\boldsymbol{D}_h\boldsymbol{x}) + g(\boldsymbol{D}_v\boldsymbol{x}), \tag{24}$$

where matrix $\boldsymbol{D}_h$, and $\boldsymbol{D}_v$ model the discrete derivative in the horizontal and vertical directions, respectively. Therefore, $\boldsymbol{D}_h$ is defined as in equation (23) and $\boldsymbol{D}_v$ is given as

$$\boldsymbol{D}_v = \begin{bmatrix} -1 & \cdots & 1 & & \\ & -1 & \cdots & 1 & \\ & & \ddots & \ddots & \\ & & -1 & \cdots & 1 \end{bmatrix}. \tag{25}$$

Now we prove that equation (24) is quasi-invex as follows.

*Proof.* Recall in the previous section we showed that each term $g(\boldsymbol{D}_h\boldsymbol{x})$ and $g(\boldsymbol{D}_v\boldsymbol{x})$ are invex (since $\boldsymbol{D}_h$, and $\boldsymbol{D}_v$ are full row-rank). Therefore, there exits $\eta_h(\boldsymbol{x},\boldsymbol{y})$ and $\eta_v(\boldsymbol{x},\boldsymbol{y})$ for any $\boldsymbol{x},\boldsymbol{y} \in \mathbb{R}^n$ such that

$$g(\boldsymbol{D}_h\boldsymbol{x}) - g(\boldsymbol{D}_h\boldsymbol{y}) \geq \boldsymbol{\zeta}_h^T\eta_h(\boldsymbol{x},\boldsymbol{y}) \tag{26}$$

for all $\boldsymbol{\zeta}_h \in \partial g(\boldsymbol{D}_h\boldsymbol{y})$ and

$$g(\boldsymbol{D}_v\boldsymbol{x}) - g(\boldsymbol{D}_v\boldsymbol{y}) \geq \boldsymbol{\zeta}_v^T\eta_v(\boldsymbol{x},\boldsymbol{y}) \tag{27}$$

for all $\boldsymbol{\zeta}_v \in \partial g(\boldsymbol{D}_v\boldsymbol{y})$. Then, assume

$$g_{TV}(\boldsymbol{x}) - g_{TV}(\boldsymbol{y}) \leq 0. \tag{28}$$

This leads to

$$g(\boldsymbol{D}_h\boldsymbol{x}) - g(\boldsymbol{D}_h\boldsymbol{y}) + g(\boldsymbol{D}_v\boldsymbol{x}) - g(\boldsymbol{D}_v\boldsymbol{y}) \leq 0$$
$$\boldsymbol{\zeta}_h^T\eta_h(\boldsymbol{x},\boldsymbol{y}) + \boldsymbol{\zeta}_v^T\eta_v(\boldsymbol{x},\boldsymbol{y}) \leq 0$$
$$\boldsymbol{\zeta}_g^T\eta_g(\boldsymbol{x},\boldsymbol{y}) \leq 0 \tag{29}$$

where $\boldsymbol{\zeta}_g = [\boldsymbol{\zeta}_h^T, \boldsymbol{\zeta}_v^T]^T \in \partial g_{TV}(\boldsymbol{y})$, and $\eta_{g_{TV}}(\boldsymbol{x},\boldsymbol{y}) = [\eta_h^T(\boldsymbol{x},\boldsymbol{y}), \eta_v^T(\boldsymbol{x},\boldsymbol{y})]^T$. Thus, from the above inequality we have that $g_{TV}(\boldsymbol{x})$ is quasi-invex. □

#### 3D TOTAL-VARIATION-LIKE REGULARIZER

Let $g$ be an admissible function. Then, the 3D total-variation-like regularizer is defined as

$$g_{TV}(\boldsymbol{x}) = g(\boldsymbol{D}_h\boldsymbol{x}) + g(\boldsymbol{D}_v\boldsymbol{x}) + g(\boldsymbol{D}_t\boldsymbol{x}), \tag{30}$$

where matrix $\boldsymbol{D}_h$, $\boldsymbol{D}_v$, $\boldsymbol{D}_t$ model the discrete derivative in the horizontal, vertical, and transversal (third dimension) directions, respectively. Therefore, $\boldsymbol{D}_h$ is defined as in equation (23) and $\boldsymbol{D}_v$ as in equation (25), and $\boldsymbol{D}_t$ is given as

$$\boldsymbol{D}_t = \begin{bmatrix} -1 & \cdots & \cdots & 1 & & \\ & -1 & \cdots & \cdots & 1 & \\ & & \ddots & \ddots & \ddots & \\ & & -1 & \cdots & \cdots & 1 \end{bmatrix}. \tag{31}$$

Now we prove that equation (24) is quasi-invex as follows.

*Proof.* Recall in the previous section we showed that each term $g(\boldsymbol{D}_h\boldsymbol{x})$, $g(\boldsymbol{D}_v\boldsymbol{x})$, and $g(\boldsymbol{D}_t\boldsymbol{x})$ are invex (since $\boldsymbol{D}_h$, $\boldsymbol{D}_v$, and $\boldsymbol{D}_t$ are full row-rank). Therefore, there exits $\eta_h(\boldsymbol{x},\boldsymbol{y})$, $\eta_v(\boldsymbol{x},\boldsymbol{y})$, and $\eta_t(\boldsymbol{x},\boldsymbol{y})$ for any $\boldsymbol{x},\boldsymbol{y}\in\mathbb{R}^n$ such that

$$g(\boldsymbol{D}_h\boldsymbol{x}) - g(\boldsymbol{D}_h\boldsymbol{y}) \geq \boldsymbol{\zeta}_h^T\eta_h(\boldsymbol{x},\boldsymbol{y}) \tag{32}$$

for all $\boldsymbol{\zeta}_h \in \partial g(\boldsymbol{D}_h\boldsymbol{y})$

$$g(\boldsymbol{D}_v\boldsymbol{x}) - g(\boldsymbol{D}_v\boldsymbol{y}) \geq \boldsymbol{\zeta}_v^T\eta_v(\boldsymbol{x},\boldsymbol{y}) \tag{33}$$

for all $\boldsymbol{\zeta}_v \in \partial g(\boldsymbol{D}_v\boldsymbol{y})$, and

$$g(\boldsymbol{D}_t\boldsymbol{x}) - g(\boldsymbol{D}_t\boldsymbol{y}) \geq \boldsymbol{\zeta}_t^T\eta_t(\boldsymbol{x},\boldsymbol{y}) \tag{34}$$

for all $\boldsymbol{\zeta}_t \in \partial g(\boldsymbol{D}_t\boldsymbol{y})$. Then, assume

$$g_{TV}(\boldsymbol{x}) - g_{TV}(\boldsymbol{y}) \leq 0. \tag{35}$$

This leads to

$$g(\boldsymbol{D}_h\boldsymbol{x}) - g(\boldsymbol{D}_h\boldsymbol{y}) + g(\boldsymbol{D}_v\boldsymbol{x}) - g(\boldsymbol{D}_v\boldsymbol{y}) + g(\boldsymbol{D}_t\boldsymbol{x}) - g(\boldsymbol{D}_t\boldsymbol{y}) \leq 0$$
$$\boldsymbol{\zeta}_h^T\eta_h(\boldsymbol{x},\boldsymbol{y}) + \boldsymbol{\zeta}_v^T\eta_v(\boldsymbol{x},\boldsymbol{y}) + \boldsymbol{\zeta}_t^T\eta_t(\boldsymbol{x},\boldsymbol{y}) \leq 0$$
$$\boldsymbol{\zeta}_g^T\eta_g(\boldsymbol{x},\boldsymbol{y}) \leq 0 \tag{36}$$

where $\boldsymbol{\zeta}_g = [\boldsymbol{\zeta}_h^T, \boldsymbol{\zeta}_v^T, \boldsymbol{\zeta}_t^T]^T \in \partial g_{TV}(\boldsymbol{y})$, and $\eta_{g_{TV}}(\boldsymbol{x},\boldsymbol{y}) = [\eta_h^T(\boldsymbol{x},\boldsymbol{y}), \eta_v^T(\boldsymbol{x},\boldsymbol{y}), \eta_g^T(\boldsymbol{x},\boldsymbol{y})]^T$. Thus, from the above inequality we have that $g_{TV}(\boldsymbol{x})$ is quasi-invex. □

## C  PROOF OF THEOREM 5

In this proof we seek to guarantee that the list of functions in Theorem 4 are admissible functions, and we proceed by cases.

EQUATION (4)

*Proof.* Take $r(w) = \log(1 + \frac{w^2}{\delta^2})$ for any $w \neq 0$, and fixed $\delta \in \mathbb{R}$. It is trivial to see that $r(0) = 0$, that $r(w)$ it is not identically zero, and non-decreasing on $(0,\infty)$. Then, we just need to show that $r(w)/w^2$ is non-increasing on $(0,\infty)$. Observe that the first derivative of $h(w) = r(w)/w^2$ is given by $h'(w) = \frac{2\left(\frac{w^2}{\delta^2+w^2} - \log(1+\frac{w^2}{\delta^2})\right)}{w^3}$. Since $\frac{w^2}{\delta^2+w^2} - \log(1 + \frac{w^2}{\delta^2}) < 0$, then we have that $h'(w) < 0$, which leads to conclude that $r(w)/w^2$ is non-increasing on $(0,\infty)$. Then it is clear that $r(w)/w^2$ is non-increasing on $(0,\infty)$. □

EQUATION (5)

*Proof.* Take $r(w) = \frac{2w^2}{w^2+4\delta^2}$ for any $w \neq 0$, and fixed $\delta \in \mathbb{R}$. It is trivial to see that $r(0) = 0$, that $r(w)$ it is not identically zero, and non-decreasing on $(0,\infty)$. Then, we just need to show that $r(w)/w^2$ is non-increasing on $(0,\infty)$. Observe that $h(w) = r(w)/w^2$ is given by $h(w) = \frac{2}{w^2+4\delta^2}$, which leads to conclude that $r(w)/w^2$ is non-increasing on $(0,\infty)$. Then it is clear that $r(w)/w^2$ is non-increasing on $(0,\infty)$. □

EQUATION (6)

Take $r(w) = 1 - \exp(-w^2/\delta^2)$ for any $w \neq 0$, and fixed $\delta \in \mathbb{R}$. It is trivial to see that $r(0) = 0$, that $r(w)$ it is not identically zero, and non-decreasing on $(0,\infty)$. Then, we just need to show that $r(w)/w^2$ is non-increasing on $(0,\infty)$. For easy of exposition we present in Figure 1(a) the plot of $r(w)/w^2$. Then it is clear that $r(w)/w^2$ is non-increasing on $(0,\infty)$.

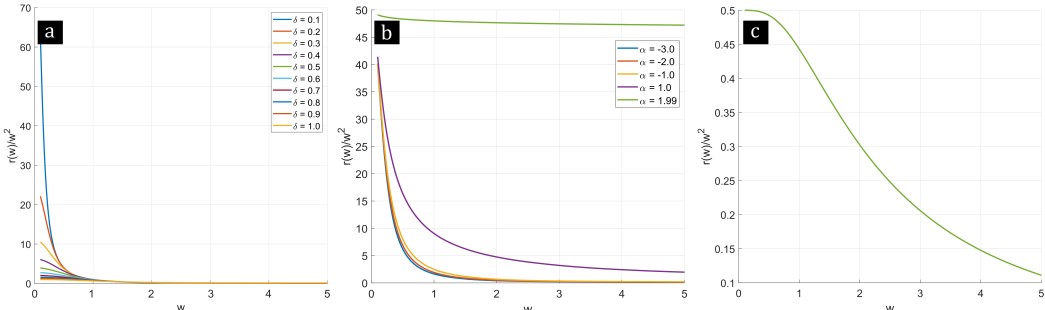

Figure 1: Plot of $r(w)/w$ for $r(w)$ being (a) equation (6), (b) equation (7) for $c = 0.1$, and (c) equation (8) and $w > 0$ to check that $r(w)/w^2$ is non-increasing on $(0, \infty)$

EQUATION (7)

Take $r(w) = \frac{|\alpha - 2|}{\alpha} \left( \left( \frac{(w/c)^2}{|\alpha - 2|} + 1 \right)^{\alpha/2} - 1 \right)$ for any $w \neq 0$, and fixed $\alpha \in \mathbb{R}$, $c > 0$. It is trivial to see that $r(0) = 0$, that $r(w)$ it is not identically zero, and non-decreasing on $(0, \infty)$. Then, we just need to show that $r(w)/w^2$ is non-increasing on $(0, \infty)$. For easy of exposition we present in Figure 1(b) the plot of $r(w)/w^2$. Then it is clear that $r(w)/w^2$ is non-increasing on $(0, \infty)$.

EQUATION (8)

Take $r(w) = \log\left(1 + w^2\right) - \frac{w^2}{2w^2 + 2}$ for any $w \neq 0$. It is trivial to see that $r(0) = 0$, that $r(w)$ it is not identically zero, and non-decreasing on $(0, \infty)$. Then, we just need to show that $r(w)/w^2$ is non-increasing on $(0, \infty)$. For easy of exposition we present in Figure 1(c) the plot of $r(w)/w^2$. Then it is clear that $r(w)/w^2$ is non-increasing on $(0, \infty)$.

## D    PROOF OF THEOREM 6

To prove that $g_\theta(\boldsymbol{x})$ is quasi-invex, for regularizers in Theorem 6, we show the existence of $\eta : \mathbb{R}^n \times \mathbb{R}^n \to \mathbb{R}^n$ satisfying Definition 3 for $g_\theta(\boldsymbol{x})$.

EQUATION (9)

*Proof.* Let $g_\theta(\boldsymbol{x}) = \lambda \sum_{i=1}^n r_\theta(\boldsymbol{x}[i])$, for $\theta \geq 1$, $\lambda \in (0, 1]$, and

$$r_\theta(w) = \begin{cases} \lambda|w| - w^2/(2\theta) & \text{if } |w| < \theta\lambda \\ \theta\lambda^2/2 & \text{if } |w| \geq \theta\lambda \end{cases} . \tag{37}$$

First, we will show that if $|w| < \theta\lambda$ then $r_\theta(w)$ is invex. In consequence, $g_\theta(\boldsymbol{x})$ is invex for any $\boldsymbol{x} \in \mathcal{D}$ (see Lemma 2), with

$$\mathcal{D} = \{\boldsymbol{z} \in \mathbb{R}^n : |\boldsymbol{z}[i]| < \theta\lambda, \forall i \in \{1, \ldots, n\}\}.$$

Once we prove this, we are able to build $\eta : \mathbb{R}^n \times \mathbb{R}^n \to \mathbb{R}^n$ such that $g_\theta(\boldsymbol{x})$ is quasi-invex for any $\boldsymbol{x} \in \mathbb{R}^n$.

Observe that $r_\theta'(w)$ for $0 < |w| < \theta\lambda$ is given by

$$r_\theta'(w) = \lambda \frac{w}{|w|} - \frac{w}{\theta}. \tag{38}$$

The above equation implies that $0 \in \partial r_\theta(w)$ only when $w = 0$. Further, since $r_\theta(0) < r_\theta(w)$ for any $0 < |w| < \theta\lambda$, then $w = 0$ is a global minimizer of $r_\theta(w)$, in the restricted domain, implying its invexity. Therefore, appealing to Lemma 2 there exists $\eta_r : \mathbb{R}^n \times \mathbb{R}^n \to \mathbb{R}^n$ such that $g_\theta(\boldsymbol{x})$ is invex in $\mathcal{D}$. It is worth mentioning that $\mathcal{D}$ is an open set, because it is the Cartesian product of $n$

open sets. This observation is important because invexity requires to compute the subgradient which is only possible on open sets. Thus, the invexity of $g_\theta(\boldsymbol{x})$ is well defined. And therefore, $g_\theta(\boldsymbol{x})$ is quasi-invex in $\mathcal{D}$.

In consequence, define the following $\eta : \mathbb{R}^n \times \mathbb{R}^n \to \mathbb{R}^n$ as

$$\eta(\boldsymbol{x}, \boldsymbol{y}) = \begin{cases} \eta_r(\boldsymbol{x}, \boldsymbol{y}) & \text{if } \boldsymbol{x}, \boldsymbol{y} \in \mathcal{D} \\ \boldsymbol{0} & \text{otherwise} \end{cases}. \tag{39}$$

Considering the above $\eta(\boldsymbol{x}, \boldsymbol{y})$ function (which is non-zero), take $\boldsymbol{x}, \boldsymbol{y} \in \mathbb{R}^n$, and assume

$$g_\theta(\boldsymbol{x}) - g_\theta(\boldsymbol{y}) \le 0. \tag{40}$$

If $\boldsymbol{x}, \boldsymbol{y} \in \mathcal{D}$, then we know $g_\theta(\boldsymbol{x})$ is invex, which implies that

$$g_\theta(\boldsymbol{x}) - g_\theta(\boldsymbol{y}) \ge \boldsymbol{\zeta}^T \eta(\boldsymbol{x}, \boldsymbol{y})$$
$$0 \ge \boldsymbol{\zeta}^T \eta(\boldsymbol{x}, \boldsymbol{y}), \tag{41}$$

for all $\boldsymbol{\zeta} \in \partial g_\theta(\boldsymbol{y})$, and therefore quasi-invex. Otherwise, if either $\boldsymbol{x} \notin \mathcal{D}$ or $\boldsymbol{y} \notin \mathcal{D}$, it is clear to conclude that $\boldsymbol{\zeta}^T \eta(\boldsymbol{x}, \boldsymbol{y}) = 0$, for all $\boldsymbol{\zeta} \in \partial g_\theta(\boldsymbol{y})$. Thus, the result holds. $\qquad\square$

EQUATION (10)

*Proof.* Let $g_\theta(\boldsymbol{x}) = \sum_{i=1}^n r_\theta(\boldsymbol{x}[i])$ be the regularizer in equation (10) for $\theta > 0$, and $r_\theta(\boldsymbol{x}[i]) = \min(r(\boldsymbol{x}[i]), \theta)$ where function $r(w)$ takes the form of

$$r(w) = \log(1 + (|w| + \epsilon)^p), p \in (0, 1), \epsilon > 0. \tag{42}$$

To prove that equation (10) is quasi-invex we first show that the unidimensional function $r(w)$ is invex and then we proceed to show the construction of $\eta : \mathbb{R}^n \times \mathbb{R}^n \to \mathbb{R}^n$. To prove that function $r(w)$ is invex observe that if $w > 0$ then we have that $\partial r(w) = \left\{ \frac{p}{(|w|+\epsilon)^{1-p} + (|w|+\epsilon)} \right\}$, which means that $0 \notin \partial r(w)$. Conversely, if $w < 0$ then $\partial r(w) = \left\{ \frac{-p}{(|w|+\epsilon)^{1-p} + (|w|+\epsilon)} \right\}$, leading to $0 \notin \partial r(w)$. Lets examinate $w^* = 0$. Note that $\lim_{w \to 0^+} r'(w) = \frac{p}{\epsilon^{1-p}+\epsilon}$, and that $\lim_{w \to 0^-} r'(w) = \frac{-p}{\epsilon^{1-p}+\epsilon}$. Additionally, since $r(w)$ is a Lipschitz continuous function, then appealing to Theorem 1 we have that $\partial r(w^* = 0) = \text{conv} \left\{ \frac{-p}{\epsilon^{1-p}+\epsilon}, \frac{p}{\epsilon^{1-p}+\epsilon} \right\} = \left[ \frac{-p}{\epsilon^{1-p}+\epsilon}, \frac{p}{\epsilon^{1-p}+\epsilon} \right]$. This means that $0 \in \partial r(0)$. Further, given the fact that $r(0) \le r(w)$ for all $w \in \mathbb{R}$, then $w^* = 0$ is a global minimizer of $r(w)$. Therefore, the function $r(w)$ is invex.

Now we show the existence of $\eta : \mathbb{R}^n \times \mathbb{R}^n \to \mathbb{R}^n$ satisfying Definition 3 for $g_\theta(\boldsymbol{x})$. First, observe that if function $r(w)$ does not reach the value of $\theta$ for any $w$, then it is clear the existence of $\eta_g : \mathbb{R}^n \times \mathbb{R}^n \to \mathbb{R}^n$ such that $g_\theta(\boldsymbol{x}) = \sum_{i=1}^n r(\boldsymbol{x}[i])$ is invex according to Lemma 2, and therefore quasi-invex. On the other hand, if we assume $r(w)$ reaches the value of $\theta$, then there exists $w^* \ge 0$ such that $r(w) < \theta$ for any $|w| < w^*$. Let $\mathcal{D} = \{\boldsymbol{z} \in \mathbb{R}^n : |\boldsymbol{z}[i]| < w^*, \forall i \in \{1, \ldots, n\}\}$. It is worth mentioning that $\mathcal{D}$ is an open set, because it is the Cartesian product of $n$ open sets. This observation is important because invexity requires to compute the subgradient which is only possible on open sets. Thus, the invexity of $g_\theta(\boldsymbol{x})$ in $\mathcal{D}$ is well defined. Now, we define the following $\eta : \mathbb{R}^n \times \mathbb{R}^n \to \mathbb{R}^n$ as

$$\eta(\boldsymbol{x}, \boldsymbol{y}) = \begin{cases} \eta_g(\boldsymbol{x}, \boldsymbol{y}) & \text{if } \boldsymbol{x}, \boldsymbol{y} \in \mathcal{D} \\ \boldsymbol{0} & \text{otherwise} \end{cases}. \tag{43}$$

Considering the above $\eta(\boldsymbol{x}, \boldsymbol{y})$ (which is non-zero), take $\boldsymbol{x}, \boldsymbol{y} \in \mathbb{R}^n$, and assume

$$g_\theta(\boldsymbol{x}) - g_\theta(\boldsymbol{y}) \le 0. \tag{44}$$

If $\boldsymbol{x}, \boldsymbol{y} \in \mathcal{D}$, then we know $g_\theta(\boldsymbol{x})$ is invex, which implies that

$$g_\theta(\boldsymbol{x}) - g_\theta(\boldsymbol{y}) \ge \boldsymbol{\zeta}^T \eta(\boldsymbol{x}, \boldsymbol{y})$$
$$0 \ge \boldsymbol{\zeta}^T \eta(\boldsymbol{x}, \boldsymbol{y}), \tag{45}$$

for all $\boldsymbol{\zeta} \in \partial g_\theta(\boldsymbol{y})$, and therefore quasi-invex. Otherwise, if either $\boldsymbol{x} \notin \mathcal{D}$ or $\boldsymbol{y} \notin \mathcal{D}$, it is clear to conclude that $\boldsymbol{\zeta}^T \eta(\boldsymbol{x}, \boldsymbol{y}) = 0$, for all $\boldsymbol{\zeta} \in \partial g_\theta(\boldsymbol{y})$. Thus, the result holds. $\qquad\square$

## D.1 Third Part of Theorem 6

*Proof.* Let $g$ be an admissible function such that $g(\boldsymbol{x}) = \sum_{i=1}^{n-1} s_g(|\boldsymbol{x}[i]|)$, and $s_g(w) \geq 0$. Define $g_{TV}(\boldsymbol{x}) = g(\boldsymbol{Dx})$ where $\boldsymbol{D}$ as defined in equation (23). Then, according to Theorem 3 we have that $g(\boldsymbol{x})$ is quasi-invex for some $\eta_g(\boldsymbol{x}, \boldsymbol{y})$, which implies that

$$g_{TV}(\boldsymbol{x}) - g_{TV}(\boldsymbol{y}) \leq 0 \implies \boldsymbol{\zeta}^T \eta_g(\boldsymbol{Dx}, \boldsymbol{Dy}) \leq 0, \tag{46}$$

for all $\boldsymbol{\zeta} \in \partial g(\boldsymbol{Dy})$. We know that matrix $\boldsymbol{D}$ is full row-rank which means that $(\boldsymbol{DD}^T)^{-1}$ exits and $(\boldsymbol{DD}^T)(\boldsymbol{DD}^T)^{-1} = \boldsymbol{I}_{n-1}$ where $\boldsymbol{I}_{n-1}$ is the identity matrix. Then, from the above inequality we obtain

$$\begin{aligned}
g_{TV}(\boldsymbol{x}) - g_{TV}(\boldsymbol{y}) \leq 0 \implies & \boldsymbol{\zeta}^T (\boldsymbol{DD}^T)(\boldsymbol{DD}^T)^{-1} \eta_g(\boldsymbol{Dx}, \boldsymbol{Dy}) \leq 0 \\
\implies & \boldsymbol{\zeta}^T \boldsymbol{D} \left( \boldsymbol{D}^T (\boldsymbol{DD}^T)^{-1} \eta_g(\boldsymbol{Dx}, \boldsymbol{Dy}) \right) \\
\implies & (\boldsymbol{D}^T \boldsymbol{\zeta})^T \left( \boldsymbol{D}^T (\boldsymbol{DD}^T)^{-1} \eta_g(\boldsymbol{Dx}, \boldsymbol{Dy}) \right) \\
\implies & (\boldsymbol{D}^T \boldsymbol{\zeta})^T \eta(\boldsymbol{x}, \boldsymbol{y}),
\end{aligned} \tag{47}$$

where $\eta(\boldsymbol{x}, \boldsymbol{y}) = \boldsymbol{D}^T (\boldsymbol{DD}^T)^{-1} \eta_g(\boldsymbol{Dx}, \boldsymbol{Dy})$. Further, since $g$ is continuously differentiable concluded from (Bagirov et al., 2014, Theorem 3.20) we know that $\boldsymbol{D}^T \boldsymbol{\zeta} \in \partial g_{TV}(\boldsymbol{y})$. Thus, from equation (47) we get

$$g_{TV}(\boldsymbol{x}) - g_{TV}(\boldsymbol{y}) \leq 0 \implies \boldsymbol{\zeta}^T \eta(\boldsymbol{x}, \boldsymbol{y}), \tag{48}$$

for all $\boldsymbol{\zeta} \in \partial g_{TV}(\boldsymbol{y})$, which implies $g_{TV}(\boldsymbol{x})$ is quasi-invex. Therefore, the result holds. $\square$

## D.2 Proof SCAD is Quasi-invex

*Proof.* Let $g_\theta(\boldsymbol{x}) = \lambda \sum_{i=1}^n r_\theta(\boldsymbol{x}[i])$, for $\theta > 2$, $\lambda \in (0, 1]$, and

$$r_\theta(w) = \begin{cases} \lambda |w| & \text{if } |w| \leq \lambda \\ \frac{-w^2 + 2\theta\lambda|w| - \lambda^2}{2(\theta-1)} & \text{if } \lambda < |w| < \theta\lambda \\ (\theta + 1)\lambda^2/2 & \text{otherwise} \end{cases}. \tag{49}$$

First, we will show that if $|w| < \theta\lambda$ then $r_\theta(w)$ is invex. In consequence, $g_\theta(\boldsymbol{x})$ is invex for any $\boldsymbol{x} \in \mathcal{D}$ (see Lemma 2), with

$$\mathcal{D} = \{\boldsymbol{z} \in \mathbb{R}^n : |\boldsymbol{z}[i]| < \theta\lambda, \forall i \in \{1, \ldots, n\}\}.$$

Once we prove this, we are able to build $\eta : \mathbb{R}^n \times \mathbb{R}^n \to \mathbb{R}^n$ such that $g_\theta(\boldsymbol{x})$ is quasi-invex for any $\boldsymbol{x} \in \mathbb{R}^n$.

Observe that $r'_\theta(w)$ for $0 < |w| < \theta\lambda$ is given by

$$r'_\theta(w) = \begin{cases} \lambda \frac{w}{|w|} & \text{if } |w| \leq \lambda \\ \frac{-w + \frac{\theta\lambda w}{|w|}}{\theta - 1} & \text{if } \lambda < |w| < \theta\lambda. \end{cases} \tag{50}$$

The above equation implies that $0 \in \partial r_\theta(w)$ only when $w = 0$. Further, since $r_\theta(0) < r_\theta(w)$ for any $0 < |w| < \theta\lambda$, then $w = 0$ is a global minimizer of $r_\theta(w)$, in the restricted domain, implying its invexity. Therefore, appealing to Lemma 2 there exists $\eta_r : \mathbb{R}^n \times \mathbb{R}^n \to \mathbb{R}^n$ such that $g_\theta(\boldsymbol{x})$ is invex in $\mathcal{D}$. It is worth mentioning that $\mathcal{D}$ is an open set, because it is the Cartesian product of $n$ open sets. This observation is important because invexity requires to compute the subgradient which is only possible on open sets. Thus, the invexity of $g_\theta(\boldsymbol{x})$ is well defined. And therefore, $g_\theta(\boldsymbol{x})$ is quasi-invex in $\mathcal{D}$.

In consequence, define the following $\eta : \mathbb{R}^n \times \mathbb{R}^n \to \mathbb{R}^n$ as

$$\eta(\boldsymbol{x}, \boldsymbol{y}) = \begin{cases} \eta_r(\boldsymbol{x}, \boldsymbol{y}) & \text{if } \boldsymbol{x}, \boldsymbol{y} \in \mathcal{D} \\ \boldsymbol{0} & \text{otherwise} \end{cases}. \tag{51}$$

Considering the above $\eta(\boldsymbol{x}, \boldsymbol{y})$ function (which is non-zero), take $\boldsymbol{x}, \boldsymbol{y} \in \mathbb{R}^n$, and assume

$$g_\theta(\boldsymbol{x}) - g_\theta(\boldsymbol{y}) \leq 0. \tag{52}$$

If $\boldsymbol{x}, \boldsymbol{y} \in \mathcal{D}$, then we know $g_\theta(\boldsymbol{x})$ is invex, which implies that

$$\begin{aligned}
g_\theta(\boldsymbol{x}) - g_\theta(\boldsymbol{y}) \geq & \boldsymbol{\zeta}^T \eta(\boldsymbol{x}, \boldsymbol{y}) \\
0 \geq & \boldsymbol{\zeta}^T \eta(\boldsymbol{x}, \boldsymbol{y}),
\end{aligned} \tag{53}$$

for all $\boldsymbol{\zeta} \in \partial g_\theta(\boldsymbol{y})$, and therefore quasi-invex. Otherwise, if either $\boldsymbol{x} \notin \mathcal{D}$ or $\boldsymbol{y} \notin \mathcal{D}$, it is clear to conclude that $\boldsymbol{\zeta}^T \eta(\boldsymbol{x}, \boldsymbol{y}) = 0$, for all $\boldsymbol{\zeta} \in \partial g_\theta(\boldsymbol{y})$. Thus, the result holds. $\square$

### D.3 Proof Capped-$\ell_1$ is Quasi-invex

*Proof.* Let $g_\theta(\boldsymbol{x}) = \lambda \sum_{i=1}^n r_\theta(\boldsymbol{x}[i])$, for $\theta > 0$, $\lambda \in (0, 1]$, and

$$r_\theta(w) = \min(|w|, \theta).$$

Now we show the existence of $\eta : \mathbb{R}^n \times \mathbb{R}^n \to \mathbb{R}^n$ satisfying Definition 3 for $g_\theta(\boldsymbol{x})$. Let $\mathcal{D} = \{\boldsymbol{z} \in \mathbb{R}^n : |\boldsymbol{z}[i]| < \theta, \forall i \in \{1, \ldots, n\}\}$. Thus, we define $\eta : \mathbb{R}^n \times \mathbb{R}^n \to \mathbb{R}^n$ as

$$\eta(\boldsymbol{x}, \boldsymbol{y}) = \begin{cases} \boldsymbol{x} - \boldsymbol{y} & \text{if } \boldsymbol{x}, \boldsymbol{y} \in \mathcal{D} \\ \boldsymbol{0} & \text{otherwise} \end{cases}. \tag{54}$$

Considering the above $\eta(\boldsymbol{x}, \boldsymbol{y})$ (which is non-zero), take $\boldsymbol{x}, \boldsymbol{y} \in \mathbb{R}^n$, and assume

$$g_\theta(\boldsymbol{x}) - g_\theta(\boldsymbol{y}) \leq 0. \tag{55}$$

If $\boldsymbol{x}, \boldsymbol{y} \in \mathcal{D}$, then we know $g_\theta(\boldsymbol{x})$ is the $\ell_1$-norm, which implies that

$$g_\theta(\boldsymbol{x}) - g_\theta(\boldsymbol{y}) \geq \boldsymbol{\zeta}^T(\boldsymbol{x} - \boldsymbol{y})$$
$$0 \geq \boldsymbol{\zeta}^T(\boldsymbol{x} - \boldsymbol{y}), \tag{56}$$

for all $\boldsymbol{\zeta} \in \partial g_\theta(\boldsymbol{y})$, and therefore quasi-invex. Otherwise, if either $\boldsymbol{x} \notin \mathcal{D}$ or $\boldsymbol{y} \notin \mathcal{D}$, it is clear to conclude that $\boldsymbol{\zeta}^T \eta(\boldsymbol{x}, \boldsymbol{y}) = 0$, for all $\boldsymbol{\zeta} \in \partial g_\theta(\boldsymbol{y})$. Thus, the result holds. $\square$

## E Proof of Lemma 1

*Proof.* Let $f, h : \mathbb{R}^n \to \mathbb{R}$ be two invex functions (not necessarily with respect to the same $\eta$). Define the function $q(\boldsymbol{x}) = \max(f(\boldsymbol{x}), h(\boldsymbol{x}))$. Now we will show that $q(\boldsymbol{x})$ satisfies Definition 3. Then, take $\boldsymbol{x}, \boldsymbol{y} \in \mathbb{R}^n$ and assume

$$q(\boldsymbol{x}) - q(\boldsymbol{y}) \leq 0. \tag{57}$$

The above inequality implies that

$$\max(f(\boldsymbol{x}), h(\boldsymbol{x})) - \max(f(\boldsymbol{y}), h(\boldsymbol{y})) \leq 0$$
$$\max(f(\boldsymbol{x}), h(\boldsymbol{x})) \leq \max(f(\boldsymbol{y}), h(\boldsymbol{y})). \tag{58}$$

Now we proceed by cases.

1. Assume $q(\boldsymbol{y}) = \max(f(\boldsymbol{y}), h(\boldsymbol{y})) = f(\boldsymbol{y})$, then from equation (58) we have

$$f(\boldsymbol{x}) \leq \max(f(\boldsymbol{x}), h(\boldsymbol{x})) \leq \max(f(\boldsymbol{y}), h(\boldsymbol{y})) = f(\boldsymbol{y})$$
$$f(\boldsymbol{x}) \leq f(\boldsymbol{y}). \tag{59}$$

Since $f$ is an invex function, then we know there exits $\eta_f : \mathbb{R}^n \times \mathbb{R}^n \to \mathbb{R}^n$ such that equation (59) implies

$$f(\boldsymbol{x}) - f(\boldsymbol{y}) \geq \boldsymbol{\zeta}^T \eta_f(\boldsymbol{x}, \boldsymbol{y})$$
$$0 \geq \boldsymbol{\zeta}^T \eta_f(\boldsymbol{x}, \boldsymbol{y}) \tag{60}$$

for all $\boldsymbol{\zeta} \in \partial f(\boldsymbol{y}) = \partial q(\boldsymbol{y})$.

2. Assume $q(\boldsymbol{y}) = \max(f(\boldsymbol{y}), h(\boldsymbol{y})) = h(\boldsymbol{y})$, then from equation (58) we have

$$h(\boldsymbol{x}) \leq \max(f(\boldsymbol{x}), h(\boldsymbol{x})) \leq \max(f(\boldsymbol{y}), h(\boldsymbol{y})) = h(\boldsymbol{y})$$
$$h(\boldsymbol{x}) \leq h(\boldsymbol{y}). \tag{61}$$

Since $h$ is an invex function, then we know there exits $\eta_h : \mathbb{R}^n \times \mathbb{R}^n \to \mathbb{R}^n$ such that equation (61) implies

$$h(\boldsymbol{x}) - h(\boldsymbol{y}) \geq \boldsymbol{\zeta}^T \eta_h(\boldsymbol{x}, \boldsymbol{y})$$
$$0 \geq \boldsymbol{\zeta}^T \eta_h(\boldsymbol{x}, \boldsymbol{y}) \tag{62}$$

for all $\boldsymbol{\zeta} \in \partial h(\boldsymbol{y}) = \partial q(\boldsymbol{y})$.

From the above discussed cases, we build $\eta : \mathbb{R}^n \times \mathbb{R}^n \to \mathbb{R}^n$ as

$$\eta(\boldsymbol{x}, \boldsymbol{y}) = \begin{cases} \eta_f(\boldsymbol{x}, \boldsymbol{y}) & \text{if } f(\boldsymbol{y}) > h(\boldsymbol{y}) \\ \mathbf{0} & \text{if } f(\boldsymbol{y}) = h(\boldsymbol{y}) \\ \eta_h(\boldsymbol{x}, \boldsymbol{y}) & \text{otherwise} \end{cases} . \tag{63}$$

Thus, from equation (59), equation (61), and the above definition of $\eta(\boldsymbol{x}, \boldsymbol{y})$, we conclude

$$q(\boldsymbol{x}) - q(\boldsymbol{y}) \leq 0 \implies \boldsymbol{\zeta}^T \eta(\boldsymbol{x}, \boldsymbol{y}) \leq 0, \tag{64}$$

$\forall \boldsymbol{x}, \boldsymbol{y} \in \mathbb{R}^n$, and $\forall \boldsymbol{\zeta} \in \partial q(\boldsymbol{y})$. Therefore the desired result holds. $\qquad \square$

## F GLOBAL OPTIMA GUARANTEES OF PROGRAM (1)

To prove the global optima guarantees of optimization problem in (1) we formulate and prove the following theorem.

**Theorem 10.** Let $f, g : \mathbb{R}^n \to \mathbb{R}$ be admissible functions as in Definition 4. Assume $\boldsymbol{x}^\star$ is a solution of program (1) for $\epsilon = 0$, then $\boldsymbol{x}^\star$ is a global minimizer.

*Proof.* Assume $\boldsymbol{x}^*$ is a minimizer of constrained optimization problem in equation (1), and define $\mathcal{D} = \{\boldsymbol{x} \in \mathbb{R}^n | f(\boldsymbol{x}) = 0\}$. This means that $\boldsymbol{x}^*$ is a minimizer of $g(\boldsymbol{x})$ in $\mathcal{D}$ i.e. $\boldsymbol{x}^* \in \mathcal{D}$. Since, $f(\boldsymbol{x})$ is an admissible function, then is invex. Therefore, $\boldsymbol{x}^*$ must be a global minimizer. Thus the result holds. $\qquad \square$

## G PROOF OF THEOREM 7

In order to prove Theorem 7 we introduce an auxiliary definition and lemmata as follows.

**Definition 5.** (*Sparseness measure* (Gribonval & Nielsen, 2007)) Let $g : \mathbb{R}^n \to \mathbb{R}$ such that $g(\boldsymbol{x}) = \sum_{i=1}^n r(\boldsymbol{x}[i])$, where $r : [0, \infty) \to [0, \infty)$ and increasing. If $r$, not identically zero, with $r(0) = 0$ such that $r(w)/w$ is non-increasing on $(0, \infty)$, then $g(\boldsymbol{x})$ is said to be a *sparseness measure*.

Considering the above definitions we obtain the following conclusions.

**Lemma 3.** (Gribonval & Nielsen, 2007, Proposition 1) Let $g : \mathbb{R}^n \to \mathbb{R}$ be a function satisfying Definition 5. Then, $g$ satisfies the triangle inequality.

Observe that the above lemma provides a practical methodology to verify when the function $g(\boldsymbol{x})$ satisfies the triangle inequality.

**Lemma 4.** Let $f : \mathbb{R}^n \to \mathbb{R}$ be an admissible function. Take $\boldsymbol{x} \in \mathbb{R}^n$ such that $\|\boldsymbol{x}\|_\infty < \infty$. If $f(\boldsymbol{x}) < \infty$, then $\|\boldsymbol{x}\|_2^2 < \frac{1}{c(\boldsymbol{x})} f(\boldsymbol{x})$ for a constant $c(\boldsymbol{x}) > 0$.

*Proof.* Since $f$ is an admissible function, then $f(\boldsymbol{x}) = \sum_{i=1}^n s(\boldsymbol{x}[i])$ and $s(w)/w^2$ is nonincreasing. Therefore, we have for all $i = 1, \ldots, n$

$$\frac{s(\boldsymbol{x}[i])}{\boldsymbol{x}^2[i]} \geq \frac{s(\|\boldsymbol{x}\|_\infty)}{\|\boldsymbol{x}\|_\infty^2} = c(\boldsymbol{x}) > 0. \tag{65}$$

Thus, we obtain

$$\|\boldsymbol{x}\|_2^2 \leq \frac{1}{c(\boldsymbol{x})} f(\boldsymbol{x}). \tag{66}$$

Thus the result holds. $\qquad \square$

With the above definitions and lemmata we proceed to prove Theorem 7. This proof assumes the regularizer $g(\boldsymbol{x})$ satisfies Definition 5. This assumption is proved at the end of this section.

*Proof.* From hypothesis we assume matrix $\boldsymbol{A}$ satisfies the RIP condition for any $k$-sparse vector with $\delta_{2k} < \frac{4}{\sqrt{41}}$. Then, taking $\mathcal{S} \subset \{1, \cdots, n\}$ the k-support of $\boldsymbol{x}$, from (Foucart & Rauhut, 2013, Theorem 6.13) and for constants $\rho \in (0,1), \tau > 0$ we have

$$\|\boldsymbol{x}_{\mathcal{S}}\|_1 \le \rho \|\boldsymbol{x}_{\mathcal{S}^c}\|_1 + \tau \|\boldsymbol{A}\boldsymbol{x}^* - \boldsymbol{y}\|_2, \tag{67}$$

where the notation means that $\boldsymbol{x}_{\mathcal{S}}$ coincides with $\boldsymbol{x}$ on the indices in $\mathcal{S}$ and is extended to zero in its complement $\mathcal{S}^c$. Observe that by adding the term $\rho \|\boldsymbol{x}_S\|_1$ to equation (67) we conclude

$$\|\boldsymbol{x}_{\mathcal{S}}\|_1 \le \frac{\rho}{1+\rho} \|\boldsymbol{x}\|_1 + \frac{\tau}{1+\rho} \|\boldsymbol{A}\boldsymbol{x}^* - \boldsymbol{y}\|_2. \tag{68}$$

Since we assume $g(\boldsymbol{x})$ satisfies Definition 5, from (Gribonval & Nielsen, 2007, Theorem 5) we know that $\frac{g(\boldsymbol{x}_{\mathcal{S}})}{g(\boldsymbol{x})} \le \frac{\|\boldsymbol{x}_{\mathcal{S}}\|_1}{\|\boldsymbol{x}\|_1}$. Thus, combining this inequality with equation (68) we get

$$g(\boldsymbol{x}_{\mathcal{S}}) \le \frac{\rho}{1+\rho} g(\boldsymbol{x}) + \frac{\tau'}{1+\rho} \|\boldsymbol{A}\boldsymbol{x}^* - \boldsymbol{y}\|_2$$
$$g(\boldsymbol{x}_{\mathcal{S}}) \le \rho g(\boldsymbol{x}_{\mathcal{S}^c}) + \tau' \|\boldsymbol{A}\boldsymbol{x}^* - \boldsymbol{y}\|_2, \tag{69}$$

where $\tau' = \tau \frac{g(\boldsymbol{x})}{\|\boldsymbol{x}\|_1} < \infty$. Observe that since $g(\boldsymbol{x})$ holds Lemma 3, then from (Woodworth & Chartrand, 2016, Proposition 4.4), and equation (69) we get

$$g(\boldsymbol{x}^* - \boldsymbol{x}) \le 2 \frac{1+\rho}{1-\rho} g(\boldsymbol{x}_{\mathcal{S}^c}) + 4 \frac{\tau'}{1-\rho} \|\boldsymbol{A}\boldsymbol{x}^* - \boldsymbol{y}\|_2. \tag{70}$$

Now, since $f(\boldsymbol{x})$ is an admissible function, from Lemma 4, and equation (70) we obtain

$$g(\boldsymbol{x}^* - \boldsymbol{x}) \le 2 \frac{1+\rho}{1-\rho} g(\boldsymbol{x}_{\mathcal{S}^c}) + 4 \frac{\tau'}{1-\rho} \sqrt{\epsilon} \sqrt{1/c(\boldsymbol{\eta})}, \tag{71}$$

where constant $c(\boldsymbol{\eta})$ as in equation (65). It is important to remark that the above inequality is only relevant for $g(\boldsymbol{x}) = \sum_{i=1}^{n} s_g(|\boldsymbol{x}[i]|)$ quasi-invex when $s_g(\|\boldsymbol{x}\|_\infty) \le \gamma$ (which is assumed in this proof) for $\gamma = \min\{t \in (0, \infty) | s_g'(w) = 0\}$, otherwise $s_g(w)$ does not reflect the behaviour of $\boldsymbol{x}$. This assumption also applies to $f(\boldsymbol{\eta})$ quasi-invex which implies the constrain $s_f(\|\boldsymbol{\eta}\|_\infty) \le \gamma$. In addition, for regularizers $g(\boldsymbol{x})$ satisfying Definition 5 we also know from (Gribonval & Nielsen, 2007, Lemma 1) that exists a constant $b(\boldsymbol{x})$ such that $\|\boldsymbol{x}\|_1 \le \frac{1}{b(\boldsymbol{x})} g(\boldsymbol{x})$. Then, from equation (71) we conclude

$$\|\boldsymbol{x}^* - \boldsymbol{x}\|_1 \le C \beta_{k,g}(\boldsymbol{x}) + D \sqrt{\epsilon} \upsilon_{f,\eta}(\boldsymbol{x}), \tag{72}$$

where $C = 2\frac{1+\rho}{1-\rho}$, $\beta_{k,g}(\boldsymbol{x}) = \frac{g(\boldsymbol{x}_{\mathcal{S}^c})}{b(\boldsymbol{x})}$, $D = 4\frac{\tau'}{1-\rho}$, and $\upsilon_{f,\eta}(\boldsymbol{x}) = \sqrt{1/c(\boldsymbol{\eta})}/b(\boldsymbol{x})$. Thus the result holds. $\qquad\square$

### SATISFYING DEFINITION 5

In this section we show that regularizers $g(\boldsymbol{x})$ in equation (3), equation (9), equation (10) satisfy Definition 5.

### EQUATION (3)

*Proof.* Take $r(w) = \log(1 + |w|^p)$ for any $w \ne 0$, and fixed $p \in (0,1)$. It is trivial to see that $r(0) = 0$, that $r(w)$ it is not identically zero, and non-decreasing on $(0, \infty)$. Then, we just need to show that $r(w)/w$ is non-increasing on $(0, \infty)$. Observe that the first derivative of $h(w) = r(w)/w$ is given by $h'(w) = \frac{1}{w^2} \left( \frac{pw^p}{1+w^p} - \log(1+w^p) \right)$. Since $\frac{pw^p}{1+w^p} - \log(1+w^p) < 0$, then we have that $h'(w) < 0$, which leads to conclude that $r(w)/w$ is non-increasing on $(0, \infty)$. Then it is clear that $r(w)/w$ is non-increasing on $(0, \infty)$. $\qquad\square$

EQUATION (9)

*Proof.* Take

$$r_\theta(w) = \begin{cases} \lambda|w| - w^2/(2\theta) & \text{if } |w| < \theta\lambda \\ \theta\lambda^2/2 & \text{if } |w| \geq \theta\lambda \end{cases},$$

for any $w \in \mathbb{R}$ and fixed $\lambda > 0$, $\theta \geq 1$. It is trivial to see that $r_\theta(0) = 0$, and that $r_\theta(w)$ it is not identically zero. Then, we just need to show that $h(w) = r_\theta(w)/w$ is non-increasing on $(0, \infty)$. Observe that the first derivative of $h(w) = r(w)/w$ is given by

$$h'(w) = \begin{cases} -1/(2\theta) & \text{if } |w| < \theta\lambda \\ -\theta\lambda^2/(2w^2) & \text{if } |w| \geq \theta\lambda \end{cases}.$$

Since $h'(w) < 0$, then $r(w)/w$ is non-increasing on $(0, \infty)$. Thus it is clear that $r(w)/w$ is non-increasing on $(0, \infty)$. $\qquad\square$

## H  PROOF OF THEOREM 8

*Proof.* Let $g : \mathbb{R}^n \to \mathbb{R}$ be an admissible function. Then from Lemma 4 we know that $g(\boldsymbol{x}) \geq c\|\boldsymbol{x}\|_2^2$ for some $c > 0$, and therefore $g_{TV}(\boldsymbol{x}) \geq c'\|\boldsymbol{x}\|_2^2$ for some $c' > 0$. Then from (Pallaschke & Rolewicz, 2013, Proposition 5.2.13) we know there exits constant $d' > 0$ such that program (11) has a unique minimizer for any $\lambda \leq \frac{1}{2d'}$. This fact implies that program (11) is invex. It is easy to verify that the TV-regularizer version of mappings in equation (4), equation (9), and equation (10) the constant $d'$ is equal to one.

Now for the second part of this theorem, we have that $h(\boldsymbol{x}) = g_{TV}(\boldsymbol{x}) + \frac{1}{2\lambda}\|\boldsymbol{x} - \boldsymbol{u}\|_2^2$ for some $\boldsymbol{u} \in \mathbb{R}^n$ is an invex function (this is also true for $h(\boldsymbol{x}) = g(\boldsymbol{x}) + \frac{1}{2\lambda}\|\boldsymbol{x} - \boldsymbol{u}\|_2^2$ with $\lambda \leq \frac{1}{2d}$), then Theorem 2 states that any global minimizer $\boldsymbol{y}$ of $h$ satisfies that $\boldsymbol{0} \in \partial h(\boldsymbol{y})$. This condition implies that $\boldsymbol{0} \in \partial g_{TV}(\boldsymbol{y}) + \frac{1}{\lambda}(\boldsymbol{y} - \boldsymbol{u})$, from which we obtain that $\boldsymbol{y} \in (\lambda \partial g_{TV} + \mathbf{I})^{-1}(\boldsymbol{u})$. Thus, we have that $\mathbf{prox}_{g_{TV}}(\boldsymbol{u}) = (\lambda \partial g_{TV} + \mathbf{I})^{-1}(\boldsymbol{u})$ (this is also true for $g(\boldsymbol{x})$) from which the result holds. $\quad\square$

### H.1  REMARKS ON PROX-REGULAR FUNCTIONS

In this section we prove a prox-regular function is quasi-invex. To that end, we introduce the following definition first.

**Definition 6.** Let $f : \mathbb{R}^n \to \mathbb{R}$ be a lower semi-continuous function, and $\boldsymbol{u} \in \mathbb{R}^n$. Then $f$ is said to be *prox-regular* if $f(\boldsymbol{x}) + \frac{1}{2\lambda}\|\boldsymbol{x} - \boldsymbol{u}\|_2^2$ is convex for some $\lambda > 0$.

Now we proceed with the proof.

*Proof.* Let $f : \mathbb{R}^n \to \mathbb{R}$ be a prox-regular function for some $\lambda > 0$. Then we know that for any $\boldsymbol{x}, \boldsymbol{y} \in \mathbb{R}^n$

$$f(\boldsymbol{x}) - f(\boldsymbol{y}) \geq \boldsymbol{\zeta}^T(\boldsymbol{x} - \boldsymbol{y}) - \frac{1}{2\lambda}\|\boldsymbol{x} - \boldsymbol{y}\|_2^2 \tag{73}$$

for all $\boldsymbol{\zeta} \in \partial f(\boldsymbol{y})$. Define function $\eta(\boldsymbol{x}, \boldsymbol{y})$ as

$$\eta(\boldsymbol{x}, \boldsymbol{y}) = \begin{cases} \boldsymbol{0} & \text{if } \boldsymbol{0} \in \partial f(\boldsymbol{y}) \\ \boldsymbol{x} - \boldsymbol{y} - \frac{\|\boldsymbol{x} - \boldsymbol{y}\|_2^2}{2\lambda\|\boldsymbol{\zeta}^*\|_2^2}\boldsymbol{\zeta}^* & \text{otherwise} \end{cases}, \tag{74}$$

where $\boldsymbol{\zeta}^*$ is an element in $\partial f(\boldsymbol{y})$ of minimum norm. Take $\boldsymbol{x}, \boldsymbol{y} \in \mathbb{R}^n$ and assume

$$f(\boldsymbol{x}) - f(\boldsymbol{y}) \leq 0. \tag{75}$$

Observe that if $\boldsymbol{0} \in \partial f(\boldsymbol{y})$ then we get $\boldsymbol{\zeta}^T\eta(\boldsymbol{x}, \boldsymbol{y}) = 0$, for all $\boldsymbol{\zeta} \in \partial f(\boldsymbol{y})$. Additionally, if $\boldsymbol{0} \notin \partial f(\boldsymbol{y})$, then from equation (73) we obtain

$$\begin{aligned} 0 &\geq \boldsymbol{\zeta}^T(\boldsymbol{x} - \boldsymbol{y}) - \frac{1}{2\lambda}\|\boldsymbol{x} - \boldsymbol{y}\|_2^2 \\ &\geq \boldsymbol{\zeta}^T\left(\boldsymbol{x} - \boldsymbol{y} - \frac{\|\boldsymbol{x} - \boldsymbol{y}\|_2^2}{2\lambda\|\boldsymbol{\zeta}^*\|_2^2}\boldsymbol{\zeta}^*\right) = \boldsymbol{\zeta}^T\eta(\boldsymbol{x}, \boldsymbol{y}), \end{aligned} \tag{76}$$

where the second inequality comes from the fact that $\boldsymbol{\zeta}^*$ is an element in $\partial f(\boldsymbol{y})$ of minimum norm i.e. $\frac{\boldsymbol{\zeta}^T \boldsymbol{\zeta}^*}{\|\boldsymbol{\zeta}^*\|_2^2} \geq 1$ for all $\boldsymbol{\zeta} \in \partial f(\boldsymbol{y})$ (Bazaraa & Shetty, 2012, Theorem 2.4.4). From the above inequality the result holds. $\qquad\square$

## I    PROOF OF THEOREM 9

Here we prove Theorem 9 following a similar strategy as presented in (Parikh & Boyd, 2014).

*Proof.* Since $(\boldsymbol{x}^*, \boldsymbol{z}^*, \boldsymbol{v}^*)$ is a saddle point for $\mathcal{L}_0$, we have

$$\mathcal{L}_0(\boldsymbol{x}^*, \boldsymbol{z}^*, \boldsymbol{v}^*) \leq \mathcal{L}_0(\boldsymbol{x}^{(t+1)}, \boldsymbol{z}^{(t+1)}, \boldsymbol{v}^*). \tag{77}$$

Using $\boldsymbol{S}\boldsymbol{x}^* + \boldsymbol{P}\boldsymbol{z}^* = \boldsymbol{y}$ the left hand side is $h^* = \inf\{h_1(\boldsymbol{x}) + h_2(\boldsymbol{z}) \mid \boldsymbol{S}\boldsymbol{x} + \boldsymbol{P}\boldsymbol{z} = \boldsymbol{y}\}$. With $h^{(t+1)} = h_1(\boldsymbol{x}^{(t+1)}) + h_2(\boldsymbol{z}^{(t+1)})$, this can be written as

$$h^* \leq h^{(t+1)} + (\boldsymbol{v}^*)^T \boldsymbol{q}^{(t+1)}, \tag{78}$$

for $\boldsymbol{q}^{(t+1)} = \boldsymbol{S}\boldsymbol{x}^{(t+1)} + \boldsymbol{P}\boldsymbol{z}^{(t+1)} - \boldsymbol{y}$. Now, by definition, $\boldsymbol{x}^{(t+1)}$ minimizes $\mathcal{L}_\rho(\boldsymbol{x}, \boldsymbol{z}^{(t)}, \boldsymbol{v}^{(t)})$. From Appendix H and the fact that $\rho\sigma_n(\boldsymbol{S}) \geq 1$, $\rho\sigma_p(\boldsymbol{P}) \geq 1$, we know that the (necessary and sufficient) optimality condition for $\mathcal{L}_\rho(\boldsymbol{x}, \boldsymbol{z}^{(t)}, \boldsymbol{v}^{(t)})$ is given by

$$\boldsymbol{0} \in \partial\mathcal{L}_\rho(\boldsymbol{x}^{(t+1)}, \boldsymbol{z}^{(t)}, \boldsymbol{z}^{(t)}) = \partial h_1(\boldsymbol{x}^{(t+1)}) + \boldsymbol{S}^T \boldsymbol{v}^{(t)} + \rho \boldsymbol{S}^T(\boldsymbol{S}\boldsymbol{x}^{(t+1)} + \boldsymbol{P}\boldsymbol{z}^{(t)} - \boldsymbol{y}). \tag{79}$$

Since $\boldsymbol{v}^{(t+1)} = \boldsymbol{v}^{(t)} + \rho\boldsymbol{q}^{(t+1)}$, we can plug in $\boldsymbol{v}^{(t)} = \boldsymbol{v}^{(t+1)} - \rho\boldsymbol{q}^{(t+1)}$ and rearrange to obtain

$$\boldsymbol{0} \in \partial h_1(\boldsymbol{x}^{(t+1)}) + \boldsymbol{S}^T(\boldsymbol{v}^{(t+1)} - \rho\boldsymbol{P}(\boldsymbol{z}^{(t+1)} - \boldsymbol{z}^{(t)})). \tag{80}$$

From Appendix H, this implies that $\boldsymbol{x}^{(t+1)}$ uniquely minimizes

$$h_1(\boldsymbol{x}) + (\boldsymbol{v}^{(t+1)} - \rho\boldsymbol{P}(\boldsymbol{z}^{(t+1)} - \boldsymbol{z}^{(t)}))^T \boldsymbol{S}\boldsymbol{x}. \tag{81}$$

A similar argument shows that $\boldsymbol{z}^{(t+1)}$ uniquely minimizes $h_2(\boldsymbol{z}) + (\boldsymbol{v}^{(t+1)})^T \boldsymbol{P}\boldsymbol{z}$. It follows that

$$\begin{aligned} h_1(\boldsymbol{x}^{(t+1)}) + (\boldsymbol{v}^{(t+1)} - \rho\boldsymbol{P}(\boldsymbol{z}^{(t+1)} - \boldsymbol{z}^{(t)}))^T \boldsymbol{S}\boldsymbol{x}^{(t+1)} \\ \leq h_1(\boldsymbol{x}^*) + (\boldsymbol{v}^{(t+1)} - \rho\boldsymbol{P}(\boldsymbol{z}^{(t+1)} - \boldsymbol{z}^{(t)}))^T \boldsymbol{S}\boldsymbol{x}^*, \end{aligned} \tag{82}$$

and that

$$h_2(\boldsymbol{z}^{(t+1)}) + (\boldsymbol{v}^{(t+1)})^T \boldsymbol{P}\boldsymbol{z}^{(t+1)} \leq h_2(\boldsymbol{z}^*) + (\boldsymbol{v}^{(t+1)})^T \boldsymbol{P}\boldsymbol{z}^*. \tag{83}$$

Adding the two inequalities above, using $\boldsymbol{S}\boldsymbol{x}^* + \boldsymbol{P}\boldsymbol{z}^* = \boldsymbol{y}$, and rearranging, we obtain

$$h^{(t+1)} - h^* \leq -(\boldsymbol{v}^{(t+1)})^T \boldsymbol{q}^{(t+1)} - \rho(\boldsymbol{P}(\boldsymbol{z}^{(t+1)} - \boldsymbol{z}^{(t)}))^T(-\boldsymbol{q}^{(t+1)} + \boldsymbol{P}(\boldsymbol{z}^{(t+1)} - \boldsymbol{z}^*)). \tag{84}$$

On the other hand, adding equation (78), and equation (84), regrouping terms, and multiplying through by 2 gives

$$\begin{aligned} 2(\boldsymbol{v}^{(t+1)} - \boldsymbol{v}^*)^T \boldsymbol{q}^{(t+1)} - 2\rho(\boldsymbol{P}(\boldsymbol{z}^{(t+1)} - \boldsymbol{z}^{(t)}))^T \boldsymbol{q}^{(t+1)} \\ + 2\rho(\boldsymbol{P}(\boldsymbol{z}^{(t+1)} - \boldsymbol{z}^{(t)}))^T(\boldsymbol{P}(\boldsymbol{z}^{(t+1)} - \boldsymbol{z}^*)) \leq 0. \end{aligned} \tag{85}$$

Now by rewriting the first term in equation (85), and substituting $\boldsymbol{v}^{(t+1)} = \boldsymbol{v}^{(t)} + \rho\boldsymbol{q}^{(t+1)}$ it gives

$$2(\boldsymbol{v}^{(t+1)} - \boldsymbol{v}^*)^T \boldsymbol{q}^{(t+1)} + \rho\|\boldsymbol{q}^{(t+1)}\|_2^2 + \rho\|\boldsymbol{q}^{(t+1)}\|_2^2, \tag{86}$$

and substituting $\boldsymbol{q}^{(t+1)} = (1/\rho)(\boldsymbol{v}^{(t+1)} - \boldsymbol{v}^{(t)})$ in the first two terms gives

$$(2/\rho)(\boldsymbol{v}^{(t)} - \boldsymbol{v}^*)^T(\boldsymbol{v}^{(t+1)} - \boldsymbol{v}^{(t)}) + (1/\rho)\|\boldsymbol{v}^{(t+1)} - \boldsymbol{v}^{(t)}\|_2^2 + \rho\|\boldsymbol{q}^{(t+1)}\|_2^2. \tag{87}$$

Since $\boldsymbol{q}^{(t+1)} - \boldsymbol{q}^{(t)} = (\boldsymbol{q}^{(t+1)} - \boldsymbol{q}^*) - (\boldsymbol{q}^{(t)} - \boldsymbol{q}^*)$, this can be written as

$$(1/\rho)(\|\boldsymbol{v}^{(t+1)} - \boldsymbol{v}^*\|_2^2 - \|\boldsymbol{v}^{(t)} - \boldsymbol{v}^*\|_2^2) + \rho\|\boldsymbol{q}^{(t+1)}\|_2^2. \tag{88}$$

We now rewrite the remaining terms

$$\rho\|\boldsymbol{q}^{(t+1)}\|_2^2 - 2\rho(\boldsymbol{P}(\boldsymbol{z}^{(t+1)} - \boldsymbol{z}^{(t)}))^T\boldsymbol{q}^{(t+1)} + 2\rho(\boldsymbol{P}(\boldsymbol{z}^{(t+1)} - \boldsymbol{z}^{(t)}))^T(\boldsymbol{P}(\boldsymbol{z}^{(t+1)} - \boldsymbol{z}^*)), \quad (89)$$

where $\rho\|\boldsymbol{q}^{(t+1)}\|_2^2$ is taken from equation (88). Substituting

$$\boldsymbol{z}^{(t+1)} - \boldsymbol{z}^* = (\boldsymbol{z}^{(t+1)} - \boldsymbol{z}^{(t)}) + (\boldsymbol{z}^{(t)} - \boldsymbol{z}^*), \quad (90)$$

in the last term gives

$$\rho\|\boldsymbol{q}^{(t+1)} - \boldsymbol{P}(\boldsymbol{z}^{(t+1)} - \boldsymbol{z}^{(t)})\|_2^2 + \rho\|\boldsymbol{P}(\boldsymbol{z}^{(t+1)} - \boldsymbol{z}^{(t)})\|_2^2$$
$$+ 2\rho(\boldsymbol{P}(\boldsymbol{z}^{(t+1)} - \boldsymbol{z}^{(t)}))^T(\boldsymbol{P}(\boldsymbol{z}^{(t+1)} - \boldsymbol{z}^*)), \quad (91)$$

and substituting

$$\boldsymbol{z}^{(t+1)} - \boldsymbol{z}^{(t)} = (\boldsymbol{z}^{(t+1)} - \boldsymbol{z}^*) - (\boldsymbol{z}^{(t)} - \boldsymbol{z}^*), \quad (92)$$

in the last two terms, we get

$$\rho\|\boldsymbol{q}^{(t+1)} - \boldsymbol{P}(\boldsymbol{z}^{(t+1)} - \boldsymbol{z}^{(t)})\|_2^2 + \rho\left(\|\boldsymbol{P}(\boldsymbol{z}^{(t+1)} - \boldsymbol{z}^*)\|_2^2 - \|\boldsymbol{P}(\boldsymbol{z}^{(t)} - \boldsymbol{z}^*)\|_2^2\right). \quad (93)$$

With the previous step, this implies that equation (85) can be written as

$$V^{(t)} - V^{(t+1)} \geq \rho\|\boldsymbol{q}^{(t+1)} - \boldsymbol{P}(\boldsymbol{z}^{(t+1)} - \boldsymbol{z}^{(t)})\|_2^2, \quad (94)$$

where $V^{(t)} = (1/\rho)\|\boldsymbol{v}^{(t)} - \boldsymbol{v}^*\|_2^2 + \rho\|\boldsymbol{P}(\boldsymbol{z}^{(t)} - \boldsymbol{z}^*)\|_2^2$.

Now, we show that the middle term $-2\rho(\boldsymbol{q}^{(t+1)})^T(\boldsymbol{P}(\boldsymbol{z}^{(t+1)} - \boldsymbol{z}^{(t)}))$ of the expanded right hand side of equation (94) is positive. To see this, recall that $\boldsymbol{z}^{(t+1)}$ minimizes $h_2(\boldsymbol{z}) + (\boldsymbol{v}^{(t+1)})^T\boldsymbol{P}\boldsymbol{z}$, and $\boldsymbol{z}^{(t)}$ minimizes $h_2(\boldsymbol{z}) + (\boldsymbol{v}^{(t)})^T\boldsymbol{P}\boldsymbol{z}$, so we can add

$$h_2(\boldsymbol{z}^{(t+1)}) + (\boldsymbol{v}^{(t+1)})^T\boldsymbol{P}\boldsymbol{z}^{(t+1)} \leq h_2(\boldsymbol{z}^{(t)}) + (\boldsymbol{v}^{(t+1)})^T\boldsymbol{P}\boldsymbol{z}^{(t)}, \quad (95)$$

and

$$h_2(\boldsymbol{z}^{(t)}) + (\boldsymbol{v}^{(t)})^T\boldsymbol{P}\boldsymbol{z}^{(t)} \leq h_2(\boldsymbol{z}^{(t+1)}) + (\boldsymbol{v}^{(t)})^T\boldsymbol{P}\boldsymbol{z}^{(t+1)}, \quad (96)$$

to get that

$$(\boldsymbol{v}^{(t+1)} - \boldsymbol{v}^{(t)})^T\boldsymbol{P}(\boldsymbol{z}^{(t+1)} - \boldsymbol{z}^{(t)}) \leq 0. \quad (97)$$

Substituting $\boldsymbol{v}^{(t+1)} - \boldsymbol{v}^{(t)} = \rho\boldsymbol{q}^{(t+1)}$ gives the result, since $\rho > 0$. Thus, from equation (94), and equation (97) we obtain

$$V^{(t+1)} \leq V^{(t)} - \rho\|\boldsymbol{q}^{(t+1)}\|_2^2 - \rho\|\boldsymbol{P}(\boldsymbol{z}^{(t+1)} - \boldsymbol{z}^{(t)})\|_2^2, \quad (98)$$

which states that $V^{(t)}$ decreases in each iteration by an amount that depends on the norm of the residual $\boldsymbol{q}^{(t)}$ and on the change in $\boldsymbol{z}^{(t)}$ over one iteration. Then, because $V^{(t)} \leq V^{(0)}$, it follows that $\boldsymbol{v}^{(t)}$ and $\boldsymbol{P}\boldsymbol{z}^{(t)}$ are bounded. Iterating the inequality above gives that

$$\rho\sum_{t=0}^{\infty}\left(\|\boldsymbol{q}^{(t+1)}\|_2^2 + \|\boldsymbol{P}(\boldsymbol{z}^{(t+1)} - \boldsymbol{z}^{(t)})\|_2^2\right) \leq V^{(0)}, \quad (99)$$

which implies that $\boldsymbol{q}^{(t)} = \boldsymbol{S}\boldsymbol{x}^{(t)} + \boldsymbol{P}\boldsymbol{z}^{(t)} - \boldsymbol{y} \rightarrow 0$, and $\boldsymbol{P}(\boldsymbol{z}^{(t+1)} - \boldsymbol{z}^{(t)}) \rightarrow 0$ as $t \rightarrow \infty$. Additionally, applying (Deng et al., 2017, Lemma 1.2) on equation (99) we obtain a convergence rate for $\boldsymbol{q}^{(t)}, \boldsymbol{P}(\boldsymbol{z}^{(t+1)} - \boldsymbol{z}^{(t)})$ to zero of $\mathcal{O}(1/t)$. equation (99) also implies that the right hand side in equation (84) goes to zero as $t \rightarrow \infty$, because $\boldsymbol{P}(\boldsymbol{z}^{(t+1)} - \boldsymbol{z}^*)$ is bounded and both $\boldsymbol{q}^{(t+1)}$ and $\boldsymbol{P}(\boldsymbol{z}^{(t+1)} - \boldsymbol{z}^{(t)})$ go to zero. The right hand side in equation (78) goes to zero as $t \rightarrow \infty$, since $\boldsymbol{q}^{(t)}$ goes to zero. Thus we have $\lim_{t\rightarrow\infty} h^{(t)} = h^*$, i.e., objective convergence. Therefore the result of Theorem 9 holds. $\qquad\square$

## I.1 Remarks on Stability of ADMM

In this section we wish to emphasize that the stability and reliability (of the ADMM) can be established from the following: 1) ADMM decomposes the overall optimization problem into a number of simpler subproblems that have a unique solution (as rigorously demonstrated in previous section), aiding convergence and stability. 2) As shown in previous section, the sequences $\boldsymbol{x}^{(t+1)}$, $\boldsymbol{z}^{(t+1)}$, and $\boldsymbol{v}^{(t+1)}$, constructed by ADMM algorithm, always converge to global optima irrespective of the initial states $\boldsymbol{x}^{(0)}, \boldsymbol{z}^{(0)}$, and $\boldsymbol{v}^{(0)}$, conferring a steadfast assurance of reliable attainment of optimal solutions, and finally, 3) The effectiveness of the ADMM is well demonstrated in the literature using a range of real-world applications (Boyd et al., 2011; Wang et al., 2019b; Glowinski, 2014; Xie et al., 2019), which we believe can reaffirm the reliability (and potentially the stability) of ADMM.

### Remarks on optimization problem in 1

In this section we show how program 1 is expressed as equation (12), where the non-linear constrain is in the form of $f(\boldsymbol{Ax} - \boldsymbol{y})$. From (Chen et al., 2001; Cai & Wang, 2011), we know that

$$\text{minimize } g(\boldsymbol{x}) \text{ subject to } f(\boldsymbol{Ax} - \boldsymbol{y}) < \epsilon,$$

can be equivalently rewritten as

$$\text{minimize } g(\boldsymbol{x}) + \lambda f(\boldsymbol{Ax} - \boldsymbol{y}),$$

for some $\lambda > 0$ that helps to satisfy the condition $f(\boldsymbol{Ax} - \boldsymbol{y}) < \epsilon$. Thus, if introducing a variable $\boldsymbol{z} = \boldsymbol{Ax} - \boldsymbol{y}$, program (1) can be finally rewritten as

$$\text{minimize } g(\boldsymbol{x}) + \lambda f(\boldsymbol{z}) \text{ subject to } \boldsymbol{z} = \boldsymbol{Ax} - \boldsymbol{y},$$

which is in the optimization form of ADMM as pointed out in equation (12). In this way, optimization problem in (1) is converted from non-linear constraints into linear ones.

## I.2 Additional Discussion on Theorem 9

In this section, we discuss why the assumptions $\rho\sigma_n(\boldsymbol{S}) \geq 1$, and $\rho\sigma_p(\boldsymbol{P}) \geq 1$ for $\rho > 0$ in Theorem 9 are mild, by showing practical imaging examples that satisfy these conditions. We list those applications in the following.

1. **Image Fusion:** Image fusion has been receiving increasing attention in the research community to investigate general formal solutions to a broad spectrum of applications (Vargas et al., 2019). For example, in the remote sensing field, the increasing availability of spaceborne imaging sensors operating in various ground scales and spectral bands undoubtedly provides strong motivations (Camacho et al., 2022). Because of the trade-off imposed by the physical constraint between spatial and spectral resolutions, spatial enhancement of poor-resolution multispectral (MS) data is desirable (Camacho et al., 2022). Another example is medical imaging, in which the goal of image fusion is to create new images more suitable for human visual perception (Meyer-Bäse et al., 2004).

Mathematically the image fusion problem in spectral imaging can be modeled as in equation (12) defining matrices $\boldsymbol{S}$, and $\boldsymbol{P}$ as (Vargas et al., 2019)

$$\boldsymbol{S} = \begin{bmatrix} \boldsymbol{0} \\ \boldsymbol{I} \\ \boldsymbol{0} \\ \boldsymbol{B}_s \\ \boldsymbol{G} \\ \boldsymbol{D} \\ \boldsymbol{I} \end{bmatrix}, \boldsymbol{P} = \begin{bmatrix} \boldsymbol{I} & -\boldsymbol{R}_\lambda\overline{\boldsymbol{M}} & \boldsymbol{0} & \boldsymbol{0} & \boldsymbol{0} & \boldsymbol{0} & \boldsymbol{0} \\ \boldsymbol{0} & \boldsymbol{I} & \boldsymbol{0} & \boldsymbol{0} & \boldsymbol{0} & \boldsymbol{0} & \boldsymbol{0} \\ \boldsymbol{0} & \boldsymbol{0} & \boldsymbol{I} & -\overline{\boldsymbol{M}} & \boldsymbol{0} & \boldsymbol{0} & \boldsymbol{0} \\ \boldsymbol{0} & \boldsymbol{0} & \boldsymbol{0} & \boldsymbol{I} & \boldsymbol{0} & \boldsymbol{0} & \boldsymbol{0} \\ \boldsymbol{0} & \boldsymbol{0} & \boldsymbol{0} & \boldsymbol{0} & \boldsymbol{I} & \boldsymbol{0} & \boldsymbol{0} \\ \boldsymbol{0} & \boldsymbol{0} & \boldsymbol{0} & \boldsymbol{0} & \boldsymbol{0} & \boldsymbol{I} & \boldsymbol{0} \\ \boldsymbol{0} & \boldsymbol{0} & \boldsymbol{0} & \boldsymbol{0} & \boldsymbol{0} & \boldsymbol{0} & \boldsymbol{I} \end{bmatrix} = \boldsymbol{I} + \boldsymbol{H}. \quad (100)$$

Observe that $\sigma_n(\boldsymbol{S}) = 2 + \delta > 1$ for $\delta > 0$ where is the smallest singular value of $\boldsymbol{B}_s^T\boldsymbol{B}_s + \boldsymbol{G}^T\boldsymbol{G} + \boldsymbol{D}^T\boldsymbol{D}$. In addition, $\sigma_n(\boldsymbol{P}) = 1 + \zeta > 0$ where $\zeta$ is the smallest singular value of $\boldsymbol{H} + \boldsymbol{H}^T + \boldsymbol{H}^T\boldsymbol{H}$. Thus, taking $\rho \geq \max\{0.5, 1/(1 + \zeta)\}$ the needed conditions in Theorem 9 are satisfied.

2. **Spectral Imaging:** Spectral imaging combines two disciplines - spectroscopy and photography - to sample image data at many wavelength bands. In general, spectral imaging is separated into either

multispectral (¡ 20 wavelength bands sampled) or hyperspectral (¿ 20 wavelength bands) (Vargas et al., 2018a). Mathematically the spectral imaging problem can be modeled as in equation (12) defining matrices $\boldsymbol{S}$, and $\boldsymbol{P}$ are given by (Vargas et al., 2018a)

$$\boldsymbol{S} = \left[ \underbrace{\boldsymbol{I}, \dots, \boldsymbol{I}}_{K}, \boldsymbol{\Psi}, \boldsymbol{L}^T \right]^T, \boldsymbol{P} = -\boldsymbol{I}, \tag{101}$$

where $\boldsymbol{\Psi}$ is an orthogonal matrix. Observe that $\sigma_n(\boldsymbol{S}) = K + 1 + \delta > 1$ for $\delta > 0$ where is the smallest singular value of $\boldsymbol{L}^T \boldsymbol{L}$. In addition, $\sigma_n(\boldsymbol{P}) = 1$. Thus, taking $\rho \geq 1$ the needed conditions in Theorem 9 are satisfied.

3. **Computer Tomography:** Computer tomography refers to a computerized X-ray imaging procedure in which a narrow beam of X-rays is aimed at a patient and quickly rotates around the body, producing signals processed by the machine's computer to generate cross-sectional images (Wang et al., 2019a). These slices are called tomographic images and can give a clinician more detailed information than conventional X-rays. Mathematically, the computer tomography problem can be modeled as in equation (12) defining matrices $\boldsymbol{S}$, and $\boldsymbol{P}$ are given by $\boldsymbol{S} = \boldsymbol{I}$, and $\boldsymbol{P} = -\boldsymbol{I}$ (Vargas et al., 2018a). Thus, taking $\rho \geq 1$ the needed conditions in Theorem 9 are satisfied.

4. **Magnetic Resonance Imaging:** Magnetic Resonance Imaging (MRI) is a non-invasive imaging technique providing both functional and anatomical information for clinical diagnosis. Imaging speed is a fundamental challenge. Fast MRI techniques are essentially demanded to accelerate data acquisition while still reconstructing a high-quality image (Sun et al., 2016). Mathematically, the magnetic resonance imaging problem can be modeled as in equation (12) defining matrices $\boldsymbol{S}$, and $\boldsymbol{P}$ are given by $\boldsymbol{S} = \boldsymbol{I}$, and $\boldsymbol{P} = -\boldsymbol{\Psi}$ (Vargas et al., 2018a), for $\boldsymbol{\Psi}$ an orthogonal matrix. Thus, taking $\rho \geq 1$ the needed conditions in Theorem 9 are satisfied.

5. **Compressive Sensing:** Compressive sensing is a recent highly applicative approach. It enables efficient data sampling at a much lower rate than the requirements indicated by the Nyquist theorem. Compressive sensing possesses several advantages, such as the much smaller need for sensory devices, much less memory storage, higher data transmission rate, many times less power consumption (Ramirez et al., 2021). Due to all these advantages, compressive sensing has been used in a wide range of applications. Mathematically, the compressive sensing problem can be modeled as in equation (12) defining matrices $\boldsymbol{S}$, and $\boldsymbol{P}$ are given by $\boldsymbol{S} = \boldsymbol{\Psi}$, and $\boldsymbol{P} = -\boldsymbol{I}$ (Ramirez et al., 2021), for $\boldsymbol{\Psi}$ an orthogonal matrix. Thus, taking $\rho \geq 1$ the needed conditions in Theorem 9 are satisfied.

6. **Stepped-Frequency Radar:** Step-frequency is a radar waveform consisting of a series of sine waves with linearly increasing frequency. The radar measures the phase and amplitude on each frequency and used an inverse Fourier transform of these data to build a time domain profile (Johnston et al., 2021). Mathematically, the Stepped-Frequency Radar problem can be modeled as in equation (12) defining matrices $\boldsymbol{S}$, and $\boldsymbol{P}$ are given by $\boldsymbol{S} = \boldsymbol{I}$, and $\boldsymbol{P} = -\boldsymbol{I}$ (Johnston et al., 2021). Thus, taking $\rho \geq 1$ the needed conditions in Theorem 9 are satisfied.

### I.3 PROXIMAL SOLUTIONS

In this section we present the proximal operator for the studied invex/quasi-invex functions in this work, summarized in the following Table 4.

PROXIMAL OF FUNCTION IN EQUATION (3)

Consider $h(w) = \lambda \log(1 + |w|^p) + \frac{1}{2}(w - u)^2$ for $\lambda \in (0, 1]$, $w \neq 0$, and fixed $u \in \mathbb{R}$, $p \in (0, 1)$. We note first that we only consider $w's$ for which $\text{sign}(w) = \text{sign}(u)$, otherwise $h(w) = \lambda \log(1 + |w|^p) + \frac{1}{2}w^2 + |u||w| + \frac{1}{2}u^2$ which is clearly minimized at $w = 0$. Then, since with $\text{sign}(w) = \text{sign}(u)$ we have $(w - u)^2 = (|w| - |u|)^2$, we replace $u$ with $|u|$ and take $w \geq 0$. As $h(w)$ is differentiable for $w > 0$, re-arranging $h'(w) = 0$ gives $\psi_\lambda(w) \triangleq \frac{\lambda p w^{p-1}}{1 + w^p} + w = |u|$. Observe that $\psi'_\lambda(w)$ is always positive then it means that $\psi_\lambda(w)$ is monotonically increasing. Thus, the equation $\psi_\lambda(w) = |u|$ has unique solution i.e. at some point the quality holds. Thus, solving $\psi_\lambda(w) = |u|$ is equivalent to

$$w(w - |u|)(w^{-p} + 1) + \lambda p = 0. \tag{102}$$

We solve equation (102) using standard bisection method (Burden & Faires, 1985).

Table 4: Proximal operator for regularizers $\ell_p$-quasinorm, and equation (3), equation (9) ($\lambda \in (0, 1]$ is a thresholding parameter).

| Ref | Invex function | Proximal operator |
|---|---|---|
| (Marjanovic & Solo, 2012) | $g_\lambda(w) = \lambda|w|^p, p \in (0,1), w \neq 0.$ | $\mathrm{Prox}_{g_\lambda}(w) = \begin{cases} 0 & |t| < \tau \\ \{0, \mathrm{sign}(w)\beta\} & |t| = \tau \\ \mathrm{sign}(w)y & |t| > \tau \end{cases}$ 

 where $\beta = [2\lambda(1-p)]^{1/(2-p)}$, $\tau = \beta + \lambda p \beta^{p-1}$, $h(y) = \lambda p y^{p-1} + y - |t| = 0, y \in [\beta, |t|]$ |
| - | $g_\lambda(w) = \lambda\log(1 + |w|^p), p \in (0,1), x \neq 0$ | $\mathrm{Prox}_{g_\lambda}(w) = \begin{cases} 0 & |w| = 0 \\ \mathrm{sign}(w)\beta & \text{otherwise} \end{cases}$ 

 where $\beta(\beta - |w|)(\beta^{-p} + 1) + \lambda p = 0, \beta > 0$. We solve this equation using standard bisection method. |
| (Zhang, 2010) | MCP | $\mathrm{Prox}_{\lambda g_\theta}(w) = \begin{cases} \frac{\theta w - \mathrm{sign}(w)\theta\lambda}{\theta - 1} & |w| < \theta\lambda \\ w & |w| \geq \theta\lambda \end{cases}$ |

**Implementation details:** In order to efficiently compute the proximal solutions, which is an entry-wise operation, in Table 4, we use the Python library CuPy (Lib). The reason is that this library allows the creation of GPU kernels in the language C++ from Python (quick guide on how to create and use these kernels (ker)). Therefore, inside this GPU kernel, we implemented the entry-wise operation from the Proximal Operator column of Table 4. We report as an example how to implement the proximal solution for $\ell_p$-quasinorm. We remark that this implementation is efficient because it runs in parallel and at the GPU speed. Additional features of the CuPy library are the flexibility to connect both TensorFlow and PyTorch libraries for training neural networks (pyt).

```python
invex2D_kernel = cp.RawKernel(r'''
            extern "C"
            __global__ void invex2DFilter(const float *x, float *y, float lamb, float q, int size){

                int tid = blockDim.x * blockIdx.x + threadIdx.x;

                if (tid < size){
                    float beta = powf(2.0*lamb*(1.0-q),1.0/(2.0-q));
                    float tau  = beta + lamb*q*powf(beta,q-1.0);
                    float sign = -2.0*signbit(x[tid]) + 1.0;

                    y[tid] = 0.0f;

                    if(fabs(x[tid]) > tau){
                        float z = beta + (fabs(x[tid])-beta)/2.0;

                        for(int k=0;k < 4;k++){
                            z = fabs(x[tid]) - lamb*q*powf(z,q-1.0);
                        }

                        y[tid] = sign * z;

                    }
                }
            }
''', 'invex2DFilter')
```

Lastly, the running time to compute the proximal of invex/quasi-invex functions is also essential to compare with its convex competitor, i.e., $\ell_1$-norm. The reason is the desire to improve imaging quality, keeping the same computational complexity to obtain it. Therefore, the following Table 5 reports the running time to compute the proximal (in GPU) of equation (3), $\ell_p$-quasinorm, and equation (9) for an image of $2048 \times 2048$ pixels. Table 5 suggests that computing the proximal of the $\ell_1$-norm is faster than the proximal of invex regularizers. However, this difference is given in milliseconds, making it negligible in practice.

Table 5: Time to compute the proximal for all invex and convex regularizers, of an image with $2048 \times 2048$ pixels. The reported time is the averaged over 256 trials. For $\ell_p$-quasinorm we select $p = 0.5$, and $\epsilon = (p(1-p))^{\frac{1}{2-p}}$.

|      | equation (3) | $\ell_p$-quasinorm | equation (9) | $\ell_1$-norm |
|------|--------------|--------------------|--------------|---------------|
| Time | $2.8ms$      | $1.06ms$           | $0.86ms$     | $0.66ms$      |

## J  EXTENDED ACCELERATED PROXIMAL GRADIENT METHOD (APGM)

The accelerated proximal gradient method (APGM) (Li & Lin, 2015) has been shown to be effective in solving program (1) by minimizing $F(\boldsymbol{x}) = g(\boldsymbol{x}) + \hat{f}(\boldsymbol{x})$ where $\hat{f}(\boldsymbol{x}) = \frac{1}{2}\left(\max\{f(\boldsymbol{x}) - \epsilon, 0\}\right)^2$, achieving better quality in less iterations than its predecessors (Beck & Teboulle, 2009; Frankel et al., 2015; Boţ et al., 2016), and been frequently used by recent imaging works (Wang & Chen, 2022; Mai et al., 2022). Note that $\hat{f}$ is invex with the same $\eta$ as $f$ (which is also the same of $g$ according to Theorem 3), since $h(w) = (\max(w - \epsilon, 0))^2$ is an increasing convex function (Mishra & Giorgi, 2008, Theorem 2.14). Therefore, $F(\boldsymbol{x})$ is invex. The iterative process performed by APGM is summarized as

$$
\begin{aligned}
\boldsymbol{y}^{(t)} &= \boldsymbol{x}^{(t)} + \frac{r_{t-1}}{r_t}(\boldsymbol{z}^{(t)} - \boldsymbol{x}^{(t)}) + \frac{r_{t-1} - 1}{r_t}(\boldsymbol{x}^{(t)} - \boldsymbol{x}^{(t-1)}) \\
\boldsymbol{z}^{(t+1)} &= \mathrm{prox}_{\alpha_2 g}(\boldsymbol{y}^{(t)} - \alpha_2 \nabla \hat{f}(\boldsymbol{y}^{(t)})) \\
\boldsymbol{v}^{(t+1)} &= \mathrm{prox}_{\alpha_1 g}(\boldsymbol{x}^{(t)} - \alpha_1 \nabla \hat{f}(\boldsymbol{x}^{(t)})), r_{t+1} = \frac{\sqrt{4(r_t)^2 + 1} + 1}{2} \\
\boldsymbol{x}^{(t+1)} &= \begin{cases} \boldsymbol{z}^{(t+1)}, & \text{if } \hat{f}(\boldsymbol{z}^{(t+1)}) + g(\boldsymbol{z}^{(t+1)}) \le \hat{f}(\boldsymbol{v}^{(t+1)}) + g(\boldsymbol{v}^{(t+1)}) \\ \boldsymbol{v}^{(t+1)}, & \text{otherwise} \end{cases}
\end{aligned}
\tag{103}
$$

for some positive constants $\alpha_1, \alpha_2$, and assuming $\hat{f}$ is $L$-smooth. The reported convergence guarantees of $\boldsymbol{x}^{(t+1)}$ to global optima of APGM has been only stated for convex losses (Li & Lin, 2015). For non-convex cases, convergence to a critical point has been stated (Li & Lin, 2015). Here, we extend the general convergence guarantees of APGM to invex/quasi-invex settings in Theorem 3 so that its benefits are available to the signal restoration problems.

**Theorem 11.** Let $f, g : \mathbb{R}^n \to \mathbb{R}$ be admissible functions as in Definition 4, where $f$ is $L$-smooth. Assume $\boldsymbol{x}^\star$ a global minimizer of $F(\boldsymbol{x}) = g(\boldsymbol{x}) + \hat{f}(\boldsymbol{x})$ for $\epsilon = 0$, then

$$
F(\boldsymbol{x}^{(T+1)}) - F(\boldsymbol{x}^*) \le \frac{2}{\alpha'(T+1)^2}\|\boldsymbol{x}^{(0)} - \boldsymbol{x}^*\|_2^2,
\tag{104}
$$

where $T$ is number of iterations, $\alpha_1, \alpha_2 < \frac{\kappa}{L\kappa+4}$, $\frac{1}{\alpha'} = \left(\frac{1}{\alpha_1} - \frac{2}{\kappa}\right) > 0$, with $\kappa$ a constant that depends on $f, g$.

Proof of Theorem 11 relies on the properties of admissible functions including Theorem 8. The proof is presented below. It is worth mentioning that the convergence rate for admissible functions as in Theorem 11 is the same as in the convex case (see (Li & Lin, 2015)).

*Proof.* Step 2 in the APGM is given by

$$
\boldsymbol{z}^{(t+1)} = \underset{\boldsymbol{x} \in \mathbb{R}^n}{\arg\min} \ \left\langle \nabla f(\boldsymbol{y}^{(t)}), \boldsymbol{x} - \boldsymbol{y}^{(t)} \right\rangle + \frac{1}{2\alpha_1}\|\boldsymbol{x} - \boldsymbol{y}^{(t)}\|_2^2 + g(\boldsymbol{x}).
\tag{105}
$$

From Theorem 8 we know that equation (105) is unique for a $\alpha_1 < \frac{1}{\kappa_g}$ for some $\kappa_g > 0$. Then the optimality condition for equation (105) is given by

$$
\boldsymbol{0} \in \nabla f(\boldsymbol{y}^{(t)}) + \frac{1}{\alpha_1}(\boldsymbol{z}^{(t+1)} - \boldsymbol{y}^{(t)}) + \partial g(\boldsymbol{z}^{(t+1)}).
\tag{106}
$$

Additionally, since $g(\boldsymbol{x})$ is an admissible function, then from (Pallaschke & Rolewicz, 2013, Proposition 5.2.14) we have that

$$
g(\boldsymbol{x}) - g(\boldsymbol{z}^{(t+1)}) \ge \left\langle -\nabla f(\boldsymbol{y}^{(t)}) - \frac{1}{\alpha_1}(\boldsymbol{z}^{(t+1)} - \boldsymbol{y}^{(t)}), \boldsymbol{x} - \boldsymbol{z}^{(t+1)} \right\rangle - \frac{1}{\kappa_g}\|\boldsymbol{x} - \boldsymbol{z}^{(t+1)}\|_2^2.
\tag{107}
$$

for all $\boldsymbol{x}$. From the Lipschitz continuous of $\nabla f$, we have for all $\boldsymbol{x}$ and $\kappa_f < \frac{1}{L}$ that

$$f(\boldsymbol{x}) - f(\boldsymbol{y}^{(t)}) \geq \left\langle \nabla f(\boldsymbol{y}^{(t)}), \boldsymbol{x} - \boldsymbol{y}^{(t)} \right\rangle - \frac{1}{\kappa_f} \|\boldsymbol{x} - \boldsymbol{y}^{(t)}\|_2^2. \tag{108}$$

In addition of the Lipschitz continuity of $\nabla f$ we have that

$$F(\boldsymbol{z}^{(t+1)}) \leq g(\boldsymbol{z}^{(t+1)}) + f(\boldsymbol{y}^{(t)}) + \left\langle \nabla f(\boldsymbol{y}^{(t)}), \boldsymbol{z}^{(t+1)} - \boldsymbol{y}^{(t)} \right\rangle + \frac{L}{2} \|\boldsymbol{z}^{(t+1)} - \boldsymbol{y}^{(t)}\|_2^2$$

$$= g(\boldsymbol{z}^{(t+1)}) + f(\boldsymbol{y}^{(t)}) + \left\langle \nabla f(\boldsymbol{y}^{(t)}), \boldsymbol{x} - \boldsymbol{y}^{(t)} \right\rangle + \left\langle \nabla f(\boldsymbol{y}^{(t)}), \boldsymbol{z}^{(t+1)} - \boldsymbol{x} \right\rangle$$

$$+ \frac{L}{2} \|\boldsymbol{z}^{(t+1)} - \boldsymbol{y}^{(t)}\|_2^2. \tag{109}$$

Then, plugging equation (107), and equation (108) into equation (109) we obtain that

$$F(\boldsymbol{z}^{(t+1)}) \leq g(\boldsymbol{z}^{(t+1)}) + f(\boldsymbol{x}) + \left\langle \nabla f(\boldsymbol{y}^{(t)}), \boldsymbol{z}^{(t+1)} - \boldsymbol{x} \right\rangle + \frac{1}{\kappa_f} \|\boldsymbol{x} - \boldsymbol{y}^{(t)}\|_2^2 + \frac{L}{2} \|\boldsymbol{z}^{(t+1)} - \boldsymbol{y}^{(t)}\|_2^2$$

$$\leq F(\boldsymbol{x}) + \left\langle \nabla f(\boldsymbol{y}^{(t)}) + \frac{1}{\alpha_1} (\boldsymbol{z}^{(t+1)} - \boldsymbol{y}^{(t)}), \boldsymbol{x} - \boldsymbol{z}^{(t+1)} \right\rangle + \left\langle \nabla f(\boldsymbol{y}^{(t)}), \boldsymbol{z}^{(t+1)} - \boldsymbol{x} \right\rangle$$

$$+ \frac{1}{\kappa_f} \|\boldsymbol{x} - \boldsymbol{y}^{(t)}\|_2^2 + \frac{1}{\kappa_g} \|\boldsymbol{x} - \boldsymbol{z}^{(t+1)}\|_2^2 + \frac{L}{2} \|\boldsymbol{z}^{(t+1)} - \boldsymbol{y}^{(t)}\|_2^2$$

$$\leq F(\boldsymbol{x}) + \frac{1}{\alpha_1} \left\langle \boldsymbol{z}^{(t+1)} - \boldsymbol{y}^{(t)}, \boldsymbol{x} - \boldsymbol{z}^{(t+1)} \right\rangle + \frac{L}{2} \|\boldsymbol{z}^{(t+1)} - \boldsymbol{y}^{(t)}\|_2^2$$

$$+ \frac{1}{\kappa} \left( \|\boldsymbol{x} - \boldsymbol{y}^{(t)}\|_2^2 + \|\boldsymbol{x} - \boldsymbol{z}^{(t+1)}\|_2^2 \right), \tag{110}$$

where $\frac{1}{\kappa} = \max\{\frac{1}{\kappa_f}, \frac{1}{\kappa_g}\}$. Observe that the last term in the above equation (110) satisfies

$$\frac{1}{\kappa} \left( \|\boldsymbol{x} - \boldsymbol{y}^{(t)}\|_2^2 + \|\boldsymbol{x} - \boldsymbol{z}^{(t+1)}\|_2^2 \right) \leq \frac{1}{\kappa} \|\boldsymbol{z}^{(t+1)} - \boldsymbol{y}^{(t)}\|_2^2 - \frac{2}{\kappa} \left\langle \boldsymbol{z}^{(t+1)} - \boldsymbol{y}^{(t)}, \boldsymbol{x} - \boldsymbol{z}^{(t+1)} \right\rangle. \tag{111}$$

Combining equation (110) and equation (111) we obtain

$$F(\boldsymbol{z}^{(t+1)}) \leq F(\boldsymbol{x}) + \left( \frac{1}{\alpha_1} - \frac{2}{\kappa} \right) \left\langle \boldsymbol{z}^{(t+1)} - \boldsymbol{y}^{(t)}, \boldsymbol{x} - \boldsymbol{z}^{(t+1)} \right\rangle + \left( \frac{L}{2} + \frac{1}{\kappa} \right) \|\boldsymbol{z}^{(t+1)} - \boldsymbol{y}^{(t)}\|_2^2$$

$$= F(\boldsymbol{x}) + \left( \frac{1}{\alpha_1} - \frac{2}{\kappa} \right) \left\langle \boldsymbol{z}^{(t+1)} - \boldsymbol{y}^{(t)}, \boldsymbol{x} - \boldsymbol{y}^{(t)} \right\rangle - \left( \frac{1}{\alpha_1} - \frac{L}{2} - \frac{3}{\kappa} \right) \|\boldsymbol{z}^{(t+1)} - \boldsymbol{y}^{(t)}\|_2^2. \tag{112}$$

Thus, if $\alpha_1 < \frac{\kappa}{L\kappa + 4}$ then we have

$$F(\boldsymbol{z}^{(t+1)}) \leq F(\boldsymbol{x}) + \left( \frac{1}{\alpha_1} - \frac{2}{\kappa} \right) \left\langle \boldsymbol{z}^{(t+1)} - \boldsymbol{y}^{(t)}, \boldsymbol{x} - \boldsymbol{y}^{(t)} \right\rangle - \left( \frac{1}{2\alpha_1} - \frac{1}{\kappa} \right) \|\boldsymbol{z}^{(t+1)} - \boldsymbol{y}^{(t)}\|_2^2. \tag{113}$$

Let $\boldsymbol{x} = \boldsymbol{x}^{(t)}$ and $\boldsymbol{x}^*$, we have

$$F(\boldsymbol{z}^{(t+1)}) \leq F(\boldsymbol{x}^{(t)}) + \left( \frac{1}{\alpha_1} - \frac{2}{\kappa} \right) \left\langle \boldsymbol{z}^{(t+1)} - \boldsymbol{y}^{(t)}, \boldsymbol{x}^{(t)} - \boldsymbol{y}^{(t)} \right\rangle - \left( \frac{1}{2\alpha_1} - \frac{1}{\kappa} \right) \|\boldsymbol{z}^{(t+1)} - \boldsymbol{y}^{(t)}\|_2^2 \tag{114}$$

$$F(\boldsymbol{z}^{(t+1)}) \leq F(\boldsymbol{x}^*) + \left( \frac{1}{\alpha_1} - \frac{2}{\kappa} \right) \left\langle \boldsymbol{z}^{(t+1)} - \boldsymbol{y}^{(t)}, \boldsymbol{x}^* - \boldsymbol{y}^{(t)} \right\rangle - \left( \frac{1}{2\alpha_1} - \frac{1}{\kappa} \right) \|\boldsymbol{z}^{(t+1)} - \boldsymbol{y}^{(t)}\|_2^2, \tag{115}$$

Define $\frac{1}{\alpha'} = \left( \frac{1}{\alpha_1} - \frac{2}{\kappa} \right)$. Then, multiplying equation (114) by $r_t - 1$ and adding equation (115) we have

$$r_t F(\boldsymbol{z}^{(t+1)}) - (r_t - 1)F(\boldsymbol{x}^{(t)}) - F(\boldsymbol{x}^*)$$

$$\leq \frac{1}{\alpha'} \left\langle \boldsymbol{z}^{(t+1)} - \boldsymbol{y}^{(t)}, (r_t - 1)(\boldsymbol{x}^{(t)} - \boldsymbol{y}^{(t)}) + \boldsymbol{x}^* - \boldsymbol{y}^{(t)} \right\rangle - \frac{r_t}{2\alpha'} \|\boldsymbol{z}^{(t+1)} - \boldsymbol{y}^{(t)}\|_2^2. \tag{116}$$

So we have

$$r_t \left( F(\boldsymbol{z}^{(t+1)}) - F(\boldsymbol{x}^*) \right) - (r_t - 1) \left( F(\boldsymbol{x}^{(t)}) - F(\boldsymbol{x}^*) \right)$$
$$\leq \frac{1}{\alpha'} \left\langle \boldsymbol{z}^{(t+1)} - \boldsymbol{y}^{(t)}, (r_t - 1)(\boldsymbol{x}^{(t)} - \boldsymbol{y}^{(t)}) + \boldsymbol{x}^* - \boldsymbol{y}^{(t)} \right\rangle - \frac{r_t}{2\alpha'} \|\boldsymbol{z}^{(t+1)} - \boldsymbol{y}^{(t)}\|_2^2. \tag{117}$$

Multiplying both sides by $r_t$ and using $(r_t)^2 - r_t = (r_{t-1})^2$ from APGM we have

$$(r_t)^2 \left( F(\boldsymbol{z}^{(t+1)}) - F(\boldsymbol{x}^*) \right) - (r_{t-1})^2 \left( F(\boldsymbol{x}^{(t)}) - F(\boldsymbol{x}^*) \right)$$
$$\leq \frac{1}{\alpha'} \left\langle r_t \left( \boldsymbol{z}^{(t+1)} - \boldsymbol{y}^{(t)} \right), (r_t - 1) \left( \boldsymbol{x}^{(t)} - \boldsymbol{y}^{(t)} \right) + \boldsymbol{x}^* - \boldsymbol{y}^{(t)} \right\rangle - \frac{1}{2\alpha'} \|r_t(\boldsymbol{z}^{(t+1)} - \boldsymbol{y}^{(t)})\|_2^2$$
$$= \frac{1}{\alpha'} \left\langle r_t \left( \boldsymbol{z}^{(t+1)} - \boldsymbol{y}^{(t)} \right), (r_t - 1)\boldsymbol{x}^{(t)} - r_t \boldsymbol{y}^{(t)} + \boldsymbol{x}^* \right\rangle - \frac{1}{2\alpha'} \|r_t(\boldsymbol{z}^{(t+1)} - \boldsymbol{y}^{(t)})\|_2^2$$
$$= \frac{1}{2\alpha'} \left( \|(r_t - 1)\boldsymbol{x}^{(t)} - r_t \boldsymbol{y}^{(t)} + \boldsymbol{x}^*\|_2^2 - \|(r_t - 1)\boldsymbol{x}^{(t)} - r_t \boldsymbol{z}^{(t+1)} + \boldsymbol{x}^*\|_2^2 \right). \tag{118}$$

Define

$$U^{(t+1)} = r_t \boldsymbol{z}^{(t+1)} - (r_t - 1)\boldsymbol{x}^{(t)} - \boldsymbol{x}^*. \tag{119}$$

Let

$$U^{(t)} = r_{t-1} \boldsymbol{z}^{(t)} - (r_{t-1} - 1)\boldsymbol{x}^{(t-1)} - \boldsymbol{x}^* = r_t \boldsymbol{y}^{(t)} - (r_t - 1)\boldsymbol{x}^{(t)} - \boldsymbol{x}^*. \tag{120}$$

We have

$$\boldsymbol{y}^{(t)} = \frac{r_{t-1} \boldsymbol{z}^{(t)} - (r_{t-1} - 1)\boldsymbol{x}^{(t-1)} + (r_t - 1)\boldsymbol{x}^{(t)}}{r_t}$$
$$= \boldsymbol{x}^{(t)} + \frac{r_{t-1}}{r_t}(\boldsymbol{z}^{(t)} - \boldsymbol{x}^{(t)}) + \frac{r_{t-1} - 1}{r_t}(\boldsymbol{x}^{(t)} - \boldsymbol{x}^{(t-1)}), \tag{121}$$

which is the same in Step 1 of APGM. So we have

$$(r_t)^2 \left( F(\boldsymbol{z}^{(t+1)}) - F(\boldsymbol{x}^*) \right) - (r_{t-1})^2 \left( F(\boldsymbol{x}^{(t)}) - F(\boldsymbol{x}^*) \right) \leq \frac{1}{2\alpha'} \left( \|U^{(t)}\|_2^2 - \|U^{(t+1)}\|_2^2 \right). \tag{122}$$

If $F(\boldsymbol{z}^{(t+1)}) \leq F(\boldsymbol{v}^{(t+1)})$, then $\boldsymbol{x}^{(t+1)} = \boldsymbol{z}^{(t+1)}$

$$(r_t)^2 \left( F(\boldsymbol{z}^{(t+1)}) - F(\boldsymbol{x}^*) \right) - (r_{t-1})^2 \left( F(\boldsymbol{x}^{(t)}) - F(\boldsymbol{x}^*) \right)$$
$$= (r_t)^2 \left( F(\boldsymbol{x}^{(t+1)}) - F(\boldsymbol{x}^*) \right) - (r_{t-1})^2 \left( F(\boldsymbol{x}^{(t)}) - F(\boldsymbol{x}^*) \right)$$
$$\leq \frac{1}{2\alpha'} \left( \|U^{(t)}\|_2^2 - \|U^{(t+1)}\|_2^2 \right). \tag{123}$$

If $F(\boldsymbol{z}^{(t+1)}) > F(\boldsymbol{v}^{(t+1)})$, then $\boldsymbol{x}^{(t+1)} = \boldsymbol{v}^{(t+1)}$. So,

$$(r_t)^2 \left( F(\boldsymbol{x}^{(t+1)}) - F(\boldsymbol{x}^*) \right) - (r_{t-1})^2 \left( F(\boldsymbol{x}^{(t)}) - F(\boldsymbol{x}^*) \right)$$
$$\leq (r_t)^2 \left( F(\boldsymbol{z}^{(t+1)}) - F(\boldsymbol{x}^*) \right) - (r_{t-1})^2 \left( F(\boldsymbol{x}^{(t)}) - F(\boldsymbol{x}^*) \right)$$
$$\leq \frac{1}{2\alpha'} \left( \|U^{(t)}\|_2^2 - \|U^{(t+1)}\|_2^2 \right). \tag{124}$$

Summing equation (124) over $t = 1, \ldots, T$ we have

$$(r_{T+1})^2 (F(\boldsymbol{x}^{(T+1)}) - F(\boldsymbol{x}^*))$$
$$= (r_{T+1})^2 (F(\boldsymbol{x}^{(T+1)}) - F(\boldsymbol{x}^*)) - (r_0)^2 (F(\boldsymbol{x}^{(1)}) - F(\boldsymbol{x}^*))$$
$$\leq \frac{1}{2\alpha'} \left( \|U^{(1)}\|_2^2 - \|U^{(T+1)}\|_2^2 \right) \leq \frac{1}{2\alpha'} \|U^{(1)}\|_2^2 = \frac{1}{2\alpha'} \|\boldsymbol{x}^{(0)} - \boldsymbol{x}^*\|_2^2. \tag{125}$$

From APGM we can easily have that $r_t \geq \frac{t+1}{2}$. So we have

$$F(\boldsymbol{x}^{(T+1)}) - F(\boldsymbol{x}^*) \leq \frac{2}{\alpha'(T+1)^2} \|\boldsymbol{x}^{(0)} - \boldsymbol{x}^*\|_2^2, \tag{126}$$

where $\frac{1}{\alpha'} = \left( \frac{1}{\alpha_1} - \frac{2}{\kappa} \right)$, and $\alpha_1 < \frac{\kappa}{L\kappa+4}$, with $\frac{1}{\kappa} = \max\{\frac{1}{\kappa_f}, \frac{1}{\kappa_g}\}$ for $k_f < \frac{1}{L}$. $\qquad\square$

REMARKS ON STABILITY OF APGM

The APGM stability and reliability come from the pivotal factors: 1) APGM employs two prox-imal steps that have a unique solution (as rigorously demonstrated in the section above), aiding convergence and stability. 2) As shown in the section above, the sequence $x^{(t+1)}$, constructed by APGM algorithm, always converge to global optima irrespective of the initial state $x^{(0)}$, conferring a steadfast assurance of reliable attainment of optimal solutions. 3) The remarkable effectiveness demonstrated across an array of real-world applications (Beck & Teboulle, 2009; Ochs et al., 2014; Boț et al., 2016) reaffirms APGM's reliability.

## K   ADDITIONAL IMPLEMENTATION DETAILS AND RESULTS

In this section we present additional numerical results and details to complement the three experi-ments in Section 5. For Experiments 1, and 2 the constants $c, \alpha$ of the adaptive invex loss in equa-tion (7) are implemented as

$$c = \text{softplus}(c_{var}) + c_{min} \tag{127}$$
$$\alpha = (\alpha_{max} - \alpha_{min})\text{sigmoid}(\alpha_{var}) + \alpha_{min} \tag{128}$$

where $c_{var}$, and $\alpha_{var}$ are parameters to be learned simultaneously with the network weights using the same Adam optimizer instance. In addition, we fix $\alpha_{max} = 1.99$, $\alpha_{min} = 0.0$, and $c_{min} = 10^{-8}$.

### K.1   EXPERIMENT 1

To train FISTA-Net[2] for the CT experiment using convex, invex and quasi-invex regularizers, we use a batch size of 64, optimized using Adam, and with seven hidden layers. Datasets used for train-ing, validation and testing were normalized to the interval of $[0, 1]$. In addition to the experiments reported in Table 3 for Experiment 1, we show the performance of invex regularizers reported in (Pinilla et al., 2022a) under FISTA-Net. These results are summarized in Table 6.

Table 6: Performance Comparison: Lower the better. We highlight the best, and the worst results for each metric. (Green: Best, and Red: Worst).

| | | Experiment 1 (Combination of functions used to train FISTA-Net) | | | | | |
|---|---|---|---|---|---|---|---|
| **Regularizer** | Metrics | equation (4) | equation (5) | equation (6) | equation (7) | equation (8) | $\ell_2$-norm |
| equation 7 in (Pinilla et al., 2022a) | AbsError | 0.2380 | 0.2106 | 0.2212 | 0.1867 | 0.1973 | 0.2556 |
| | PSNR | 38.36 | 39.09 | 38.65 | 39.13 | 39.08 | 38.08 |
| | RMSE | 0.0150 | 0.0124 | 0.0123 | 0.0105 | 0.0095 | 0.0174 |
| | SSIM | 0.9550 | 0.9574 | 0.9566 | 0.9590 | 0.9582 | 0.9558 |
| equation 8 (Pinilla et al., 2022a) | AbsError | 0.2768 | 0.2501 | 0.2352 | 0.2240 | 0.2586 | 0.2941 |
| | PSNR | 38.05 | 38.33 | 38.72 | 38.87 | 38.78 | 37.79 |
| | RMSE | 0.0179 | 0.0104 | 0.0145 | 0.0137 | 0.0117 | 0.0217 |
| | SSIM | 0.9537 | 0.9554 | 0.9563 | 0.9580 | 0.9571 | 0.9545 |
| equation 10 (Pinilla et al., 2022a) | AbsError | 0.2088 | 0.1819 | 0.1700 | 0.1601 | 0.1933 | 0.2259 |
| | PSNR | 38.68 | 39.40 | 39.32 | 39.54 | 38.99 | 38.38 |
| | RMSE | 0.0129 | 0.0087 | 0.0111 | 0.0108 | 0.0095 | 0.0146 |
| | SSIM | 0.9563 | 0.9585 | 0.9578 | 0.9600 | 0.9593 | 0.9571 |

### K.2   EXPERIMENT 2

To train MST++[3] using the invex loss functions we use a batch size of 20, optimized using Adam with parameters $\beta_1 = 0.9$, and $\beta_2 = 0.999$, and the cosine Annealing scheme is adopted for 300 epochs. Datasets used for training, validation and testing were normalized to the interval of $[0, 1]$. To complement results in Table 3 we present Figure 2 which reports some restored spectral images using MST++ trained with losses equation (4)-equation (8). To evaluate the performance we employ the absolute error.

---

[2]code of can be found at `https://github.com/jinxixiang/FISTA-Net`
[3]Implementation can be found in `https://github.com/caiyuanhao1998/MST-plus-plus`

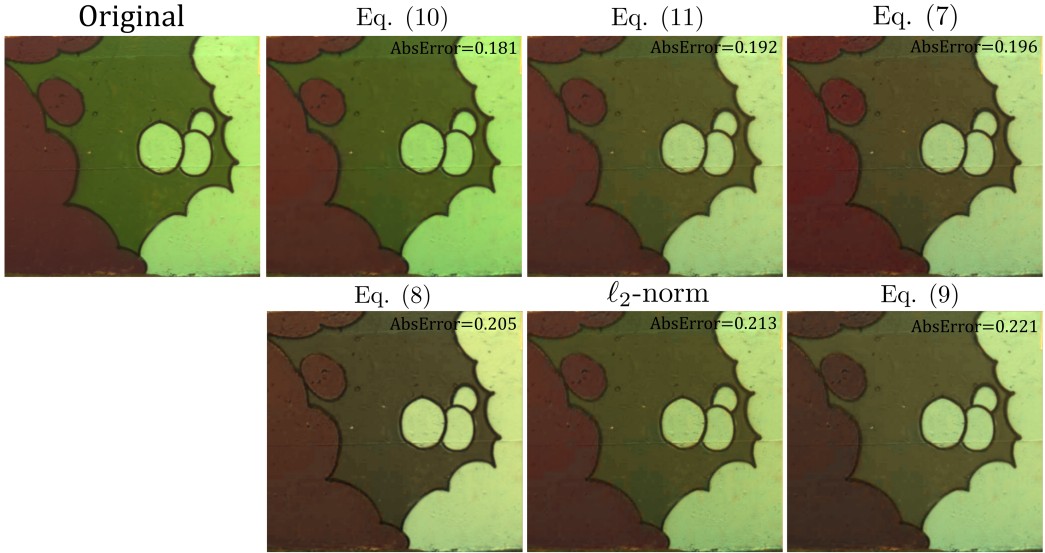

Figure 2: Reconstructed spectral images with MST++ trained with losses $\ell_2$-norm, and the invex functions equation (4)-equation (8). To evaluate the performance we employ the absolute error.

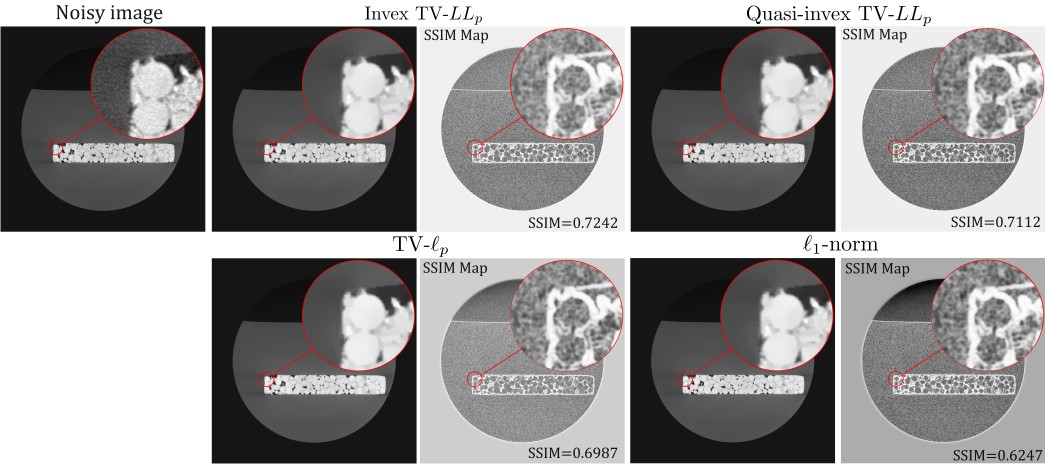

Figure 3: Restored images using the ADMM iterative process in equation (129) to solve program (11) using total-variation-like invex/quasi-invex regularizers $g$ of $\ell_p$-quasinorm, and equation (3) equation (10). To evaluate the performance we employ the structural similarity index measure (SSIM) by reporting the SSIM map for each image and its averaged value. Recall that SSIM is reported in the range [0,1] where 1 is the best achievable quality and 0 the worst. In the SSIM map small values of SSIM appear as dark pixels.

## K.3    EXPERIMENT 3

In order to implement the ADMM iterative process in equation (13) to solve the total variation filtering optimization problem in (11), we use a linear approximation to update $\boldsymbol{x}^{(t+1)}$ based on (Ouyang et al., 2015). The reason of this approximation comes from the fact that updating $\boldsymbol{x}^{(t+1)}$ for the total variation filtering optimization problem in (11) requires the computation of $(\rho \boldsymbol{D}^T \boldsymbol{D} + \boldsymbol{I})^{-1}$ which is a computationally expensive step. Therefore, we implement the following ADMM iteration process to solve program (11) for the total-variation-like invex/quasi-invex regularizers $g$ of $\ell_p$-quasinorm, and equation (3) equation (10)

$$\boldsymbol{x}_2^{(t+1)} = (1-\alpha)\boldsymbol{x}^{(t)} + \alpha\boldsymbol{x}_1^{(t)}$$

$$\boldsymbol{x}_1^{(t+1)} = \boldsymbol{x}_1^{(t)} - \frac{1}{2\beta}\left(\rho\boldsymbol{D}^T(\boldsymbol{D}\boldsymbol{x}_2^{(t+1)} - \boldsymbol{z}^{(t)} + \boldsymbol{v}^{(t)}) + \boldsymbol{x}_2^{(t+1)} - \boldsymbol{u}\right)$$

$$\boldsymbol{x}^{(t+1)} = (1-\alpha)\boldsymbol{x}^{(t)} + \alpha\boldsymbol{x}_1^{(t+1)} \tag{129}$$

$$\boldsymbol{z}^{(t+1)} = \text{Prox}_{\lambda/\rho g}(\boldsymbol{D}\boldsymbol{x}^{(t+1)} + \boldsymbol{v}^{(t)}/\rho)$$

$$\boldsymbol{v}^{(t+1)} = \boldsymbol{v}^{(t)} + \rho(\boldsymbol{D}\boldsymbol{x}^{(t+1)} - \boldsymbol{z}^{(t+1)}),$$

where $\boldsymbol{D}$ is the discrete spatial derivative (see Section B for more details), the first three equality is the linear approximation to estimate $\boldsymbol{x}^{(t+1)}$ following (Ouyang et al., 2015), $\boldsymbol{u}$ is the noisy image, and $\alpha, \rho, \beta$ are positive constants chosen to be the best for each analyzed function determined by cross-validation. This iterative procedure is initialized as $\boldsymbol{x}^{(0)} = \boldsymbol{u}$, $\boldsymbol{z}^{(0)} = \boldsymbol{D}\boldsymbol{x}^{(0)}$, and $\boldsymbol{x}_1^{(0)} = \boldsymbol{x}_2^{(0)} = \boldsymbol{v}^{(0)} = \boldsymbol{0}$. Further, the proximal of regularizers $g$ to compute $\boldsymbol{z}^{(t+1)}$ are given in Table 4. To complement results in Table 3 we present Figure 3 which reports some restored images obtained by total-variation-like invex/quasi-invex regularizers $g$ of $\ell_p$-quasinorm, and equation (3) equation (10). To numerically evaluate their performance we employ the structural similarity index measure (SSIM) by reporting the SSIM map for each denoised image and its averaged value. Recall that SSIM is reported in the range [0,1] where 1 is the best achievable quality and 0 the worst. In the SSIM map small values of SSIM appear as dark pixels.

## K.4 EXPERIMENTS WITH APGM AND VALIDITY OF THEOREM 7

In this section we assess the performance of APGM (Li & Lin, 2015) solving compressive sensing problems. Specifically, this section provides two experiments summarized in Tables 7, and 8. The aim of the first experiment is to numerically validate Theorem 7 and the second it is to show the performance of APGM in a realistic setting when invex/quasi-invex are employed.

For the first experiment, we generate noisy measurements $\boldsymbol{y} = \boldsymbol{A}\boldsymbol{x} + \boldsymbol{\eta}$ where $\boldsymbol{A} \in \mathbb{R}^{m \times n}$ follows a Gaussian random matrix with $\ell_2$-normalized columns, $\boldsymbol{x} \in \mathbb{R}^n$ is a $k$-sparse vector, and $\boldsymbol{\eta} \in \mathbb{R}^m$ models the noise. Specifically, the size $n$ and the sparsity $k$ of the unknown underlying signal $\boldsymbol{x}$ are fixed as $n = 1024$, and $k = 0.03n$. Therefore, appealing to the theoretical result in (Foucart & Rauhut, 2013, Theorem 9.13), we determined from the fixed values of $n$ and $k$ the number of measurements $m$ as $m = 0.45n$ in order to satisfy equation (67) for $\tau = 0.3$ and $\rho = 0.4$. In consequence, the constant $C$ and $D$ of equation (72) are given by $C = 4.6$, and $D = 2\frac{g(\boldsymbol{x})}{\|\boldsymbol{x}\|_1}$. Under this setting, we are ready to validate inequality in equation (72) numerically (i.e. validity of Theorem 7) where the regularizer $g(\boldsymbol{x})$ in program (1) takes the form of equations (3), (9),(10), and the data fidelity term $f(\boldsymbol{x})$ takes the form of equations (5),(7),(8) (these were chosen due their performance in Table 3). Further, in order to solve program (1) and estimate the distance $\|\boldsymbol{x} - \boldsymbol{x}^*\|_1$ in equation (72), we use APGM (see Appendix J) where $\boldsymbol{x}^*$ is the solution returned by APGM. We consider additive white Gaussian noise in the measurements data vector with three different levels of SNR (Signal-to-Noise Ratio) $= 20, 30$. The number of iterations $T$ of the APGM is fixed to $T = 1000$. The positive constants $\alpha_1, \alpha_2$ of APGM, and $c, \alpha, p$ of equations (7),(10) are chosen to be the best for each analyzed combination of functions by cross-validation.

We summarize the results in Table 7. From these numerical results, we observe that invex/quasi-invex mappings for both regularizer and fidelity term perform better than using the combination of $\ell_1$-norm and $\ell_2$-norm because the upper bound in equation (72) is smaller. Additionally, the results in Table 7 suggest this better lower upper bound is due to the regularizer. This is an expected behavior since this is a compressive sensing test, and the regularizer is the mapping that determines the reconstruction quality (Candès & Wakin, 2008).

For the second experiment, we use 24 images of size $256 \times 256$ (i.e. $n = 65536$) from the Kodak dataset (kod), and we convert them to grayscale. In order to have noisy measurements $\boldsymbol{y}$ in a compressive sensing setup, we compute the Hadamard transform over these images $\boldsymbol{x}$ (which is sparse), after we add Cauchy distributed noise with a dispersion $\sigma = 1$, and we randomly select $m = 32000$ noisy transformed coefficients. Therefore, assuming $\boldsymbol{A}$ is a submatrix of size $m \times n$ of the Hadamard transform, then $\boldsymbol{y} = \boldsymbol{A}\boldsymbol{x} + \boldsymbol{\eta}$ where $\boldsymbol{\eta}$ is the Cauchy distributed noise. Thus, these experiments are intended to recover $\boldsymbol{x}$ from the noisy measurements $\boldsymbol{y}$ using APGM, where the data

Table 7: Performance Results: Best: green, and the worst: red. Column named "dist" reports $\|\boldsymbol{x} - \boldsymbol{x}^*\|_1$ where $\boldsymbol{x}^*$ is the solution returned by APGM for each combination of $g(\boldsymbol{x})$, and $f(\boldsymbol{x})$ mappings. Additionally, column named "bound" reports $C\beta_{k,g}(\boldsymbol{x}) + D\sqrt{\epsilon}\upsilon_{f,\eta}(\boldsymbol{x})$ in equation (72) for each combination of $g(\boldsymbol{x})$, and $f(\boldsymbol{x})$ mappings.

| | | Tested $f(\boldsymbol{x})$ functions for APGM | | | | | | | |
| | | (5) | | (7) | | (8) | | $\ell_2$-norm | |
| $g(\boldsymbol{x})$ | SNR | dist | bound | dist | bound | dist | bound | dist | bound |
|---|---|---|---|---|---|---|---|---|---|
| (3) | 20dB | 0.401 | 1.330 | 0.420 | 1.330 | 0.442 | 1.330 | 0.467 | 1.330 |
| | 30dB | 0.218 | 0.453 | 0.188 | 0.453 | 0.217 | 0.453 | 0.260 | 0.453 |
| (9) | 20dB | 0.588 | 5.255 | 0.671 | 5.255 | 0.783 | 5.255 | 0.941 | 5.255 |
| | 30dB | 0.374 | 1.632 | 0.332 | 1.632 | 0.370 | 1.632 | 0.426 | 1.632 |
| (10) | 20dB | 0.477 | 1.330 | 0.516 | 1.330 | 0.565 | 1.330 | 0.624 | 1.330 |
| | 30dB | 0.275 | 0.453 | 0.240 | 0.453 | 0.273 | 0.453 | 0.323 | 0.453 |
| $\ell_1$-norm | 20dB | 0.768 | 5.431 | 0.959 | 5.431 | 1.278 | 5.431 | 1.916 | 5.431 |
| | 30dB | 0.585 | 1.857 | 0.541 | 1.857 | 0.573 | 1.857 | 0.627 | 1.857 |

fidelity functions $f(\boldsymbol{x})$ are the $\ell_2$-norm, and equation (4)-equation (8), and the regularizers $g(\boldsymbol{x})$ are the $\ell_1$-norm and equation (3), equation (9), equation (10). We point out that the combination of $\ell_2$-norm as data fidelity term and $\ell_1$-norm as regularizer is taken as the convex state-of-the-art. The number of iterations $T$ of the APGM is fixed to $T = 1000$. The positive constants $\alpha_1, \alpha_2$ of APGM, and $c, \alpha, p$ of equations (7),(10) are chosen to be the best for each analyzed combination of functions by cross-validation. The performance for each combination of functions is measured by averaging the peak-signal-to-noise-ratio (PSNR) in dB over the image set. These results are summarized in Table 8. These results suggest that the best result is obtained employing regularizer in equation (3) and fidelity term in equation (7). The intuition behind the superiority combining these two mapping comes from the possibility of adjusting the value of $p$ for equation (3) and $c, \alpha$ for equation (7) in data-dependent manner (Wu & Chen, 2013; Barron, 2019).

Table 8: Performance Results: Best: green, and the worst: red.

| | | Tested $f(\boldsymbol{x})$ functions for APGM | | | | | |
| **Regularizer** | Metric | equation (4) | equation (5) | equation (6) | equation (7) | equation (8) | $\ell_2$-norm |
|---|---|---|---|---|---|---|---|
| equation (3) | PSNR | 20.53 | 17.48 | 16.27 | 22.50 | 18.88 | 15.25 |
| equation (9) | PSNR | 18.03 | 15.64 | 14.67 | 19.52 | 16.76 | 13.83 |
| equation (10) | PSNR | 19.20 | 16.51 | 15.43 | 20.90 | 17.75 | 14.50 |
| $\ell_1$-norm | PSNR | 17.00 | 14.87 | 13.99 | 18.32 | 15.87 | 13.22 |

