# OpenReview forum: "Global Optimality for Non-linear Constrained Restoration Problems via Invexity"
_ICLR.cc/2024/Conference — ICLR 2024 poster_

### Official Review · Reviewer_4Bfb · 2023-10-28

**Soundness:** 3 good
**Presentation:** 1 poor
**Contribution:** 3 good
**Rating:** 6
**Confidence:** 3

**Summary:**

This work uses the notion of invexity and quasi-invexity in non-convex constrained inverse problems to find global optima.  Through application of admissible functions they build some mathematical tools to show invexity or quasi-invexity of some functions mentioned in the paper. These are the functions which are later used as the loss function or the regularizer in equation (1). Multiple applications of their analysis are studied such as compressive image restoration, total variation filtering, and E-ADMM. Experimental results are provided to support their claims.

**Strengths:**

1- sound paper,

2- well-established connection between invexity and various applications,

3- interesting theoretical results.

**Weaknesses:**

1- In my opinion, the paper needs major revision in writing, organization, and presentation.

2- In many applications the computational complexity of signal recovery is very high (e.g. in image restoration, application of low-rank matrix recovery in sparse image recovery). This issue was not discussed.

3-The limitations are not mentioned.

4- Some points which might help improving the presentation of the paper

(a) The text is generic in some parts. E.g. in conclusion you mention that your results are very promising, but you do not specify in what sense.

(b) I think Table 3 does not need the second best results highlighted.

(c)There are some"hard to read" sentence in the text. One example in the introduction is:

"The lack of approaches for guaranteeing the global minima in non-convex settings leaves any optimization-based algorithmic solution incomplete, non-unique, and hence cannot be categorically accepted as the best possible solution despite their improved performance over convex functions."

(d) The E-ADMM sounds like a new algorithm is proposed. Instead, you showed the convergence of the same ADMM under the assumption of invexity/quasi-invexity. The name "E-ADMM" causes confusion.

(e) (11) is referred to as equation. It is an optimization problem.

5- Background on applications in experiments 1,2,3 is not suggested. This is the case at the beginning of every section. Maybe having a separate related work section could help with organizing the paper.

6- I appreciate the vast context of the paper (signal restoation in a general sense). I think in such cases when the topic entails many applications, you can limit your attention to the main pioneer works in the field. This helps the reader to follow references easier. When one faces 5,6 references on similar topic it might be discouraging or hard to choose references efficiently. I believe it is the writer’s task to present the most efficient references related to their work.

**Questions:**

1- Was there any computational difference when replacing $L_1$-norm and $L_2$-norm in your first experiment with the proposed functions?

2- Theorems 5 and 6  prove that some previously known functions satisfy invexity/quasi invexity property. Is it only due to invexity/quasi-invexity that these functions have superior performance in applications listed in table 2?

3- Do you have any experiments confirming the result in Theorem 7?

---

> ### Author Response · Authors · 2023-11-17
> **Response to reviewer 4Bfb (part 1)**
>
> Thank you for the time reviewing our manuscript. We appreciate the reviewer's advice to improve our manuscript. We address the comments about weaknesses. The changes to the manuscript and supplemental material are highlighted blue.
>
> # Discussion about weaknesses
> **W1.** Thank you for the time reviewing our manuscript. We appreciate the reviewer's advice to improve our manuscript. We believe we have reviewed the whole manuscript and performed major changes to improve writing, organization and presentation following the reviewer guidelines. These changes are highlighted in the revised manuscript in blue, and due to the length of the changes, we are not reporting the modified texts here. We kindly ask the reviewer to see the new version.
>
> **W2.** We agree that there are many applications within the realm of  signal recovery where the computational complexity is high such as matrix completion. We believe that to address the reviewer's comment properly a deeper and separate study has to be done since we purely rely on ADMM and APG methods, and as such, efficient implementation is beyond the scope of this paper. To clarify this we have modified Section 6 of our manuscript clearly stating that this is an interesting direction for a future work, which we intend to explore.
>
> In contrast, what we have covered in this manuscript regarding computational complexity it is the convergence rate of ADDM and APG methods using our proposed family of invex/quasi-invex functions. Specifically, we have proven that both ADMM and APG methods using our proposed family of invex/quasi-invex functions are as efficient as solving the well-established convex optimization problems. We reported this in Theorems 9 and 11 (supp. material) by stating that ADMM and APG for our invex/quasi-invex functions have convergence rates of $\mathcal{O}(1/t)$ and $\mathcal{O}(1/t^{2})$ respectively, which are the same in the traditional convex case.
>
> Therefore, considering reviewer's comment we have modified our manuscript to further clarify that ADMM and APG methods using our proposed family of invex/quasi-invex functions are as efficient as solving the well-established convex optimization problems. We kindly ask the reviewer to see the new Introduction and Conclusions sections.
>
> **W3.** Thank you for highlighting this aspect. Considering the comment we have modified our manuscript and include limitations of our work. This reads as follows:
>
> **Modified text in Section 6:** While the theoretical analysis presented here to handle signal restoration problems is the first of its kind and performs well on a number of imaging tasks, its applications beyond imaging tasks are yet to be explored. More specifically, the application of these algorithms around deep learning research yet to be explored, which can pave a way to improve several downstream applications, such as low-rank matrix recovery and alike. As a pure algorithm, a number of things can be improved, such as decoupling the $\eta$ from $\upsilon_{f,\eta}(x)$, improving the efficiency of the algorithm beyond what ADMM and APG can offer.
>
> **W4.1** We have reviewed the whole manuscript including the mentioned sentence, and performed major changes to improve writing, organization and presentation following the reviewer guidelines. These changes are highlighted in the revised manuscript in blue, and due to the length of the changes, we are not reporting the modified texts here. We kindly ask the reviewer to see the new version.
>
> **W4.2** We have now modified Table 3 only to highlight the best and worst results.
>
> **W4.3** In addition revising the whole manuscript, we have also improved this sentence (by breaking them into two sentences) and performed similar changes throughout the paper. These changes are highlighted in the revised manuscript in blue, and due to the length of the changes, we are not reporting the modified texts here. We kindly ask the reviewer to see the new version.
>
> **W4.4** We have modified our manuscript by removing "E-" from Section 4.3 and thus avoid any further confusions.
>
> **W4.5** We have modified our manuscript by referring to optimization problem in 1 like "Program 1" or "optimization problem in 1". These changes are highlighted in the revised manuscript in blue, and due to the length of the changes, we are not reporting the modified texts here.
>
> **W5.** To address this comment we have provided a short background at the beginning of each experiment guiding the reader with meaningful references related to our work, instead of a separate section. We believe that this organisation ensures that the advances to the invexity/quasi-invexity within the field of optimization are valued than perceiving them as tied to specific imaging applications. These changes are highlighted in the revised manuscript in blue, and due to the length of the changes, we are not reporting the modified texts here.

---

> ### Author Response · Authors · 2023-11-17
> **Response to reviewer 4Bfb (part 2)**
>
> We address the last comments about weaknesses and questions by the reviewer. The changes to the manuscript and supplemental material are highlighted blue.
>
> # Discussion about weaknesses
> **W6.** We have now modified our manuscript to limit the reader's attention to the main works related to our paper at most three references.
>
> # Questions
> **Q1.** We would like to mention that in the supp. material, in Sections I.3 and K we provide implementation details for the experiments presented in our manuscript. From these implementation details and the nature of Experiment 1, we just need to report the wall time (runtime) to compute the proximal of invex/quasi-invex regularizers. In the case of the fidelity term, we just need to compute its derivative as a close form, therefore the runtime is negligible in practice. We report the runtime of proximal solutions in Section I.3 of supp. material. We remark that these proximal solutions are the activation functions in the training process of Experiment 1. From the numerical results reported in Section I.3 we observe that computing the proximal of the $\ell_{1}$-norm is faster than the proximal of invex regularizers. However, this difference is given in milliseconds, potentially making it negligible in practice. These changes are highlighted in the revised manuscript in blue, and due to the length of the changes, we are not reporting the modified texts here.
>
> **Q2.** Yes. It is our belief that invexity/quasi-invexity offers superior performance. To elaborate on this, we remark that the main purpose of this study is to provide a new theoretical approach that allows us to globally solve constrained optimization problems with higher reconstruction quality, than what existing convex approaches can offer.  As we have pointed out in the manuscript, the essential breakthrough of our work lies in the fact that the proposed family of invex/quasi-invex functions is closed under summation (aspect that only convexity enjoys up to this point). The significance of this result manifests on two fronts. Firstly, Theorem 3 paves the way for establishing theoretical guarantees for global optima of program 1 beyond the realm of convexity. Secondly, it bestows practical benefits to practitioners, as both Definition 4 and the third statement of Theorem 3 offer a systematic methodology for constructing invex/quasi-invex functions. As a final remark, we have shown in our manuscript that our proposed family of functions do not increase the complexity in solving constrained optimization problems.
>
> We believe that these comments addresses the concerns.
>
> **Q3.** To address the comment related to the validity of Theorem 7, we have modified the manuscript by adding a numerical experiment in Section K.4 of supp. material. There, we explain how $\beta_{k}^{g}(x)$ and $\upsilon_{f}(x)$ can be estimated in a specific compressive sensing setting,  therefore confirming numerically Theorem 7. Due to the length of the changes, we are not reporting the modified texts here. We kindly ask the reviewer to review the supp. material.

---

> ### Comment · Reviewer_4Bfb · 2023-11-19
>
> Thanks for your response and detailed explanations. I had a quick pass on the revised version of your paper. I believe there is still room for improvement of the presentation. It might be a matter of preference, but equation numbers are not referred to in parenthesis which can cause confusion. Many long sentences exist which make the text hard to read. Many sentences can be easily shorten. For example, Under Definition 4:
>
> "Definition 4 places a requirement on $f(x)$ and $g(x)$ in optimization problem 1 to operate element-wise on positive values."
>
> can become
>
> "Definition 4 requires $f(x)$ and $g(x)$ in (1) to operate element-wise on positive values."
>
> I repeat that this was just an example and many of these kind of sentences exist in the paper. I keep my score unchanged. Thanks!

---

> > ### Author Response · Authors · 2023-11-19
> > **Response to reviewer 4Bfb**
> >
> > Thank you for the time reviewing our manuscript. We appreciate the reviewer's advice to improve our manuscript. We address this comment by adding parenthesis to the equation numbers and therefore avoid further confusions. More importantly, we are interested to know whether your initial concerns have been addressed, and whether anything else requires clarification. This can help us improving the overall manuscript.

---

### Official Review · Reviewer_a9wE · 2023-10-29

**Soundness:** 2 fair
**Presentation:** 2 fair
**Contribution:** 2 fair
**Rating:** 3
**Confidence:** 4

**Summary:**

This paper studies the problem of nonconvex constrained optimization in which the objective and constraints are both assumed to be invex (or quasi-invex) functions. The main results are a list of new invex/quasi-invex regularizers and data fidelity loss functions which are constructed based on the concept of admissible functions. Several applications of the proposed invex/quasi-invex functions to compressive image restoration and total variation filtering are provided, along with an extended ADMM algorithm for global optimization. Numerical experiments on relevant tasks are conducted to evaluate the performance of the proposed models and algorithms.

**Strengths:**

The idea of using invex functions for both data fidelity and regularization terms for nonconvex image restoration sounds interesting. The proposed family of admissible functions seems useful for constructing invex loss functions with a unified treatment. The numerical results show some benefits of the proposed approach in a bunch of image restoration tasks.

**Weaknesses:**

The main concerns are in regard with the significance and correctness of technical contribution.

- As highlighted in Section 3, the authors mainly introduced a set of new invex /quasi-invex functions from the perspective of admissible functions. While these invexity-type functions are guaranteed to be globally minimal at any stationary point, a more challenging problem is how to find these points in a computationally efficient way. This challenge, however, is largely left unaddressed in the present study. The authors did mention an ADMM-style algorithm at the end of Section 4, but it is not quite clear how this algorithm could be exactly used for solving the considered problem formulation in Equation (1).

- The statements and proof arguments for some of the results in Section 4 are unclear in many places. For example, the statement of Theorem 7 reads a bit confusing, mainly due to that: 1) the comments on extensions are included at the end of the statement, and 2) the quantities $\beta_k^g(x)$ and $v_f(x)$, which are functions of $x$, are claimed as universal constants. Moreover, according the last but one line of the proof of Theorem 7, the quantity $v_f(x)$ should also be dependent on the noise vector $\eta$. In regard with Theorem 8, the first claim that $h$ is convex seems only valid for a proper range of $\lambda$, which needs to be explicitly clarified in the theorem.

**Questions:**

- Q1. Can you explain in more details how the extended ADMM algorithm as developed in Section 4.3 might be used to solve the constrained invex problem (1) with global guarantees?

- Q2. In Section 4.1, what is the exact invex problem formulation designed for compressive image restoration? Why the quantities $\beta_k^g(x)$ and $v_f(x)$ in Theorem 7 can be regarded as constants while they are obviously functions of $x$? Also can you say a bit more on the scales of these quantities in the considered problem setting?

---

> ### Author Response · Authors · 2023-11-17
> **Response to reviewer a9wE (part 1)**
>
> Thank you for the time reviewing our manuscript. We appreciate the reviewer's advice to improve our manuscript. We address the comments about weaknesses. The changes to the manuscript and supplemental material are highlighted blue.
>
> # Discussion about weaknesses
> **W1.** As this comment concerns multiple aspects, please allow us to respond these in turn, as follows:
> We start by clarifying that we have provided two different algorithms to compute these points: the alternating direction method of multipliers (ADMM) and accelerated proximal gradient methods (APG). Given that ADMM and accelerated APG are well-established convex optimization algorithms, we believe this answers the question of finding an implementation. As for the efficient implementation, we purely rely on ADMM and APG, as efficient implementation is beyond the scope of this paper. To clarify this, we prove that both ADMM and APG methods using our proposed family of invex/quasi-invex functions are as efficient as solving the well-established convex optimization problems. We reported this in Theorems 9 and 11 by stating that ADMM and APG for our invex/quasi-invex functions have convergence rates of $\mathcal{O}(1/t)$ and $\mathcal{O}(1/t^{2})$ respectively, which are the same in the traditional convex case. Additionally, realizing the significance of the implementation details, we have included additional details on the implementation for the proximal expressions in Section I.3 of the Supplementary Material.
>
> Secondly, we included a text in Section 4 stating that Appendix I includes details around how the ADMM algorithm can be used to solve problems in the form of Eq. 1 (Section titled "Remarks on Equation 1").
>
> We write here how Eq. 1 is expressed in the form needed for Theorem 9. We recall that Eq. 1 is in the form of $$\text{minimize } g(x) \text{ subject to }f(x)<\epsilon.$$
>
> This means Eq. 1 can be equivalently rewritten as $$ \text{minimize } g(x) + \lambda f(x), $$ for some $\lambda > 0$ that helps to satisfy the condition $ f(x)<\epsilon$ (please see [R1, R2] provided below that supports this argument). Thus, by introducing a variable $z=x$, Eq. (1) can be finally rewritten as $$ \text{minimize } g(x) + \lambda f(x) \text{ subject to } z=x, $$ which is in the optimization form of Theorem 9, where $h_1(x)=g(x)$, and $h_{2}(z)=\lambda f(z)$.
>
> R1 Atomic decomposition by basis pursuit.
>
> R2 Orthogonal matching pursuit for sparse signal recovery with noise.
>
> We have also mentioned in the second paragraph of Section 4, stating that Appendix J presents the extended APG.
>
> Regarding the comment on solving Eq. 1 efficiently, we remark that both Theorem 9 (about ADMM), and Theorem 11 in Appendix J (about APG) mentioned that the convergence rate of ADMM and APG when $f$, and $g$ in Eq. 1 are the invex/quasi-invex functions studied in this work is the same as in the convex traditional case.
>
> Taking these comments, we have now modified the manuscript (changes are in blue) to further clarify this aspect.
>
> **W2.** This question concerns multiple aspects, and as such,  please allow us to respond to these in turn.
>
> **W2.1.** Considering the comments, we have now modified the statement of Theorem 7, mentioning the extension over quasi-invex functions in earlier part of the statement of Theorem 7.
>
> **W2.2.** The constants $\beta_{k}^{g}(x)$ and $\upsilon_{f}(x)$ are indeed universal because they depend on the unknown desirable underlying signal $x$. To support this statement, we highlight that our Theorem 7 is in the same form as the well-established and known convex recovery result (Foucart, Rauhut, 2013, Theorem 6.12). For the sake of completeness, we present below Theorem 6.12 from (Foucart, Rauhut, 2013).
>
> **Theorem 6.12** Suppose that the $2k$th restricted isometry constant of the matrix $A\in \mathbb{R}^{m\times n}$ satisfies $\delta_{2k}<\frac{4}{\sqrt{41}}$. Also, the noise vector $\eta$ satisfies $\lVert Ax -y\rVert_{2}<\epsilon$ for $\epsilon>0$ and $x\in \mathbb{R}^{m}$ is a noisy measurement vector. If $g(x)$ is the $\ell_{1}$-norm, then the solution $x^{\star}$ of program~1 holds $\lVert x - x^{\star} \rVert_{1} \leq C \sigma_{k}(x) + D\sqrt{k}\epsilon$, where the constants $C,D>0$ depend only on $\delta_{2k}$. Additionally, we have that $$ \sigma_{k}(x) = \inf_{\lVert z\rVert_{0}<k} \lVert z - x\rVert_{1}.$$
>
> In summary, the above result highlights the significance of our Theorem 7 since it follows the same form of (Foucart \& Rauhut, 2013, Theorem 6.12), and we have modified our manuscript to mention this.
>
> However, we agree with the reviewer that $\upsilon_{f}(x)$ also depends on $\eta$. Therefore, we have corrected this typo by changing $\upsilon_{f}(x)$ to $\upsilon_{f,\eta}(x)$.
>
> **W2.3.** We agree with the reviewer that Theorem 8 h is convex for a range of $\lambda$. We have corrected this typo by adding this change to the statement of Theorem 8.
>
> Simon Foucart and Holger Rauhut. A Mathematical Introduction to Compressive Sensing.

---

> ### Author Response · Authors · 2023-11-17
> **Response to reviewer a9wE (part 2)**
>
> We address the questions by the reviewer. The changes to the manuscript and supplemental material are highlighted blue.
>
> # Questions
> **Q1.** We have addressed this in the previous comment, and we kindly ask the reviewer to refer to the first part of our rebuttal.
>
> **Q2.** As we mentioned in the manuscript, specifically in Section 4.1 and Theorem 7, the invex version of optimization problem in 1 for compressive sensing is when the constraint takes the form of $f(Ax-y)$ where $A \in \mathbb{R}^{m\times n}$ with $m\ll n$ ($f$ being an admissible function e.g. equation 4), and the regularizers $g(x)$ takes the form of admissible functions in Definition 4 (e.g. equation 3).
>
> Regarding the question about the quantities $\beta_{k}^{g}(x)$ and $\upsilon_{f}(x)$, we have addressed this in the previous comment, and we kindly ask the reviewer to refer to the previous comment.
>
> Lastly, to address the comment related to the scale of the mentioned quantities, we have modified the manuscript by adding a numerical experiment in Section K.4 of supp. material, where we explain how $\beta_{k}^{g}(x)$ and $\upsilon_{f}(x)$ can be estimated in a specific compressive sensing setting. Due to the length of the changes, we are not reporting the modified texts here. We kindly ask the reviewer to review the supp. material.

---

> > ### Comment · Reviewer_a9wE · 2023-11-22
> > **Response to authors**
> >
> > Thank you for your detailed responses. The concern on the presentation quality and technical correctness of Theorem 7 is much clarified. However, the main concern regarding the significance of contribution still remains largely unresolved. In particular, for optimization, the authors basically proposed to convert the original constrained form in Eq.(1) to an equivalent regularized form so that the ADMM/APG algorithms can be applied and the convergence guarantees in Theorem 9 can be established. Unless something missed here, the problem of regularized invex optimization has already been studied in an existing work of Pinilla et al. ( 2022a), and the present implementation just reiterates the algorithms and results therein. In this sense, it is not clear where the current work should be positioned with respect to the prior arts in literature.

---

> > > ### Author Response · Authors · 2023-11-22
> > > **Response to reviewer a9wE**
> > >
> > > Regarding the significance of the contribution, we would like to clarify following points: Firstly, casting a constrained problem to an equivalent regularized form is a standard procedure in optimization to analyze programs in the form of Eq. (1). In our case, this casting is to the form of ADMM/APG algorithms, which is the key challenge, which we believe, went undermined.  We have purposefully not covered these background as these are mostly seen as fundamental topics in optimization (see texts B1 and B2 below), and we assumed does not need to be covered when reviewed by expert reviewers. We are afraid the efforts of converting programs in the form of Eq. (1) to a regularized forms have been largely undermined or considered as trivial. This is a rather underpinning method that underpins several contributions in the optimization community. If such a mapping is considered as trivial, by the same token, a large body of works (see below) also had to be considered as trivial and must be dismissed - which will be very detrimental to the entire community where we are striving to innovate. Several bodies of work such as ((Parikh \& Boyd, 2014), (Beck, 2017), (Chen et al., 2001), (Cai \& Wang, 2011)) have paved a way for serious developments in optimization. For more details on this please see Chapter 5 of B1; or  book B2, which we believe would help establishing the significance of these tasks. We also would like to highlight the following references to highlight the weigh of the contributions in~((Zalinescu, 2014), (Sun \& Pong, 2022), (Mishra \& Giorgi, 2008)). If we had not cast Eq. (1) as we formulated, one should resort to an interior-point method, which would require an initial estimation of the underlying unknown signal $x$ that satisfies the constraint in Eq. (1). This is a non-trivial task (please see texts in (Parikh \& Boyd, 2014), (Beck, 2017) in Optimization and Signal Processing domains). Please see chapter 11 in B1 for more details. Even if undermined as a trivial approach, methods for estimating $x$ (which it self forms a whole domain of problems in various signal processing areas, such as target tracking) may be more limiting than solving the converted optimization problem that we are analyzing in this paper.
> > >
> > > Secondly, we recognise that appreciating the contribution and differences between this paper and (Pinilla, 2022a) requires considerable background knowledge, and we have made genuine efforts to highlight these differences (however subtle they are) very clearly in the last round of the response. In a nutshell, (Pinilla, 2022a) did not study quasi-invex objectives or invex data fidelity terms, which we believe are significant enough and cannot go unnoticed. Expressly, studies in (Pinilla, 2022a) are limited to five invex functions; here, we report a family of invex and quasi-invex functions suitable for optimization. Additionally, we proved in Theorem 3 that the sum of two of these invex/quasi-invex constructs is again invex/quasi-invex. As we pointed out in our manuscript, these clearly mark the contributions and in fact forms a basis for very significant breakthrough with the first family of invex/quasi-invex functions closed under summation. It is particularly important to recognise that these contributions extends what the community has enjoyed only with convexity hitherto. To be more explicit, this is not covered in (Pinilla, 2022a) or in the existing optimization literature. Lastly, results for Theorem 7, which is a counterpart to Theorem 4 (Pinilla, 2022a), are reported for the proposed family of invex/quasi-invex functions, including the noisy case. These results far exceed those presented in (Pinilla, 2022a), where the focus has only been on five invex functions and a noiseless case. This clearly highlights the differences and our paper here not just reiterates the algorithms and results therein. We use a number of common baselines to show the benefits of the applicability of the technique in the context of quasi invex case (so that they can be compared in a fair manner), which was not studied before. The results clearly show these. As such, we sincerely hope that these highlight the positioning of the work with the prior art and they are not mere reiteration of (Pinilla, 2022a). Our efforts here span well over a year since the breakthroughs highlighted in (Pinilla, 2022a), and very disheartened to see a review simply summing it up as "just reiterates the algorithms and results therein" than failing to see the real values therein.
> > >
> > > We hope these highlight the significance of the contributions, and we believe the values can be seen in combination of the background material below, namely, [B1, B2, (Parikh \& Boyd, 2014), (Sun \& Pong, 2022), (Mishra \& Giorgi, 2008), (Pinilla, 2022a)] than our work simply being dismissed at superficial level based on the misunderstanding or misinterpretations. Thank you again.
> > >
> > > [B1] Convex optimization
> > >
> > > [B2] First-order methods in optimization

---

### Official Review · Reviewer_ta5r · 2023-11-03

**Soundness:** 2 fair
**Presentation:** 3 good
**Contribution:** 2 fair
**Rating:** 5
**Confidence:** 3

**Summary:**

This paper studies the global optimality of a constrained nonconvex (particularly, invex/quasi-invex) optimization problem for signal restoration, which was found to provide an improved signal quality over the convex optimization-based signal restoration. The invexity property is interesting as it makes any critical point a global minimizer, while being quite nonconvex. This paper not only identifies the invexity of existing non-convex functions (for signal restoration) but also comes up with new invexity functions, based on a new definition of admissible function, yielding a global optimality guarantee. This paper then shows that the standard ADMM also works for constrained "invex" problems. Various experiments are provided to validate the effectiveness of using nonconvex but invex functions for both fidelity and regularization terms.

**Strengths:**

- This identifies various nonconvex but invex constrained signal restoration problems with a global optimality guarantee, theoretically supporting the empirical success of using nonconvex functions in signal restoration.
- This paper provides a set of admissible functions that are invex/quasi-invex, making it easy to identify or construct invex/quasi-invex functions.

**Weaknesses:**

- The global optimality of quasi-invex function was mentioned in the beginning, but it is not clearly stated after all.
- (Pinilla, 2022a) studied unconstrained signal restoration with invex regularizers, and the contributions of this paper over (Pinilla, 2022a), such as quasi-invexity and the assumptions on the extended ADMM (see question below), are not clearly stated. So they do not seem significant. (The fact that this paper considers a constrained case seems a straightforward extension of (Pinilla, 2022a).) I will reconsider my score depending on the clarification of these contributions.

**Questions:**

- Are the invex functions considered in this paper prox-friendly? In other words, how is the proximal step of ADMM implemented?
- Does Theorem 9 assume either invexity or quasi-invexity? I was not able to see where such property is used in the proof. It seems the prox-regularity condition is used, which is only a sufficient condition for quasi-invexity. If I am misunderstanding, I suggest revising this part
so that everything is easier to follow.

*Minor*
- abstract: largest "class of" optimizable
- page 1: $f(w) = f(Ax - y)$ should be fixed
- page 6: use "of" Lemma 1
- page 7: constrain"t"
- page 8: What do you mean by the uniqueness result in Theorem 3?

---

> ### Author Response · Authors · 2023-11-17
> **Response to reviewer ta5r (part 1)**
>
> Thank you for the time reviewing our manuscript. We appreciate the reviewer's advice to improve our manuscript. We address the comments about weaknesses. The changes to the manuscript and supplemental material are highlighted blue.
>
> # Discussion about weaknesses
> **W1.** As mentioned in the manuscript, Theorems 7, 8, and 9 present global optimality results that impact many applications for the family of quasi-invex functions stated in Theorem 3. More specifically, Theorem 7 and 8 include the sentences "We show that a similar result holds for quasi-invex constructs as in Theorem 3", and, "Consider the optimization problem in 11 for invex and quasi-invex TV regularizers as in Theorems 4 and 6, respectively", respectively. Moreover, in the case of Theorem 9, the sentence "Let $h_{1}(x)$, $h_{2}(z)$ be any of the constructs in Theorem 3,..." was intended to point out the systematic methodology for constructing invex/quasi-invex functions studied in this work in Theorem 3. With these, we assumed the notion of global optimality of the results for quasi-invex functions are obvious.
>
> We also remark that quasi-invex functions do not share the property that every local minimizer is a global minimizer (as in the case of invex functions). The global optimality of Equation 1 when $f(x)$, and $g(x)$ are the family of quasi-invex functions stated in Theorem 3 is covered in Theorem 9 (by solving the primal-dual problem). This statement is explained in Appendix I (see Section titled "Remarks on Equation 1"). For completeness, we write here how Equation 1 is expressed in the form needed for Theorem 9.
>
> We recall that Equation 1 is in the form of $$\text{minimize } g(x) \text{ subject to }f(x)<\epsilon.$$. This means Equation 1 can be equivalently rewritten as $$ \text{minimize } g(x) + \lambda f(x), $$ for some $\lambda > 0$ that helps to satisfy the condition $ f(x)<\epsilon$ (please see [R1, R2] provided below that supports this argument). Thus, by introducing a variable $z=x$, Eq. (1) can be finally rewritten as $$ \text{minimize } g(x) + \lambda f(z) \text{ subject to } z=x, $$ which is in the optimization form of Theorem 9, where $h_1(x)=g(x)$, and $h_{2}(z)=\lambda f(z)$.
>
> R1. Atomic decomposition by basis pursuit.
>
> R2.Orthogonal matching pursuit for sparse signal recovery with noise.
>
> Therefore, considering the above response and the reviewer's comment, we have modified the manuscript to make more explicit the global optimality results related to the family of quasi-invex functions in Section 4 and Theorem 9. We believe the statements of Theorems 7 and 8 clearly invoke the proposed family of quasi-invex functions, and thus, we are leaving their statements intact.
>
> **W2.** In order to answer the question about the significance and impact of our work over (Pinilla, 2022a), we list the notable differences between this work and (Pinilla, 2022a) as follows:
> - Studies in (Pinilla, 2022a) are limited to five invex functions; here, we report a family of invex and quasi-invex functions suitable for optimization.
> - Theorem 3 of this manuscript reports a systematic methodology for constructing invex/quasi-invex functions, an important aspect missing in (Pinilla, 2022a) and the current optimization literature.
> - We proved in Theorem 3 that the sum of two of these invex/quasi-invex constructs is again invex/quasi-invex. As we pointed out in our manuscript, this is a very significant breakthrough with the first family of invex/quasi-invex functions closed under summation identified and is the first family of functions beyond convexity that we can prove global optima in constrained optimization problems. This is not covered in (Pinilla, 2022a) or the existing optimization literature.
> - As highlighted here, we have extended the convergence guarantees of ADMM for the proposed family of invex/quasi-invex functions. This result is missing in (Pinilla, 2022a).
> - Theoretical results for Theorem 7 in our paper, which is a counterpart or equivalent to Theorem 4 (Pinilla, 2022a), are reported for the proposed family of invex/quasi-invex functions, including the noisy case. These results far exceed those presented in (Pinilla, 2022a), where the focus has only been on five invex functions and a noiseless case.
> - We report in Theorems 4 and 6 on how to build invex/quasi-invex-like total variation regularizers. This is a missing aspect not only in (Pinilla, 2022a) but also in the current optimization literature.
> - Collectively, the contributions here in this paper, the techniques more applicable to more practical cases, opposed to the technique presented in (Pinilla, 2022a).
>
> We have taken these comments and have modified the manuscript to clarify the contributions of this paper over (Pinilla, 2022a).

---

> ### Author Response · Authors · 2023-11-17
> **Response to reviewer ta5r (part 2)**
>
> We address the questions and the minor comments by the reviewer. The changes to the manuscript and supplemental material are highlighted blue.
>
> # Questions
> **Q1.** Yes, the invex functions considered in this paper are, indeed, prox-friendly. Realizing the significance of the implementation details, we have included additional details (as part of the Supp. Material) surrounding the implementation for the proximal expressions presented in Section I.3 of the Supp. Material. There we have presented as example the implementation of the proximal solution for the $\ell_{p}$-quasinorm. The text reads as follows:
>
> **Implementation details:** In order to efficiently compute the proximal solutions, which is an entry-wise operation, in Table 4, we use the Python library CuPy. The reason is that this library allows the creation of GPU kernels in the language C++ from Python (quick guide on how to create and use these kernels). Therefore, inside this GPU kernel, we implemented the entry-wise operation from the Proximal Operator column of Table 4. We remark that this implementation is efficient because it runs in parallel and at the GPU speed. Additional features of the CuPy library are the flexibility to connect both TensorFlow and PyTorch libraries for training neural networks.
>
> In addition to this, we have also highlighted this as part of the main text in Section 4 with the following modified text
>
> **Modified text:** Appendix I also provides the proximal solution for the studied invex/quasi-invex functions along with their computational time. These solutions are used to perform the numerical experiments in Section 5.
>
> Please see Appendix I for the implementation details around the proximal solution for the studied invex/quasi-invex functions.
>
> **Q2.** As this question concerns multiple aspects, please allow us to respond these in turn.
>
> To begin with, we would like to clarify that the statement of Theorem 9 includes "Let $h_{1}(x)$, $h_{2}(z)$ be any constructs in Theorem 3,...", which was meant to point out the systematic methodology for constructing invex/quasi-invex functions studied in this work in Theorem 3. As such, Theorem 9, indeed, assumes both invex and quasi-invex family of functions.
>
> Notwithstanding this, it is correct that prox-regularity implies merely quasi-invexity, as we stated in the manuscript. However, the critical disparity between our proposed invex/quasi-invex family of functions and prox-regularity is that prox-regularity does not imply uniqueness in the update steps for $x^{(t+1)}$ and $z^{(t+1)}$ in Equation (13), which is the key aspect to prove convergence of ADMM. In contrary, our family of invex/quasi-invex functions ensure uniqueness in the update steps for $x^{(t+1)}$ and $z^{(t+1)}$ as can be read in the proof of Theorem 9. This property ensures uniqueness and critical to the proof. This difference also highlights the importance of our proposed family of functions.
>
> We have now modified the manuscript to reflect  these changes and to make this aspect clearer in Section 4 and Appendix I. Below we show only the modified sentence in Section 4 due the space limitations, and we kindly ask the reviewer to see Appendix I of supp. material.
>
> **Added sentence at the end of Section 4:** Proof of Theorem 9 relies on Theorem 8, which ensures uniqueness in the estimations of $x^{(t+1)},z^{(t+1)}$ in Equation 13. This uniqueness is cannot be obtained by prox-regular functions, and thus highlights the significance of our proposed family of invex/quasi-invex functions.
>
> # Minor
> Thank you for your careful review. We have corrected all the typos pointed out by the reviewer. Additionally, we have changed the mentioned sentence to "the uniqueness result in Theorem 8".

---

> > ### Comment · Reviewer_ta5r · 2023-12-03
> >
> > Thanks for your response and revision, and I apologize for my late response. These clarified most of my concerns. I really appreciate the authors' hard efforts, but for the reasons below, I decided to increase my score to 5. I think this paper needs further clarification on their main contributions.
> >
> > **W1**
> >
> > - I agree that there already existed statements on finding a global solution for problems with "quasi-"invex functions. The revision now better explains this from the beginning. What I found confusing in the initial submission was that the term "the global optima guarantee" is somewhat vaguely used for both invex and quasi-invex cases. In particular, the former has the global optima guarantee for any optimization method that finds a stationary point due to Definition 3, while the latter only has such guarantee for the ADMM and APG (at the moment). So, I suggest the authors to better clarify this issue. What I also found awkward is that the result on finding a global solution of "quasi-"invex functions are given as a part of the Application section. If this is one of the three main contributions, doesn't it deserve a separate section? I think the authors can improve the structure of the paper
> > so that this main contribution is more apparent.
> >
> > - I just realized that Theorem 8 is stated for only invex/quasi-invex TV regularizers, so Theorem 9 only applies to the case where both $h_1$ and $h_2$ are invex/quasi-invex TV regularizers. I was expecting the theory to work for any invex/quasi-invex functions, and if that is the case, I suggest the authors to revise Theorems 8 and 9 accordingly. If not (the current proof of Theorem 8 explicitly requires $g$ being a TV regularizer), the authors should tone down one of the main contributions.
> >
> > **W2**
> > - Contributions compared to (Pinilla, 2022a) are now much clearer. Nevertheless, after reading reviewer a9wE's comment, I am curious whether this paper emphasizing on the "constrained" problem is valid enough. I agree with the authors' point that it is widely accepted to consider a regularized (11) for solving (1), but to the best of my knowledge, this only works when one can find an appropriate choice of regularization parameter. So, I don't think the authors' claim is not valid, but I think the emphasis on providing a theory on constrained problem should be toned down, since Theorem 9 is technically for solving an unconstrained (11).

---

### Meta-Review · Area_Chair_eZyA · 2023-12-07

**Metareview:**

This paper addresses the challenge of constrained optimization in signal restoration, where non-convex methods often outperform convex ones in reconstruction quality but lack global optimality guarantees. To address this, the paper introduces invex constrained optimization theory, ensuring that critical points serve as global minimizers. Furthermore, it extends this guarantee to a set of quasi-invex functions, enabling non-convex optimization to effectively handle constrained inverse problems. Experimental results indicate the promise of this approach, potentially enhancing existing convex optimization algorithms for tasks like signal restoration.


The paper underwent a thorough review process and received varying opinions among the reviewers. To provide a comprehensive evaluation, I have summarized the key comments and assessed the authors' responses:

1. Regarding the concern about the global optimality of quasi-invex functions mentioned at the outset, I believe the authors have effectively addressed this comment.
2. In response to a reviewer's request for a comparison with Pinilla, 2022a, the authors have clearly highlighted the distinctions in the revised paper. These extensions, particularly the extension of the class of invex functions and the extension of ADMM's convergence guarantee, are substantial and noteworthy.
3. A reviewer asked about the efficiency of finding optimal solutions for the introduced class of invex functions. The authors have indeed proposed two algorithms based on ADMM and proximal methods, both deeply rooted in classical convex optimization. While there may still be room for algorithmic improvements, the argument that these extended algorithms are, at minimum, as efficient as those in convex optimization is well-founded and reasonable.
4. The issue of clarity in proofs and statements, raised by two reviewers, appears to have been adequately addressed by the authors.

In my assessment, the paper's contributions are solid and exceed the acceptance criteria for ICLR. Despite initial reviewer reservations, I strongly recommend accepting the paper based on its significant contributions and the satisfactory responses provided by the authors.

**Justification For Why Not Higher Score:**

There are still some concerns raised by the reviewers that the authors need to address.

**Justification For Why Not Lower Score:**

As highlighted in my earlier meta-review, it's important to note that two out of three reviewers initially leaned toward rejecting the paper. However, in light of the varying opinions among the reviewers, I took the initiative to conduct a more thorough evaluation. After careful consideration, I believe that the paper's contributions outweigh its shortcomings. Consequently, I recommend accepting the paper.

---

### Decision · Program_Chairs · 2024-01-16

Accept (poster)